# Mutant APC reshapes Wnt signaling plasma membrane nanodomains by altering cholesterol levels via oncogenic β-catenin

Alfredo Erazo-Oliveras [1,2,3], Mónica Muñoz-Vega[1,2,3,10], Mohamed Mlih [4,10], Venkataramana Thiriveedi[5,6,7], Michael L. Salinas [1,2,3], Jaileen M. Rivera-Rodríguez[1,2,3], Eunjoo Kim[8], Rachel C. Wright[1,2], Xiaoli Wang[1,2], Kerstin K. Landrock[1,2], Jennifer S. Goldsby[1,2,3], Destiny A. Mullens [1,2,3], Jatin Roper[5,6,7], Jason Karpac [4] & Robert S. Chapkin [1,2,3,9] ✉

Although the role of the Wnt pathway in colon carcinogenesis has been described previously, it has been recently demonstrated that Wnt signaling originates from highly dynamic nano-assemblies at the plasma membrane. However, little is known regarding the role of oncogenic APC in reshaping Wnt nanodomains. This is noteworthy, because oncogenic APC does not act autonomously and requires activation of Wnt effectors upstream of APC to drive aberrant Wnt signaling. Here, we demonstrate the role of oncogenic APC in increasing plasma membrane free cholesterol and rigidity, thereby modulating Wnt signaling hubs. This results in an overactivation of Wnt signaling in the colon. Finally, using the *Drosophila* sterol auxotroph model, we demonstrate the unique ability of exogenous free cholesterol to disrupt plasma membrane homeostasis and drive Wnt signaling in a wildtype APC background. Collectively, these findings provide a link between oncogenic APC, loss of plasma membrane homeostasis and CRC development.

Colorectal cancer (CRC) is the 3rd most common type of cancer in the U.S. and accounts for an alarming 153,020 (8%) of new total cancer cases and 52,550 (9%) of total cancer deaths in 2021[1]. Overall, CRC incidence and mortality rates have decreased in the past 20 years, attributed largely to use of CRC screening and polypectomy in adults over 50 years of age. However, among younger adults, for whom screening is not recommended if at average risk, CRC incidence rates have been increasing by ~2% per year since 1994 in both men and women[1]. In the vast majority (>80%) of human sporadic CRC cases,

sequencing data indicate the presence of mutations in the adenomatous polyposis coli (APC) gene[2]. Mutations in APC (oncogenic APC) have also been identified in familial adenomatous polyposis (FAP) patients (>85%)[2,3]. This "gatekeeping" gene is a key regulator of Wnt signaling and ~90% of all human CRC cases are associated with defects in the Wnt signaling pathway[4]. Loss of APC canonical function causes aberrant stabilization of β-catenin (βcat), a crucial step in CRC initiation. Notably, attempts to target this pathway using drugs still pose multiple hurdles due to poor tumor cell targeting, negative side effects

[1]Program in Integrative Nutrition and Complex Diseases, Texas A&M University, College Station, TX 77843, USA. [2]Department of Nutrition, Texas A&M University, College Station, TX 77843, USA. [3]CPRIT Regional Center of Excellence in Cancer Research, Texas A&M University, College Station, TX 77843, USA. [4]Department of Cell Biology and Genetics, Texas A&M University, School of Medicine, Bryan, TX 77807, USA. [5]Department of Medicine, Division of Gastroenterology, Duke University School of Medicine, Durham, NC 27710, USA. [6]Department of Pharmacology and Cancer Biology, Duke University School of Medicine, Durham, NC 27710, USA. [7]Department of Cell Biology, Duke University School of Medicine, Durham, NC 27710, USA. [8]Division of Pulmonary Sciences and Critical Care Medicine, School of Medicine, University of Colorado Anschutz Medical Campus, Denver, CO 80045, USA. [9]Center for Environmental Health Research, Texas A&M University, College Station, TX 77843, USA. [10]These authors contributed equally: Mónica Muñoz-Vega, Mohamed Mlih. ✉e-mail: r-chapkin@tamu.edu

associated with required long-term treatments and poorly understood mechanisms of action[5]. More recently, there have been reports suggesting that oncogenic APC (truncated APC) displays multiple functions that drive CRC[6,7]. Most importantly, emerging mechanistic evidence indicates that oncogenic APC does not act autonomously and requires activation of additional key Wnt-associated modulators upstream of APC to drive aberrant Wnt signaling[8–13].

The distinct roles of extra- and intracellular modulators, e.g., βcat, T cell factor (TCF)/lymphocyte enhancer factor (LEF), Dishevelled (Dvl), Axin and Dickkopf (Dkk), in oncogenic APC-associated aberrant Wnt signaling have been extensively documented[14,15]. For example, in the presence of oncogenic APC, βcat stabilizes and accumulates to high levels in the cytosol, translocates to the nucleus and associates with TCF/LEF, leading to transcription of genes associated with CRC development. Aberrant Wnt signaling can be further enhanced or inhibited by inactivation or expression, respectively, of tumor suppressor proteins upstream of APC[9,14,16,17], such as cytoplasmic Dkk and secreted frizzled-related protein (sFRP), and by modifying the expression of non-canonical Wnt signaling members[18]. However, far less effort has focused on the role of the plasma membrane, which serves as a nexus integrating extra- and intracellular Wnt pathway modulators, in terms of oncogenic APC-driven aberrant Wnt signaling.

Plasma membrane biochemical and biophysical homeostasis, e.g., precise spatial compartmentalization of signaling components, regulates intracellular signal transduction[19]. Multiple signaling proteins, as well as several of their effectors, reside in highly ordered/rigid cholesterol-enriched lipid raft membrane domains (10–200 nm), including members of the canonical Wnt signaling pathway[20–24]. Moreover, factors associated with the Wnt signaling pathway require proper membrane organization within raft-like domains in order to become active and signal efficiently[25,26]. Thus, alterations in key membrane features, e.g., lipid raft membrane domains, upstream of APC may play an essential role in aberrant Wnt signaling and CRC, even in the context of a mutant APC background. Interestingly, it is also now appreciated that dysregulation of cellular cholesterol, a major component of lipid rafts, occurs in several cancers, including CRC[27]. Consistent with these reports, studies have demonstrated that cholesterol levels in tumor cells are elevated compared to healthy cells[28–30]. From a mechanistic perspective, unesterified "free" cholesterol is primarily localized to the plasma membrane, where it constitutes up to 90% of total cell cholesterol and 40–50 mol% of total plasma membrane lipid[31]. Recent evidence highlights the key role of free cholesterol as a genuine regulatory/signaling lipid that modulates the canonical Wnt pathway[32]. However, to date, the link between upstream dysregulation of plasma membrane biochemical/biophysical properties, e.g., free cholesterol homeostasis, mutant APC, and aberrant Wnt signaling, has not been investigated.

The Wnt signaling plasma membrane-localized single-membrane pass low-density lipoprotein receptor-related protein 5 or 6 (LRP5/6) and the G-protein coupled receptor protein Frizzled (Fzd) play a critical role in cell polarity[33], stemness[34,35], differentiation[36] and neoplastic transformation[34,35]. It is noteworthy that many proteins involved in CRC, including transmembrane receptors and G proteins, localize to cholesterol-enriched lipid rafts in order to form specialized signaling platforms, e.g., protein condensates or nanoclusters[20,37–42]. Nanoclusters are considered predominant features of the plasma membrane and appear to mediate critical signaling processes. For example, LRP5/6 and Fzd require lipid raft localization and nanoclustering (100–200 nm) for efficient signaling and most notably, stabilization of βcat[21,42]. Interestingly, cogent new evidence suggests that upstream Wnt receptor nanoclustering is required for cancer development on a mutant APC background in *Drosophila* and mammalian cells, further supporting the non-autonomous nature of mutant APC in CRC[10,13]. Protein condensates also play a critical role in Wnt signaling initiation, signal transduction, and downstream activation of Wnt effectors.

Recent evidence indicates that Dishvelled-2 (Dvl-2), a Wnt effector that is essential to relay the upstream signal from the Wnt ligand to downstream βcat activation, associates with Wnt receptors and other key Wnt effectors, e.g., Axin, to form protein condensates[41]. Subsequently, these protein condensates drive Wnt receptor phosphorylation, recruitment of Wnt effectors to the plasma membrane, Wnt signalosome endocytosis, and inhibition of destruction complex function, which collectively mediate downstream βcat activation[41]. Unfortunately to date, there is a paucity of data describing the mechanistic interactions associated with the upstream effects of oncogenic APC on plasma membrane Wnt-associated nanocluster/condensate structure and dysregulation of Wnt receptor activity in the context of CRC initiation. In the current study, we examined (i) whether oncogenic APC perturbs plasma membrane homeostasis, e.g., membrane free cholesterol and rigidity, and (ii) how loss of plasma membrane homeostasis influences crucial Wnt signaling-associated plasma membrane spatiotemporal properties, e.g., organization of Wnt nanocluster protein/lipid components and downstream signaling. Herein, we report that oncogenic APC increases plasma membrane free cholesterol, thereby promoting membrane rigidity and lipid raft stability. In addition, we show that APC-driven dysregulation of plasma membrane homeostasis modulates Wnt receptor nanoscale proteolipid organization and their interactions with key Wnt signaling effectors. Finally, we demonstrate that dysregulation of cellular free cholesterol, alone, is sufficient to disrupt Wnt signaling. Together, these insights suggest a central role for mutant APC in reshaping plasma membrane Wnt receptor proteolipid nanodomains and downstream signaling. This knowledge will serve as a foundation to develop new drug targets and membrane-targeted chemopreventive strategies against CRC.

## Results

### Aberrant Wnt signaling increases plasma membrane free cholesterol in colonocytes through alterations of molecular pathways associated with cholesterol homeostasis

Genetic studies demonstrate that both *Apc* alleles are modified in CRC cells via mutations or loss of heterozygosity (LOH)[43]. Notably, the majority of these gene modifications do not result in complete loss of the APC protein but rather the generation of stable truncated gene products. *Apc* mutations occur with the highest frequency within a domain of the gene known as the mutation cluster region (MCR). As shown in Fig. 1A, a staggering 90% of all mutations are truncating in nature. Therefore, we employed a series of mouse and human CRC models expressing various oncogenic truncated APC proteins (Fig. 1B) to investigate their effects on plasma membrane cholesterol homeostasis. The levels of total cellular cholesterol in CRC tumors have been previously characterized. Prior studies in both serum and tissue biopsies show an increase in total (free and esterified) cholesterol, which has been associated with an elevation in CRC risk[44–46]. More recent evidence demonstrates that unesterified "free" cholesterol acts as a modulatory signaling lipid that can selectively activate canonical Wnt signaling[32]. Together, these findings suggest free cholesterol might play a key role in CRC. Surprisingly, to date, the effects of oncogenic truncated APC on plasma membrane free cholesterol have not been determined. To highlight the physiological relevance of our studies, we confirmed the expression of various oncogenic truncated APC gene products in our CRC models by western blot analysis employing a primary antibody that specifically detects truncated APC proteins (Fig. 1C). Although the specific sequence of the expressed *Apc* transcript in the tumor-derived and Crispr engineered human colon organoid models was not characterized at the nucleic acid level, western blot analysis clearly showed the presence of an APC gene product similar in size to the truncated APC expressed in the well-known CRC cell line models. Therefore, it is reasonable to conclude that our mutant APC organoid model expresses a truncated APC protein

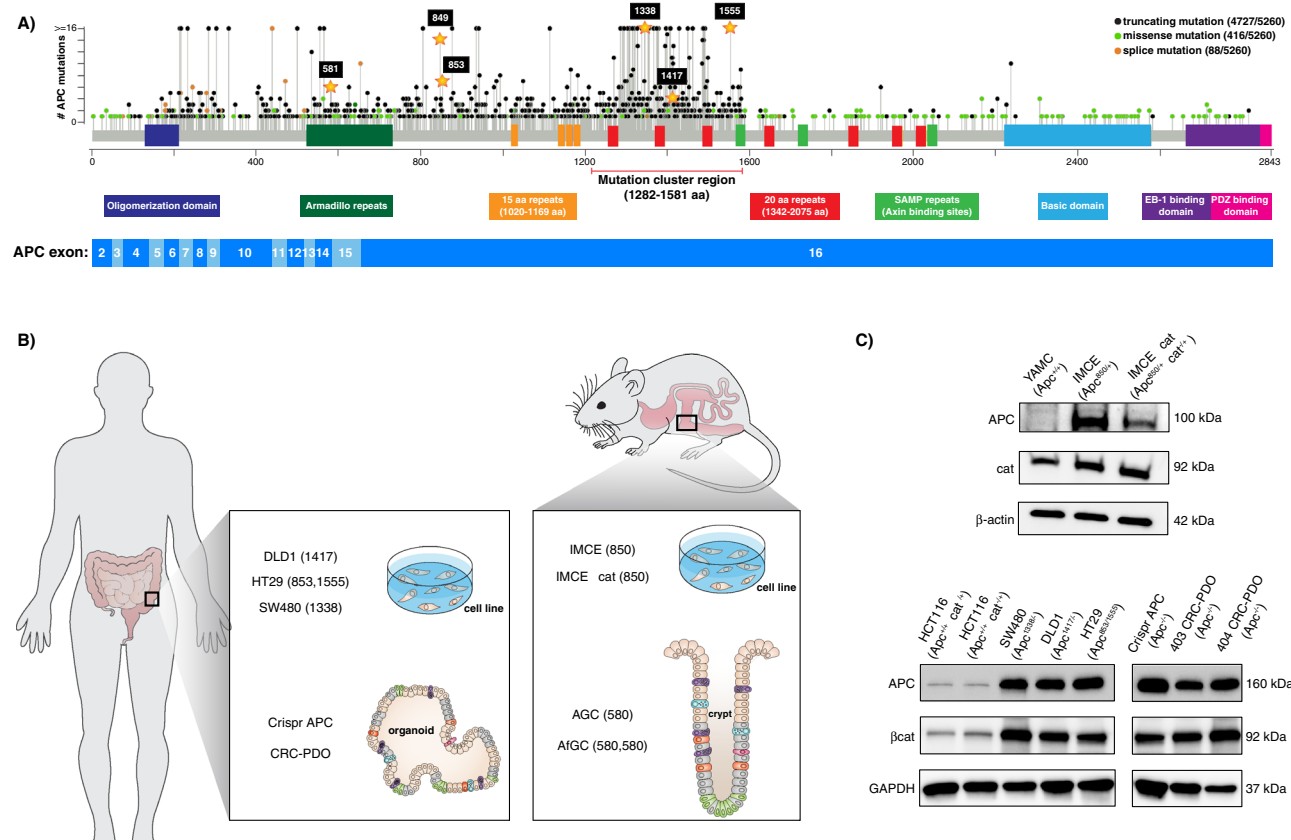

**Fig. 1 | Oncogenic truncated APC topology in CRC models. A** Lollipop plot displaying the distribution and classes of mutations in the APC protein sequence across multiple bowel-associated CRC datasets in the cBioPortal for cancer genomics (https://www.cbioportal.org). Key mutations utilized throughout our studies are highlighted. The functional domains in the APC sequence are also highlighted and matched with their respective coding exons. cBioPortal was employed to create the illustrative depiction of the APC protein. **B** Summary of all the oncogenic truncated APC models utilized herein. **C** Characterization of truncated Apc gene products. Lysates were obtained from mouse (YAMC, IMCE, and IMCE βcat) and human (HCT116Δ, HCT116 SW480, DLD1, and HT29) cultured cells as well as patient-derived organoids (PDOs) and detected with a primary antibody against truncated APC. As controls, primary antibodies against βcat and housekeeping genes, β-actin and GAPDH, were utilized. Two independent western blots were performed two times displaying similar results. Source data are provided as a Source data file.

functionally analogous to the truncated APC proteins produced by the CRC cell lines. As previously shown in the literature, the expression of oncogenic truncated APC increased the stabilization of βcat (Fig. 1C). Interestingly, the βcat protein levels in HCT116Δ (Apc +/+ βcat Δ/+) and HCT116 (Apc +/+ βcat −/+) cells were significantly lower in comparison to oncogenic truncated APC-expressing cells (Fig. 1C).

In order to characterize the plasma membrane-disrupting activity of oncogenic truncated APC, we first examined the levels of free cholesterol in an in cellulo CRC model expressing mutant APC^Min (APC 850), considered to be well-suited for the study of oncogenic APC function in intestinal tumorigenesis[47]. To quantify plasma membrane free cholesterol, isogenic young adult mouse colonic epithelium (YAMC) (Apc +/+) and Immortomouse−MIN colonic epithelial (IMCE) (Apc 850/+) colonic cell lines and their derived giant plasma membrane vesicles (GPMVs) were treated with filipin III, a fluorescent molecule that specifically binds free cholesterol[48]. Subsequently, plasma membrane filipin III fluorescence was quantified using imaging flow cytometry and confocal microscopy. Mouse colonocytes expressing oncogenic truncated APC (IMCE) exhibited increased filipin III staining when compared to wild-type (WT) APC (YAMC) (Fig. 2A, B, Fig. S1A, S1B). In addition, oncogenic truncated APC induced the loss of cholesterol homeostasis in several authentic human colorectal adenocarcinoma cell line models. For example, SW480 (Apc 1338/−), DLD1 (Apc 1417/−), and HT29 (Apc 853/1555) CRC cells, expressing different truncated forms of APC[49], displayed increased free cholesterol

compared to WT APC-expressing HCT116 (Apc +/+ βcat −/+) and HCT116Δ (Apc +/+ βcat Δ/+) CRC cells (Fig. 2C). Notably, the large increase in plasma membrane cholesterol observed in CRC cultured cells was observed in the absence of stimulation with Wnt3a ligand, highlighting the important role of constitutive activation of Wnt signaling in this CRC model when compared to "healthy" cells expressing WT APC. To further validate the effects of oncogenic truncated APC on plasma membrane free cholesterol levels without potential interference from intracellular membranes, GPMVs were isolated from colonic cell lines as described previously[50] and stained with filipin III. GPMVs are microscopic (~5–15 μm) plasma membrane spheres harvested from live cells following chemical treatment. This model has been used to probe multiple plasma membrane biological processes concurrently minimizing the number of cellular variables, e.g., cytoskeleton and cytosolic organelles, thus decreasing experimental complexity, while retaining compositional complexity, protein content and, to an extent, protein native topology[51]. Consistent with our findings using intact cells, GPMVs derived from mouse and human colonocytes expressing oncogenic truncated APC exhibited increased filipin III staining compared to colonocytes expressing WT APC (Fig. 2D, E). Interestingly, in complementary "gene dose" experiments using IMCE βcat, an isogenic line derived from IMCE that expresses oncogenic APC and oncogenic βcat (Apc −/+ βcat −/+)[52], the levels of free cholesterol were proportional to the magnitude of Wnt signaling dysregulation (Apc −/+ βcat −/+ > Apc −/+ > Apc +/+) (Fig. 2A, B, E). It is

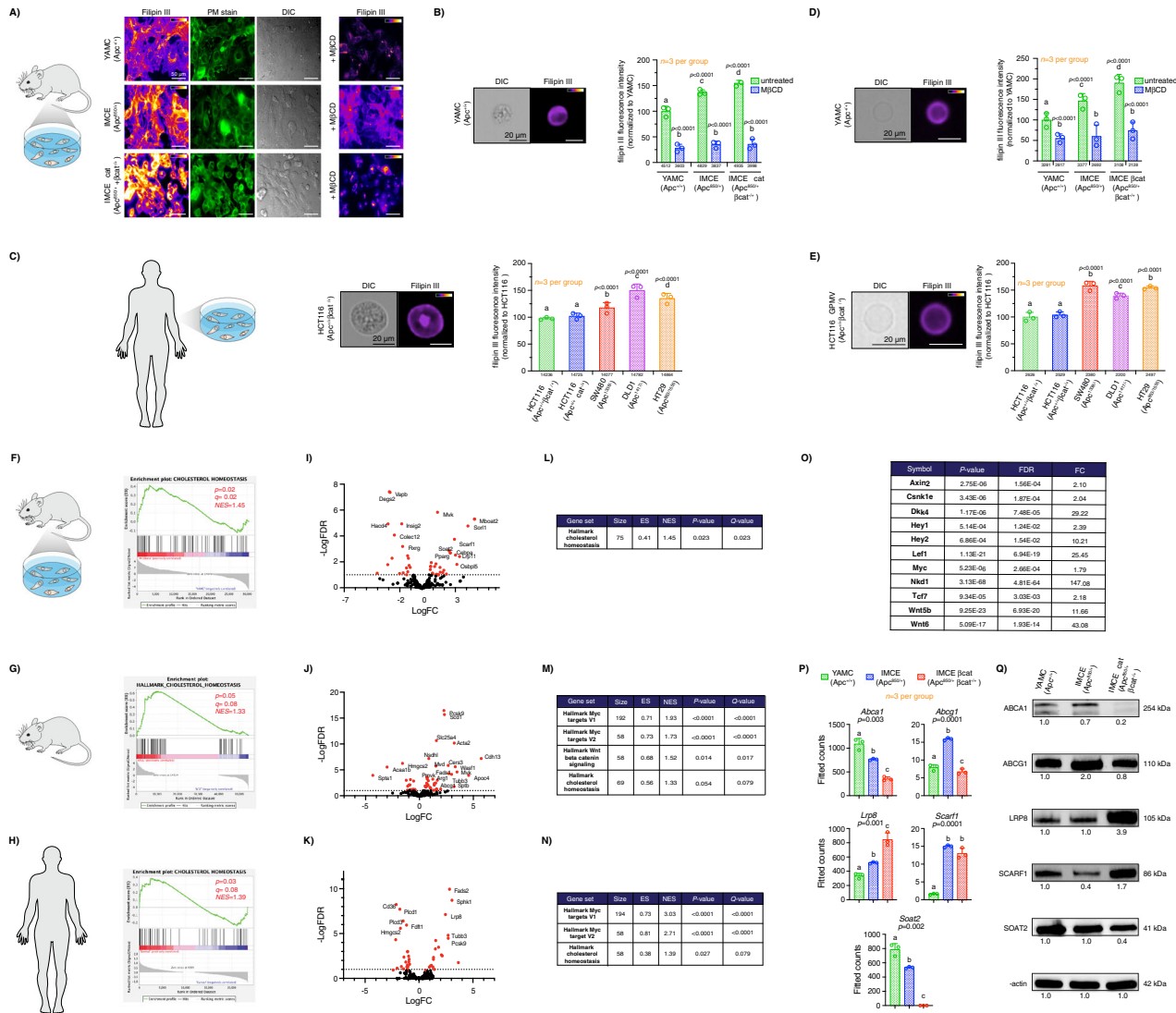

**Fig. 2 | Oncogenic truncated APC disrupts plasma membrane cholesterol homeostasis. A** Representative images of colonocytes stained with filipin III and PM stain. Scale bar: 50 μm. Quantitative analysis of cholesterol levels in **B** mouse-, **C** human CRC-cultured colonocytes, **D**, **E** their derived GPMvs and their respective representative flow cytometry images. Scale bars: 20 μm. Quantitative analysis of cholesterol levels in GPMVs. Error bars represent cells or GPMVs from $n = 3$ independent biological replicates (mean ± SD). Number of events analyzed using flow cytometry is shown below each bar. Statistical significance was determined by **B**, **D** two-way ANOVA or **C**, **E** one-way ANOVA and post Tukey's multiple comparison test. RNAseq analysis from **F** IMCE βcat and YAMC colonocytes ($n = 3$ independent biological replicates per group), **G** mouse colonic crypts isolated from AfGC and GC mice ($n = 4$ mice per group, equal number of males (♂) and females (♀)), and **H** $n = 36$ human paired samples (18 normal and 18 CRC). **F**–**H** Enrichment of the cholesterol homeostasis pathway genes in AfGC compared to GC samples. **I**–**K** Volcano plot illustrating differentially expressed genes (FDR ≤ 0.05; top 15

genes are listed) (ES, enrichment score; FDR, false discovery rate; FC, fold change). **L**–**N** Gene set enrichment analysis in AfGC compared to GC samples. Gene sets were ranked by normalized enrichment score (NES). **O** Differentially expressed genes corresponding to Wnt/βcat signaling. For all RNA expression experiments, statistical significance was determined by EdgeR-robust in several contrasts and Benjamini-Hochberg (BH) FDR ($P < 0.05$). **P** Normalized fitted counts showing mRNA levels of differentially expressed genes associated with cholesterol efflux, uptake, and esterification assessed by RNAseq analysis. Statistical significance determined by two-way ANOVA and post Tukey's multiple comparison test. Error bars represent $n = 3$ independent biological replicates (mean ± SD). **Q** Validation of RNAseq data via western blot normalized to β-actin. Results were used to calculate a relative ratio using YAMC (Apc +/+) as a control. In all cases, when provided, different letters indicate significant differences between WT APC (control) and treated/mutant APC (experimental) groups ($P < 0.05$). Source data are provided as a Source data file.

also noteworthy, that oncogenic βcat in the presence of wild-type APC (HCT116; Apc+/+ βcat−/+) was not sufficient to dysregulate plasma membrane cholesterol in cells when compared to colonocytes expressing oncogenic truncated APC or the oncogenic APC-βcat compound mutants (Fig. 2A−E). This is consistent with its low βcat protein levels (decreased downstream Wnt signaling) (Fig. 1C), and suggests that a relatively higher level of downstream Wnt oncogenic signaling is associated with the presence of truncated APC, which is essential for plasma membrane cholesterol dysregulation. Lastly, to increase the level of experimental rigor and corroborate our filipin III

experiment findings, we examined the level of total cholesterol employing the Amplex™ Red assay. This cholesterol assay is based on an enzymatic reaction that targets the conversion of cholesterol and couples a reaction byproduct as well as Amplex™ Red to the stoichiometric conversion of the latter into a fluorescently-active molecule, i.e., resorufin, with high specificity, thus enabling the quantitative analysis of cholesterol levels[53]. Consistent with our filipin III findings, oncogenic truncated APC increased the level of total cholesterol in IMCE and IMCE βcat colonocytes when compared to YAMC cells expressing WT APC (Fig. S1C, S1D).

To provide insight into the cellular processes altered by oncogenic truncated APC which could perturb cholesterol homeostasis, we performed a bulk RNA-Seq transcriptional analysis of mouse cultured colonocytes (YAMC, IMCE, and IMCE βcat) and mouse colonic crypts (AGC and AfGC). RNA-Seq identified global changes in gene expression of mouse cultured colonocytes and colonic crypts expressing homozygous truncated APC when compared to WT APC) (Fig. 2F, G). Gene set enrichment analysis (GSEA) of normalized counts of all genes in cells expressing oncogenic truncated APC compared with those in control conditions expressing WT APC using the Hallmark database showed that genes from signaling pathways associated with cholesterol homeostasis, e.g., cholesterol uptake, efflux, and synthesis, were significantly enriched (FDR = 0.023 and 0.079) in truncated APC-expressing cultured cells and crypt tissue, respectively (Fig. 2F, G). Interestingly, more modest effects were observed in heterozygous truncated APC models, e.g., hallmark cholesterol homeostasis genes were not significantly enriched in (Apc 580/+) mice (FDR = 0.403) (Fig. S2A, S2B). Nonetheless, a mutation on just one Apc allele was sufficient to enrich βcat-related genes, suggesting that downstream Wnt signaling was activated (Fig. S2C, S2D).

In human CRC, the vast majority (>75%) of colorectal tumors display biallelic mutations on the APC gene[54], similar to our employed oncogenic truncated APC models. To further assess the translational importance of our findings, we examined the expression of cholesterol homeostasis-related genes in colonic tumor and matched normal (healthy) tissue samples obtained from CRC patients. Overall, the expression of genes associated with cholesterol homeostasis were dysregulated in colon tumor tissue. For example, genes associated with cholesterol synthesis and uptake were upregulated in tumors relative to matched healthy tissue (Fig. 2H, K, N, Fig. S1E and Table S1). Interestingly, we observed a commonality between human colon, mouse crypts, and cultured colonocyte (IMCE and IMCE βcat) transcriptomes (Fig. 2F–N and Table S2) in terms of loss of cholesterol homeostasis, e.g., Abca1, Lrp8, Mvk, Orl1, Soat2, and Sorl1 genes were dysregulated in all datasets. The distribution of all differentially expressed genes (DEGs) obtained for each treatment are shown in volcano plots (Fig. 2I, J, K), mapped by Log2 Fold Change and adjusted p-value, with various genes displaying the most significant and highest expression changes including Pmvk, Mvk, and Mvd (cholesterol de novo synthesis), Lrp8 and Scarf1 (cholesterol uptake) and Abca1 and Abcg1 (cholesterol efflux). Notably, enrichment of genes associated with stem cell homeostasis, i.e., Myc and βcat signaling, which is often documented in mutant APC models, was also observed in oncogenic truncated APC-expressing cells and tumor tissue further highlighting the physiological relevance of our data (Fig. 2M–O).

Since oncogenic truncated APC increased free cholesterol in both mouse and human cultured colonocytes, we subsequently focused on various key genes linked to disturbances in cholesterol homeostasis. Notably, the directionality and fold change for these altered genes in oncogenic truncated APC-expressing cells were consistent with the notion that the elevation in plasma membrane free cholesterol is dependent on the degree of Wnt signaling dysregulation (Fig. 2P and Table S2). For example, IMCE βcat displayed a striking decrease in the expression of genes associated with cholesterol efflux, e.g., Abca1[30] and Abcg1[55] and an increase of genes associated with cholesterol uptake and availability, e.g., Lrp8[56] and Scarf1[57], as well as other important genes involved in cholesterol homeostasis (Fig. S3), when compared to IMCE and YAMC cells (Fig. 2P and Table S2). Interestingly, we also observed an increase in the expression of genes linked to intestinal stem cell (ISC)/progenitor cell stemness and tumorigenesis in a lipid-rich diet background[58]. Finally, the changes in RNA expression levels of a select group of genes were confirmed at the protein level by western blot (Fig. 2Q). Overall, oncogenic truncated APC was associated with changes in genes associated with cholesterol homeostasis and, in multiple cases, these changes were proportional to the magnitude of Wnt signaling dysregulation. Interestingly, these findings are consistent with the "gene dose" effect of mutant Apc (Apc 850/+ βcat −/+ > Apc 850/+ > Apc +/+) on plasma membrane free cholesterol (Fig. 2A, B, D). Together, our preclinical and clinical findings suggest a direct role for aberrant Wnt signaling in the loss of plasma membrane cholesterol homeostasis.

We next investigated the global changes in the expression of genes involved in cholesterol homeostasis, driven by oncogenic truncated APC, were functionally relevant at the cellular/molecular level. Typically, cholesterol levels are exquisitely regulated in cells. The abundance of cholesterol in normal cells is intrinsically dependent on three discrete processes: active cholesterol uptake from the extracellular environment, e.g., LDL particles, de novo cholesterol synthesis, which is tightly regulated by sterol regulatory element-binding proteins (SREBPs) sensing the overall levels of intracellular cholesterol, and transport of excess cellular cholesterol outside the cell by HDL particles via cholesterol efflux (Fig. 3A). As shown in Fig. 2 and Fig. S3, RNAseq data from cultured cells expressing oncogenic truncated APC, i.e., IMCE and IMCE βcat, indicate that cholesterol genes associated with uptake were significantly upregulated, while genes involved in cholesterol de novo synthesis and efflux were significantly downregulated when compared to WT APC-expressing cells. Thus, we took advantage of this apparent dependence of cultured mutant APC cell lines on the expression of genes associated with cholesterol uptake to further examine their functional relevance (Fig. 3B). To answer this question, YAMC (Apc +/+), IMCE (Apc 850/+) and IMCE βcat (Apc 850/+ βcat −/+) cells were cultured under lipoprotein depleted (LD) conditions (Fig. 3C) and cholesterol was examined using filipin III and flow cytometry. After starving cells of lipoproteins for 24 h, we observed a decrease in the levels of plasma membrane cholesterol (Fig. 3D). Notably, following 72 h of lipoprotein cellular starvation, cells displayed a further decrease in plasma membrane cholesterol, which suggests an impaired ability of these cells to regulate cholesterol levels (Fig. 3E). Interestingly, the change in cholesterol levels under lipoprotein depleted conditions was significantly greater in cells expressing oncogenic truncated APC when compared to WT APC and control conditions (Fig. 3F, G). Together, these results corroborate the functional relevance of the changes observed in the expression of cholesterol homeostatic genes driven by oncogenic truncated APC.

## Oncogenic APC induces loss of cholesterol homeostasis in the colonic crypt

It is widely reported that experimental outcomes observed from in cellulo models do not always translate to in vivo systems[59]. Therefore, in order to verify that the plasma membrane-disrupting activity of oncogenic APC is physiologically relevant, we examined the levels of plasma membrane free cholesterol in colonic crypts from mice expressing oncogenic truncated APC (APC 580) in the intestinal tract, i.e., inducible CDX2P^ΔCreERT2 x Lgr5-EGFP^ΔCreERT2 x Apc het (Apc 580/+) (AGC het) and homo (Apc 580/580) (AfGC homo) (Fig. 4A). As shown in the literature, induction of oncogenic APC expression in the mouse intestine leads to the development of colorectal adenomas and carcinomas with features akin to those observed in human colorectal lesions. Thus, AGC het and AfGC homo mice serve as versatile translational models of CRC progression[54]. Following activation of Cre recombinase with tamoxifen, recombination of the floxed Apc allele and subsequent oncogenic truncated APC expression (Fig. 4A and Fig. S4)[60], mice were terminated at the indicated times (Fig. 4B), and live crypts were isolated from colon tissue, stained with filipin III and imaged to quantify free plasma membrane cholesterol (Fig. 4C). As expected, AfGC homo mice (highest Wnt signaling dysregulation) developed a pronounced colonic polyposis following activation of oncogenic truncated APC expression (Fig. 4D). Moreover, oncogenic truncated APC increased colon and cecal weights in AfGC homo mice (Fig. S5) as well as colonic crypt length (Fig. 4E) and the degree of

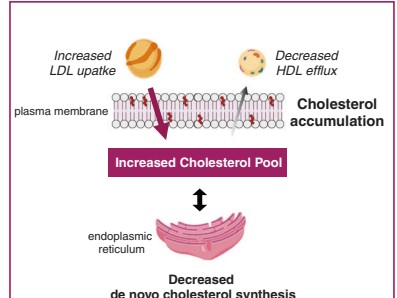
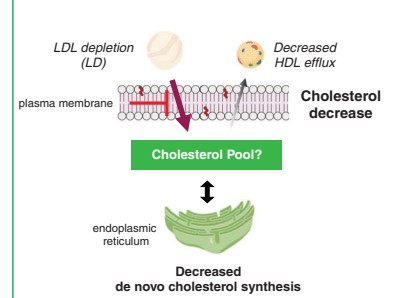

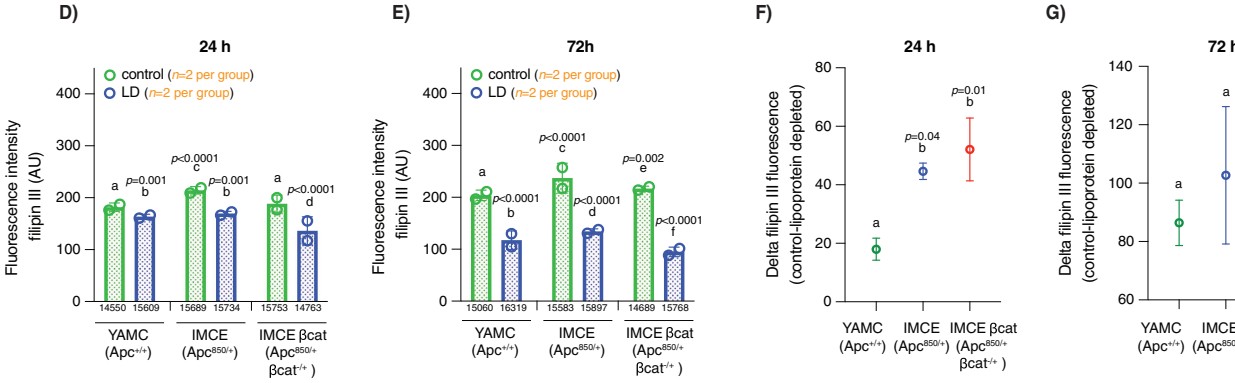

**Fig. 3 | Cholesterol uptake-related genes modulate plasma membrane biochemical composition in cells expressing oncogenic APC. A** In "normal" YAMC (Apc +/+) cells, membrane cholesterol homeostasis is regulated by cholesterol uptake (mainly from LDL) via the endocytic pathway, de novo cholesterol synthesis in the endoplasmic reticulum (ER) and cholesterol efflux via HDL. Collectively, these steps are tightly regulated to maintain "healthy" levels of cellular cholesterol. **B** In "deranged" cells, mutant APC perturbs cholesterol uptake, synthesis, and efflux, thus perturbing cholesterol homeostasis leading to changes in the pool of cellular cholesterol. **C** A putative model demonstrating the consequences of depleting extracellular cholesterol-rich LDL. This hypothetical paradigm should substantially reduce cholesterol availability for cellular endocytic uptake. To assess the contribution of exogenous cholesterol to the plasma membrane (PM), cells were maintained in LDL-depleted (LD) culture media for 24 or 72 h. **D, E** Quantification of PM cholesterol using filipin III fluorescence. **F, G** Change in filipin III fluorescence intensity. To quantify changes in PM cholesterol, cells cultured under LD conditions were compared to control (delta filipin fluorescence intensity). Filipin fluorescence intensity was determined from filipin III fluorescence images (mean ± SEM, $n = 2$ independent biological replicates, exact number of cells analyzed per condition is shown below each bar). Statistical significance was determined by two-way ANOVA and post Tukey's multiple comparison test. Different letters indicate significant differences between WT APC (control) and treated/mutant APC groups (experimental) ($P < 0.05$). Illustrations were created with BioRender.com. Source data are provided as a Source data file.

inflammation, tissue injury, epithelial cell proliferation, and mucin staining (Fig. S6A). However, AGC het mice exhibited no significant increase in polyp formation as compared to control GC (WT APC) mice in the allotted 10-wk duration of the study (Fig. S6B). Interestingly, colonic crypts from both AGC het and AfGC homo mice exhibited increased filipin III staining when compared to GC mice (Fig. 4F).

To further validate the effects of oncogenic truncated APC on plasma membrane free cholesterol levels without potential interference from intracellular membranes, GPMVs were isolated from colonic crypts and stained with filipin III (Fig. 4G). We clearly demonstrate that GPMVs bud from the membrane of crypts to form spherical structures displaying strong membrane filipin III staining, while excluding internal membranous staining (Fig. 4G, top panel). In contrast, intact single colonocytes in the vicinity of GPMVs exhibited both plasma and internal membrane filipin III staining, further verifying the fidelity of the GPMV model system. Consistent with our findings using intact crypts, GPMVs derived from mouse colonic crypts expressing oncogenic truncated APC exhibited increased filipin III staining compared to GPMVs expressing WT APC (Fig. 4G). Overall, a marked oncogenic APC gene dose effect on plasma membrane free cholesterol was observed (double mutant allele Apc 580/580 > single mutant allele Apc 580/+ > WT Apc +/+) (Fig. 4F), similar to cultured colonocytes (Fig. 2).

Central to CRC initiation is the concept that tumor development is fueled by a dedicated group of cells, i.e., cancer stem cells (CSCs)[61,62]. Colonic CSCs arise following gene mutations and/or dysregulation of key genetic programs in their healthy counterparts and are considered the "beating heart" of tumors. Since intestinal stem cells (ISCs) are exquisitely sensitive to malignant transformation by mutations in the Wnt pathway[34,63], as compared to other cell types in the crypt, we quantified plasma membrane free cholesterol in ISCs expressing oncogenic APC. Colonic stem cells from AfGC homo mice expressing the fluorescent reporter GFP under the control of the stem cell-specific marker gene *Lgr5* (Lgr5[+]) were isolated using fluorescence-activated cell sorting (FACS), stained with filipin III and analyzed using flow cytometry. As expected, expression of oncogenic APC increased the percentage of Lgr5[+] stem cells in crypts from AfGC homo mice (Fig. S6C), reminiscent of intestinal crypt malignant transformation[64]. Importantly, Lgr5[+] stem cells from AfGC homo (oncogenic APC) colonic crypts exhibited increased free cholesterol staining when compared to GC (WT APC) colonic crypts (Fig. 4H). Furthermore, the levels of plasma membrane free cholesterol in colonic stem cells increased over time following activation of oncogenic APC. These data suggest that loss of plasma membrane free cholesterol homeostasis occurs in ISCs, potentially marking an early step associated with malignant transformation of the colonic mucosa.

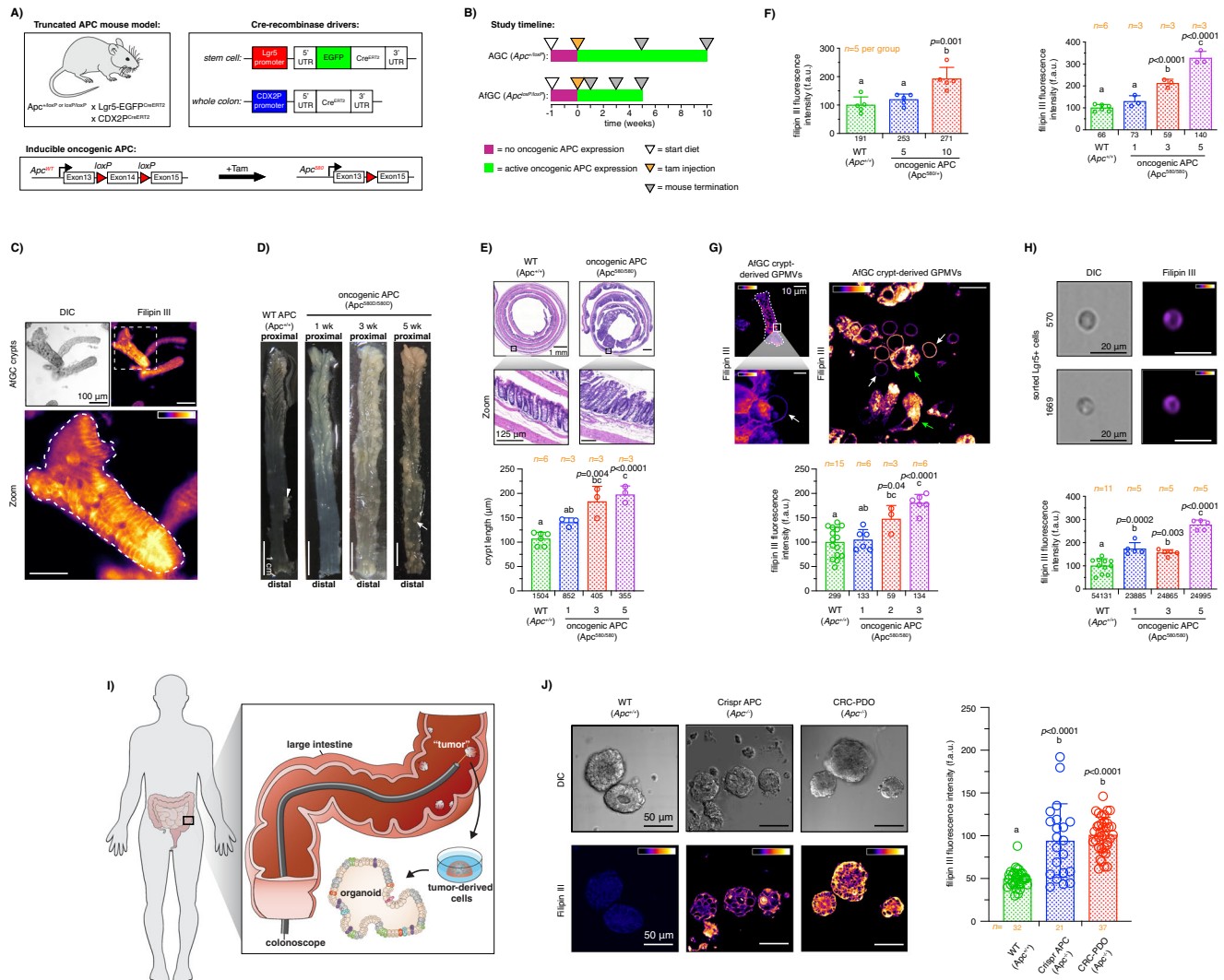

**Fig. 4 | Oncogenic truncated APC drives dysregulation of free cholesterol homeostasis in vivo. A** CRC mouse model expressing oncogenic APC. **B** Mouse experimental design. **C** Representative images of crypts stained with filipin III from AfGC homo mice. Scale bars: 100 μm. **D** Colon tissue from GC and AfGC homo mice exhibiting polyposis. Arrowhead, mesenteric adipose tissue; arrow, polyp. Scale bars: 1 cm. **E** H&E staining of colonic swiss roll (top) and quantitative analysis of crypt length. Scale bars: 1 mm; zoom, 125 μm. **F** Whole colon quantitative fluorescence intensity analysis of filipin III in GC, AGC het, and AfGC homo. **G** Crypt-derived GPMV quantitative analysis of plasma membrane free cholesterol levels. Representative filipin III fluorescence images of crypt from AfGC (top left), formation of a GPMV (blebbing, white arrow) from a crypt (bottom left), mixed GPMVs (white arrow), and whole cells (green arrow) (top right), and quantitative analysis of GPMV filipin III fluorescence intensity (bottom). Scale bars: 10 μm. **H** CSC quantitative analysis of filipin III in AfGC homo mice. Representative images of filipin III-

stained CSCs (top) from AfGC homo and filipin III quantitative analysis (bottom). Scale bars: 20 μm. **I** Representation of colon tumor biopsy harvesting a patient's tumor to generate CRC-PDOs. **J** PDO quantitative analysis of plasma membrane free cholesterol. Representative images of PDOs and filipin III quantitative intensity analysis. Scale bars: 50 μm. For all experiments, error bars represent **E, F** crypts from n = 3–6 mice, **E** (8♂,7♀), **F** (15♂,15♀), **G** crypt-derived GPMVs from n = 3–15 mice (15♂,15♀), **H** CSCs from n = 5–11 mice (13♂,13♀), normalized to WT APC mice or **J** organoids from n = 21–37 fields of view (FOV) (2–3 PDOs per FOV) (mean ± SD, exact n value is shown in each graph). The exact total number of crypts, GPMVs, CSCs, and FOVs analyzed are provided below each bar. For all experiments, statistical significance was determined by one-way ANOVA and post Tukey's multiple comparison test. Different letters indicate significant differences between WT APC (control) and mutant APC groups (experimental) at each time point (P < 0.05). Source data are provided as a Source data file.

Patient-derived organoids (PDOs) are three-dimensional (3D) models are established from both cancer and "healthy" human tissue samples obtained from different organs, including the large intestine[65]. PDOs hold remarkable resemblance with many of the structural and functional features associated with their organ of origin[65]. Colonic PDO models can therefore be employed to mimic various pathologies of the colon, such as CRC, "in a dish" and gain insights into different mechanisms of colon tumorigenesis. In order to validate the physiological relevance and impact of our findings, we examined the effect of oncogenic truncated APC on plasma membrane cholesterol using colonic PDOs. PDOs were grown from healthy and cancerous, i.e.,

cancer tumor, human tissue obtained from colon biopsies (Fig. 4I). Furthermore, since CRC-PDO could potentially harbor mutations in various genes other than *Apc*, we generated an isogenic PDO model expressing mutant APC by editing the WT *Apc* gene in healthy PDOs using CRISPR-Cas9[66]. Following filipin III staining and imaging, we found that Crispr APC and CRC-PDOs exhibited an increase in plasma membrane free cholesterol when compared to WT APC PDOs (Fig. 4J). Overall, these data indicate that oncogenic APC dysregulates free cholesterol homeostasis both in mice and humans and that loss of membrane cholesterol homeostasis could play a key role in the modulation of colon tumorigenesis.

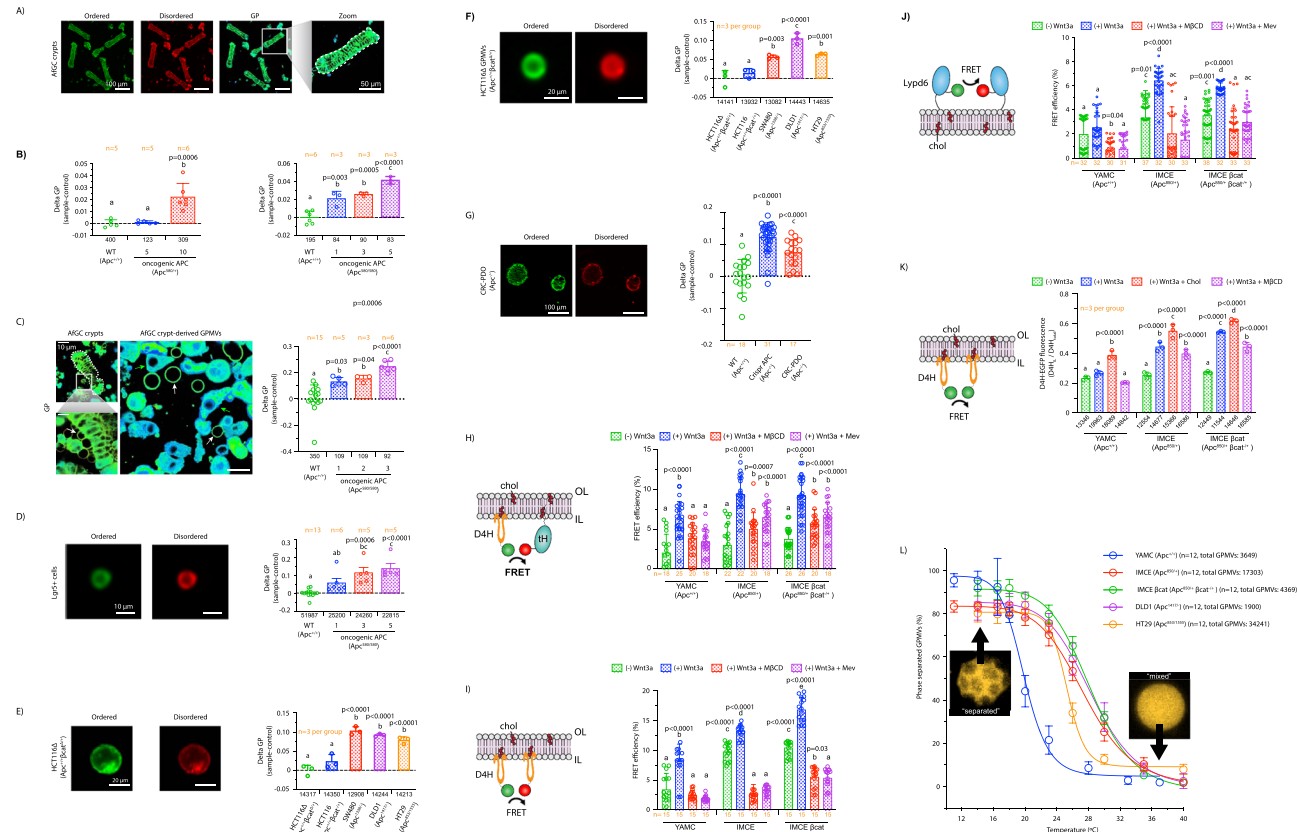

**Fig. 5 | Oncogenic truncated APC modifies the organization of Wnt signaling plasma membrane domains. A** Di-4-stained crypts from AfGC mice. **B** Quantitative analysis of di-4 from crypts and **C** their derived GPMVs (blebbing, left white arrow; GPMVs, right white arrow; whole cells green arrow), **D** CSCs, **E** human CRC cells and **F** their derived GPMVs, and **G** PDOs. For all d-i4 experiments, error bars represent **B** n = 3–6 crypts (16♂,15♀), **C** n = 3–15 GPMVs (15♂,14♀), **D** n = 5–13 CSCs (15♂,14♀) from mice or **E** cells and **F** their derived GPMVs from n = 3 independent biological replicates or **G** n = 17–31 FOV (2–3 PDOs per FOV) (mean ± SD, n value shown in each graph). Number of crypts, CSCs, cultured colonocytes, and their GPMVs, and FOVs analyzed are provided below each bar. Statistical significance was determined by one-way ANOVA and post hoc Tukey's test. Different letters indicate significant differences between WT and mutant APC groups (P < 0.05). **H** Effects of oncogenic APC on plasma membrane cholesterol organization. Cells co-expressing fluorescently-labeled **I** D4H and tH or **J** Lypd6 were used to for FLIM-FRET analyses. **K** Quantitative analysis of IL cholesterol. D4H-EGFP plasma membrane fluorescence

intensity was normalized to total D4H-EGFP fluorescence intensity. Cells were pre-treated with mevastatin (5 μM, 24 h), MβCD (10 mM, 30 min), or cholesterol (2 mM, 30 min) and subsequently incubated with control or Wnt3a-conditioned media (30 min). FRET efficiency was calculated from FLIM data averaged per FOV (mean ± SD, n = # FOVs provided below each bar, FOV containing 2–6 cells). For flow cytometry IL cholesterol experiments, error bars represent n = 3 independent biological replicates (mean ± SEM, cell number provided below each bar). Statistical significance determined by **H–K** two-way ANOVA and post hoc Tukey's. Different letters indicate significant differences between control and experimental groups (P < 0.05). **L** Quantitative analysis of Ld and Lo domain stability. The percentage of phase separated GPMVs was determined by GPMV$_{separated}$/GPMV$_{total}$ ratio (mean ± SEM, n = 12 FOVs, analyzed GPMV number shown in graph). Representative images and scale bars are provided for microscopy data. A sigmoidal four parameter logistic regression model was used to determine T$_{misc}$. Source data file provided.

## Oncogenic APC dysregulates plasma membrane "hot spots" involved in Wnt signaling

The consistent effects of oncogenic APC on plasma membrane free cholesterol homeostasis motivated us to interrogate the consequences of these perturbations with respect to lipid rafts. Lipid rafts are of particular interest since they are enriched in free cholesterol and serve as the signaling platforms of the Wnt pathway in *Drosophila* and mammals[21,67]. From a mechanistic standpoint, there is now compelling evidence indicating that free cholesterol selectively activates canonical Wnt signaling over non-canonical Wnt signaling[32]. This is noteworthy, since the binding of Wnt ligands to its receptors, Fzd and LRP5/6, has been shown to trigger the enrichment of activated receptors in cholesterol-enriched microdomains, reminiscent of lipid rafts. Therefore, we hypothesized that oncogenic APC altered the membrane environment of lipid raft domains. To examine the effect of oncogenic APC on plasma membrane order/rigidity in vivo, a key feature of raft domains, live colonic crypts were imaged using di-4-ANEPPDHQ (Di-4) spectral microscopy. When localized to membranes, the Di-4 emission spectrum is sensitive to changes in cholesterol, membrane potential

and local lipid packing, and generates a progressively red-shifted emission with increasing membrane fluidity (disorder)[68]. This property of Di-4 allows the construction of a generalized polarization (GP) image and a quantitative assessment of relative plasma membrane order. Live colonic crypts, their derived GPMVs and stem cells from AGC het and AfGC homo mice labeled with Di-4 exhibited a significant increase in plasma membrane rigidity when compared to WT APC (Fig. 5A–D). In addition, human CRC cell lines, their derived GPMVs and PDOs expressing oncogenic truncated APC displayed enhanced plasma membrane rigidity when compared to their WT APC counterparts (Fig. 5E–G). Interestingly, similar to free cholesterol data from mice colonic crypts and Lgr5+ stem cells as well as human CRC cultured colonocytes and PDOs (Figs. 2 and 3), oncogenic APC-induced a gene dose effect on plasma membrane order (double mutant allele Apc 580/580 > single allele Apc 580/+ > WT Apc +/+) (Fig. 5B, C). Overall, the single-cell membrane order data from both in vivo and in cellulo models were consistent with our in vivo live mouse colonic crypt observations (Fig. S7). In addition, the increase in plasma membrane rigidity observed in CRC cultured cells occurred in the absence of

stimulation with Wnt3a ligand, further indicating the significance of constitutive activation of Wnt signaling in this CRC model when compared to normal cells.

Cholesterol levels and organization in the plasma membrane are tightly regulated by various mechanisms, including de novo biosynthesis, uptake, endocytosis, and cholesterol-sensing[69]. In addition, plasma membrane cholesterol organization greatly influences lipid raft dynamics and functionality[70,71]. Thus, an increase in plasma membrane free cholesterol levels (Figs. 2 and 3) could lead to an increase of free cholesterol in raft membrane domains, thereby disrupting lipid packing, the localization of raft proteins, and raft-associated signaling. Since plasma membrane-associated raft domains serve as Wnt signaling "hot spots", we examined free cholesterol dynamics in raft membrane domains using the fluorescently-labeled cholesterol-specific sensor D4H (D434S mutant of the D4 domain; D4H-EGFP)[70] in combination with fluorescently-labeled truncated HRas (tH; a cholesterol-sensitive lipid raft marker)[72]. Specifically, the nanoscale organization of these probes was assessed using fluorescence lifetime imaging combined with fluorescence resonance energy transfer (FLIM-FRET) microscopy. When co-expressed with a suitable FRET pair, such as RFP or mCherry (acceptor), a reduction of EGFP (donor) fluorescence lifetime is indicative of tight donor–acceptor interactions and an increase in FRET efficiency. The FLIM-FRET method is highly favored over intensity-based FRET measurements because fluorescent lifetime is an intrinsic property of the fluorescent molecule and is generally insensitive to weak signal, excitation source, and variations in the donor–acceptor ratio[73]. To examine the effect of oncogenic APC on the lipid raft cholesterol pool, colonocytes were processed and FLIM images were subsequently generated. Notably, most of the fluorescence signal emitted by both reporters, i.e., D4H and tH, localized at the plasma membrane (Fig. S8A, S8B). FLIM images from Wnt3a-stimulated IMCE (Apc 850/+) and IMCE βcat (Apc 850/+ βcat −/+) cell lines expressing D4H-EGFP and tH-RFP exhibited an increase in the apparent FRET efficiency when compared to YAMC (Apc +/+) (Fig. 5H). Furthermore, since FRET occurs in a distance-dependent manner, our data indicate that the proximity (lower FRET calculated distance) between tHRas and cholesterol at the plasma membrane was increased by oncogenic APC (Fig. S9A) when compared to WT APC. As a positive control, the level of plasma membrane free cholesterol in cultured colonocytes was modulated by incubation with the cholesterol-lowering drug methyl-β-cyclodextrin (MβCD)[74]. As expected, treatment with MβCD led to a decrease in the FRET efficiency in all colonic cell lines. In addition, mevastatin, an HMG-CoA inhibitor was used as a positive control to reduce tH membrane anchorage, by blocking its farnesylation, thereby suppressing FRET efficiency[75]. As expected, both MβCD and mevastatin treatments reduced D4H-tH FRET efficiency. Collectively, these data indicate that the organization of free cholesterol in the plasma membrane, including cholesterol associated with ordered raft domains, is altered by oncogenic APC.

An increase in apparent FRET efficiency can also be indicative of enhanced molecular nanoclustering, a dominant feature of plasma membrane organization that modulates signaling[76]. Since cholesterol has been shown to form membrane clusters[77] and regulate signaling from the plasma membrane[32], including Wnt signaling, we used EGFP- and mCherry-tagged D4H to investigate the effect of oncogenic APC on cholesterol clustering by FLIM-FRET. This method provides a quantitative measure of inner leaflet lipid raft topology and stability[78]. FLIM images from Wnt-stimulated IMCE βcat (Apc 850/+ βcat −/+), IMCE (Apc 850/+), and YAMC (Apc +/+) cells exhibited increased D4H-associated FRET efficiency when compared to their unstimulated counterparts (Fig. 5I). In addition, IMCE and IMCE βcat cells exhibited a higher D4H FRET efficiency compared to YAMC cells in the presence of Wnt. Strikingly, the percentage of D4H FRET efficiency in unstimulated IMCE and IMCE βcat cells was higher than Wnt-stimulated YAMC cells.

As expected, MβCD and mevastatin disrupted free cholesterol clustering. To further validate the relevance of our mouse cultured colonocyte model, we examined the effects of mutant APC on cholesterol clustering by employing our human CRC cell model expressing truncated APC gene products of varying amino acid lengths. After transfection, cells were fixed and imaged as described previously[79]. All three APC truncations, which display their respective mutation within the MCR region of the *Apc* gene, induced an increase in cholesterol clustering (Fig. S10A), consistent with IMCE and IMCE βcat cells (Fig. 5I and Fig. S10B). Furthermore, our FLIM-FRET data indicate that the nanoscale topology of plasma membrane liquid ordered-lipid raft-like domains are increased by oncogenic APC (Fig. S9B).

As shown by the plasma membrane free cholesterol findings, the increase in plasma membrane rigidity observed in CRC cultured cells occurred in the absence of stimulation with Wnt3a ligand, further indicating the significance of constitutive activation of Wnt signaling in this CRC model when compared to normal cells. Since oncogenic APC altered cholesterol clustering, we also examined the effect of oncogenic APC on the clustering of additional Wnt-associated raft resident effectors, e.g., Ly6 family protein LY6/PLAUR domain-containing 6 (Lypd6). Lypd6 is a Wnt pathway positive feedback regulator, which by virtue of its glycerophosphatidylinositol (GPI) anchor, preferentially localizes to lipid rafts[20]. This Wnt pathway regulator ensures phosphorylation of LRP6 as well as activation of Wnt receptors within plasma membrane raft domains[20]. Notably, Lypd6 has been shown to form ~100 nm protein clusters[80]. Since Lypd6 nanoclustering may serve an important regulatory role in the phosphorylation of Wnt receptors and stabilization of active Wnt signaling clusters in lipid rafts, we investigated the effect of oncogenic APC on Lypd6 spatial organization in colonocytes expressing EGFP- and mCherry-tagged Lypd6 (Fig. S8C). FLIM images from IMCE βcat, IMCE and YAMC cells indicated that oncogenic APC increased Lypd6-associated FRET efficiency of untreated and Wnt3a-treated cells (Fig. 5J). Interestingly, the percentage of Lypd6 FRET efficiency in unstimulated IMCE and IMCE βcat cells was higher than Wnt-stimulated YAMC cells, consistent with the effects of oncogenic APC on cholesterol clustering (Fig. 5I). Similar to cholesterol clustering, MβCD and mevastatin treatment lowered Lypd6 clustering. Moreover, our FLIM-FRET data indicate that the distance between Lypd6 molecules at the plasma membrane was decreased (tighter interactions) by oncogenic APC (Fig. S9C).

There is now compelling evidence indicating that free cholesterol selectively activates canonical Wnt signaling over non-canonical Wnt signaling[32]. From a mechanistic standpoint, this is noteworthy since the binding of Wnt ligands to its receptors, i.e., Fzd and LRP5/6, has been shown to trigger the enrichment of activated receptors in cholesterol-enriched microdomains, reminiscent of lipid rafts[32]. Moreover, stimulation with Wnt ligands induces enrichment of cholesterol at the plasma membrane inner leaflet (IL), where it acts as a lipid modulator to regulate Wnt signaling transduction[81]. It is worth noting that other cellular processes could potentially modulate IL cholesterol. Nonetheless, the effect of oncogenic APC and Wnt3a stimulation on IL cholesterol, an important pool of cellular cholesterol, was assessed using imaging flow cytometry. Specifically, we correlated the changes in the fluorescence intensity ratio ($D4H_{IL}/D4H_{total}$) of membrane-bound and total D4H-EGFP (cytosolic + membrane-bound D4H-EGFP) to changes in IL plasma membrane cholesterol. The $D4H_{IL}/D4H_{total}$ ratio was similar in colonocytes expressing oncogenic APC relative to their WT APC counterpart under non-stimulated conditions. However, IMCE and IMCE βcat cells displayed a significant increase in the $D4H_{IL}/D4H_{total}$ ratio compared to YAMC cells in the presence of Wnt3a (Fig. 5K). Furthermore, depletion of plasma membrane cholesterol by exogenous treatment of colonocytes with MβCD or mevastatin resulted in a reduction of $D4H_{IL}/D4H_{total}$, as expected. In contrast, enrichment of free cholesterol at the plasma membrane by exogenous treatment with cholesterol complexed to MβCD

(MβCD:cholesterol) increased $D4H_{IL}/D4H_{total}$ in all stimulated cell lines. Interestingly, plasma membrane cholesterol enrichment of WT APC-expressing YAMC cells resulted in $D4H_{IL}/D4H_{total}$ levels similar to oncogenic APC-expressing IMCE cells. These observations are noteworthy, since changes in IL cholesterol concentration, specifically in the vicinity of Wnt receptors, is an essential step required for Wnt signaling initiation[32,81].

To investigate a potential thermodynamic coupling between oncogenic APC-driven increase in plasma membrane cholesterol and lipid raft dysregulation, we measured the stability of liquid disordered (Ld)-liquid ordered (Lo) immiscibiltity in GPMVs (Fig. 5L). Cell-derived GPMVs allow for the examination of phase separation in a plasma membrane system displaying biochemical and biophysical complexity[51]. At relatively low temperatures, GPMVs separated into coexisting Ld-Lo domains (Fig. 5L, top left image), which "melted" (became miscible) at relatively higher temperatures (Fig. 5L, bottom right image). The temperature at which 50% of GPMVs are phase separated is defined as the miscibility transition temperature or $T_{misc}$. Since the level of plasma membrane cholesterol has been shown to be directly related to the formation and abundance of the Lo phase fraction[82], determining the effect of oncogenic APC on the $T_{misc}$ could shed light on the stability of Wnt-associated Lo domains. Thus, to quantify $T_{misc}$, GPMVs derived from cells expressing WT APC (YAMC (Apc +/+)) or oncogenic truncated APC (IMCE (Apc 850/+), IMCE βcat (Apc 850/+ βcat −/+), DLD1 (Apc 1417/-) and HT29 (Apc 853/1555)) were stained with Fast DiI, a disordered lipid phase marker, and imaged by confocal microscopy to measure the temperature dependence on the abundance of phase separated (immiscible) and miscible GPMVs. Isogenic IMCE and IMCE βcat cell lines, which we have demonstrated display an increase in plasma membrane cholesterol and rigidity, exhibited an increased $T_{misc}$, 27.6 °C and 28.4 °C, respectively, when compared to their WT APC counterpart (YAMC; $T_{misc}$ = 20.0 °C) (Fig. 5L). Moreover, two key CRC-derived cell lines expressing oncogenic truncated APC, i.e., DLD1 ($T_{misc}$ = 28.3 °C) and HT29 ($T_{misc}$ = 25.3 °C), also exhibited an increase in $T_{misc}$. These data are consistent with the notion that oncogenic truncated APC, by increasing plasma membrane cholesterol and rigidity, imparts enhanced stability to Lo domains in the plasma membrane of CRC cells. This is noteworthy, since the organization of plasma membrane Lo-Ld domains is driven by the partial interactions between specific proteolipid components, thus giving rise to functionally active lateral plasma membrane signaling domains. Moreover, the physicochemical basis for this organization is highly cholesterol-dependent and can have long-range interdomain membrane organization effects.

### Oncogenic APC-driven plasma membrane perturbations alter interactions between proteolipid components of Wnt condensates

The conspicuous effects of oncogenic APC on plasma membrane homeostasis in colonocytes motivated us to interrogate the consequences of cholesterol-dependent biophysical and biochemical perturbations with respect to the spatially compartmentalized Wnt signaling machinery, also known as biomolecular condensates, at the plasma membrane. These proteolipid condensates form through multivalent interactions between biomacromolecules to elicit highly organized distinct signaling compartments[83–85]. From a mechanistic context, there is now compelling evidence indicating that free cholesterol promotes recruitment and interactions between Wnt pathway receptors, e.g., LRP5/6 and Fzd, membrane lipids, e.g., phosphatidylinositol-4,5-bisphosphate ($PI(4,5)P_2$) and key cytosolic effectors, e.g., Disheveled (Dvl) and Axin, in the plasma membrane thereby promoting signaling activation[32]. The binding of Wnt ligand to its receptors, triggers enrichment of free cholesterol in the local proximity of activated receptors by bringing Fzd and LRP5/6 together into a cholesterol-enriched microenvironment. With this in mind, we sought to elucidate the effect of oncogenic APC on the interactions between plasma membrane free cholesterol and Wnt pathway receptors/effectors. To examine free cholesterol-Wnt receptor interactions, colonocytes expressing EGFP-tagged LRP6 or Fzd7 (donor) (Fig. S8D, S8E) and D4H-mCherry (acceptor) were imaged by FLIM-FRET. FLIM images from Wnt3a-stimulated YAMC, IMCE, and IMCE βcat cell lines exhibited increased LRP6- and Fzd7-D4H FRET efficiency when compared to their unstimulated counterparts (Fig. 6A, B). Cells expressing oncogenic APC exhibited a higher LRP6- and Fzd7-D4H FRET efficiency in comparison to WT APC-expressing cells in the presence of Wnt3a. Interestingly, the changes in FRET efficiency of stimulated vs unstimulated cells were greater for Fzd7 when compared to LRP6 (Fig. 6A, B). In addition, oncogenic truncated APC decreased the distance between cholesterol and Wnt receptors at the plasma membrane (Fig. S9D, S9E). As a control, MβCD and mevastatin significantly decreased LRP6- and Fzd7-D4H FRET efficiency in Wnt3a-treated colonocytes and normalized the phenotype induced by oncogenic APC down to WT APC levels. Similar to our cholesterol clustering experiments, we examined the effects of truncated APC on LRP6-cholesterol interactions utilizing our human CRC cell model expressing truncated APC gene products of varying amino acid lengths. We found that all three APC truncations increased the interaction between LRP6 and cholesterol (Fig. S10C), consistent with IMCE and IMCE βcat cells (Fig. 6B and Fig. S10D). Relative to our plasma membrane free cholesterol and rigidity experiments, the enhancement in LRP6-cholesterol interactions in CRC cultured cells occurred without Wnt3a stimulation. This is yet another key piece of evidence indicating the relevance of constitutive activation of Wnt signaling in this CRC model when compared to normal cells.

The effect of mutant APC on another key signaling lipid, $PI(4,5)P_2$, which is predominantly localized in the plasma membrane IL[86] and has been shown to modulate Wnt receptor organization and activation[87] was also examined. To interrogate $PI(4,5)P_2$-Wnt receptor interactions, we used the pleckstrin homology (PH) domain of phospholipase C δ1 (PLC-δ1), a $PI(4,5)P_2$ sensor that displays a high selectivity and binding efficiency towards $PI(4,5)P_2$[88]. FLIM images of stimulated YAMC, IMCE, and IMCE βcat cells expressing EGFP-tagged LRP6 or Fzd7 and PLC-δ1-mCherry (Fig. S8F), exhibited increased FRET efficiency when compared to their unstimulated counterparts (Fig. 6C, D). In addition, IMCE and IMCE βcat cells exhibited a higher LRP6- and Fzd7-PLC-δ1 FRET efficiency in comparison to YAMC cells in the presence of Wnt3a. Interestingly, LRP6 was particularly sensitive to the effects of oncogenic APC in unstimulated cells as indicated by an increase in LRP6-PLC-δ1 FRET efficiency (Fig. 6D), while Fzd7 remained unaffected (Fig. 6C). As a control, phenylarsine oxide (PAO), a specific inhibitor of phosphatidylinositol 4-kinase type IIα (PI4KIIα) that blocks the synthesis of $PI(4,5)P_2$ from phosphatidylinositol (PI)[89], significantly decreased LRP6- and Fzd7-PLC-δ1 FRET efficiency in Wnt3a-stimulated cells (Fig. 6C, D). Consistent with these results, the proximity between $PI(4,5)P_2$ and Wnt receptors was increased by oncogenic APC (Fig. S9F, S9G). To further examine the effect of oncogenic APC on $PI(4,5)P_2$ dynamics, we measured the $PLC-\delta1_{IL}/PLC-\delta1_{total}$ ratio in cultured colonocytes. As expected, Wnt3a stimulation induced an increase in the $PLC-\delta1_{IL}/PLC-\delta1_{total}$ ratio[87]. Similar to plasma membrane IL-free cholesterol, the $PLC-\delta1_{IL}/PLC-\delta1_{total}$ ratio was significantly increased in IMCE and IMCE βcat relative to YAMC cells under Wnt3a-stimulated conditions suggesting an increase in the levels of plasma membrane IL $PI(4,5)P_2$ (Fig. 6E). In contrast, unstimulated cells remained unaltered. Moreover, treatment of cultured colonocytes with PAO decreased the $PLC-\delta1_{IL}/PLC-\delta1_{total}$ ratio in Wnt3a-stimulated colonocytes (Fig. 6E). To confirm the role of these important lipids, i.e., cholesterol and $PI(4,5)P_2$, in downstream Wnt signaling activation, we probed the effect of oncogenic APC in regard to their interactions with Dvl, a cytosolic effector necessary for Wnt signal transduction and

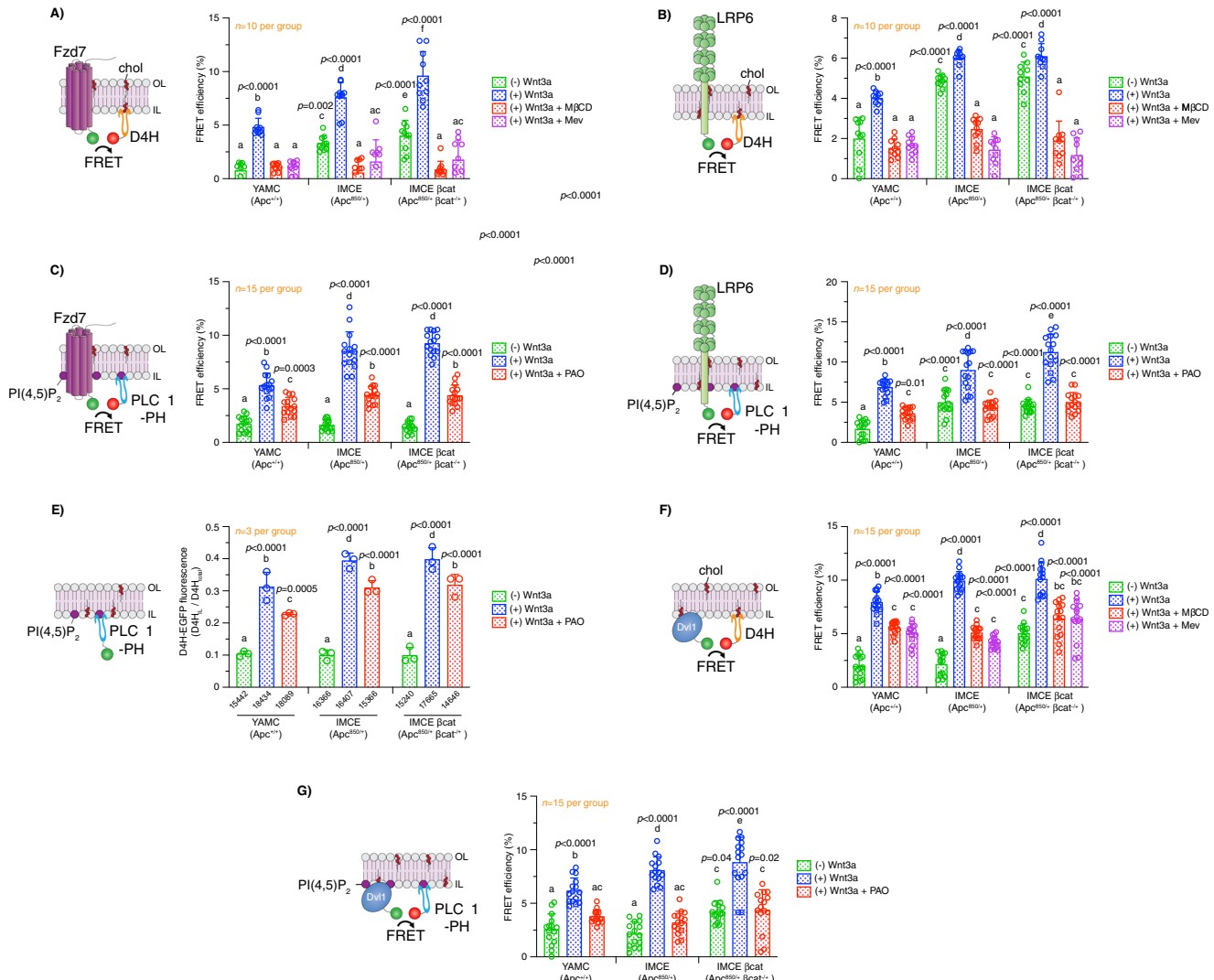

**Fig. 6 | Oncogenic APC alters interactions between Wnt receptors and their effectors.** To examine the effect of oncogenic APC on interactions between Wnt receptors and key lipids, cells co-expressing EGFP-tagged **A**, **C** Fzd7 or **B**, **D** LRP6 and mCherry-tagged **A**, **B** D4H or **C**, **D** pleckstrin homology (PH) domain of phospholipase C δ1 (PLC-δ1, PI(4,5)P₂ sensor) were used to perform FLIM FRET. **E** To examine the effect of oncogenic APC on PI(4,5)P₂ plasma membrane levels, cells expressing EGFP-tagged PLC-δ1-PH were used to measure membrane-associated EGFP fluorescence intensity using flow cytometry. EGFP plasma membrane fluorescence intensity was normalized to total EGFP fluorescence intensity. To examine the effect of oncogenic APC on the interactions between Dvl1 and key lipids, cell co-expressing EGFP-tagged **F**, **G** Dvl1 and mCherry-tagged **F** D4H or **G** PLC-δ1 were used to perform FLIM-FRET. YAMC, IMCE, and IMCE βcat cells were pre-treated with mevastatin (5 μM, 24 h), MβCD (10 mM, 30 min), or phenylarsine oxide (PAO) (20 μM, 30 min) and washed, as indicated. Subsequently, cells were incubated with Wnt3a-conditioned media or control media without Wnt3a for 30 min, washed, fixed, and imaged. For FLIM-FRET experiments, the apparent FRET efficiency was calculated from FLIM data averaged per FOV (mean ± SD, $n = 10-15$ FOVs containing 3–8 cells were examined per condition, exact $n$ value is shown in each graph). For flow cytometry IL PI(4,5)P₂ experiments, cells were imaged to calculate EGFP fluorescence intensity (mean ± SEM, from $n = 3$ independent biological replicates, total number of cells analyzed is provided below each bar). For all experiments, statistical significance was determined by two-way ANOVA and post Tukey's multiple comparison test. Different letters indicate significant differences between WT APC (control) and mutant APC/treatment groups (experimental) ($P < 0.05$). Source data are provided as a Source data file.

activation of βcat. This pleiotropic Wnt effector has been shown to directly interact with Fzd, free cholesterol, and anionic lipids, e.g., phosphatidylserine (PS) and PI(4,5)P₂, in the plasma membrane[32,90,91]. Thus, YAMC, IMCE, and IMCE βcat cells expressing EGFP-Dvl1 (Fig. S8G) and mCherry-tagged D4H (cholesterol sensor) or PLC-δ1 (PI(4,5)P₂ sensor) were examined by FLIM-FRET. FLIM images from Wnt3a-stimulated colonocytes exhibited an increase in Dvl1-D4H and Dvl1-PLC-δ1 FRET efficiency when compared to their unstimulated counterparts (Fig. 6F, G). In addition, cells expressing oncogenic APC exhibited a higher Dvl1-D4H and Dvl1-PLC-δ1 FRET efficiency in comparison to WT cells in the presence of Wnt3a. As expected, low Dvl1-D4H and Dvl1-PLC-δ1 FRET efficiency was observed in almost all

unstimulated cells, since most of the Dvl1 population remains in the cytosol and does not interact with the plasma membrane in the absence of Wnt3a[92]. It is noteworthy that IMCE βcat, which displays the highest degree of Wnt dysregulation, exhibited a significant increase in Dvl1-D4H and Dvl1-PLC-δ1 FRET efficiency in the absence of Wnt3a when compared to unstimulated IMCE and YAMC (Fig. 6F, G). As a control, both MβCD and mevastatin- or PAO-treated Wnt3a-stimulated colonocytes exhibited a decreased Dvl1-D4H and Dvl1-PLC-δ1 FRET efficiency, respectively. Altogether, these data indicate that plasma membrane perturbations induced by oncogenic APC modulate interactions between essential receptors and effectors found in Wnt signaling-associated proteolipid condensates.

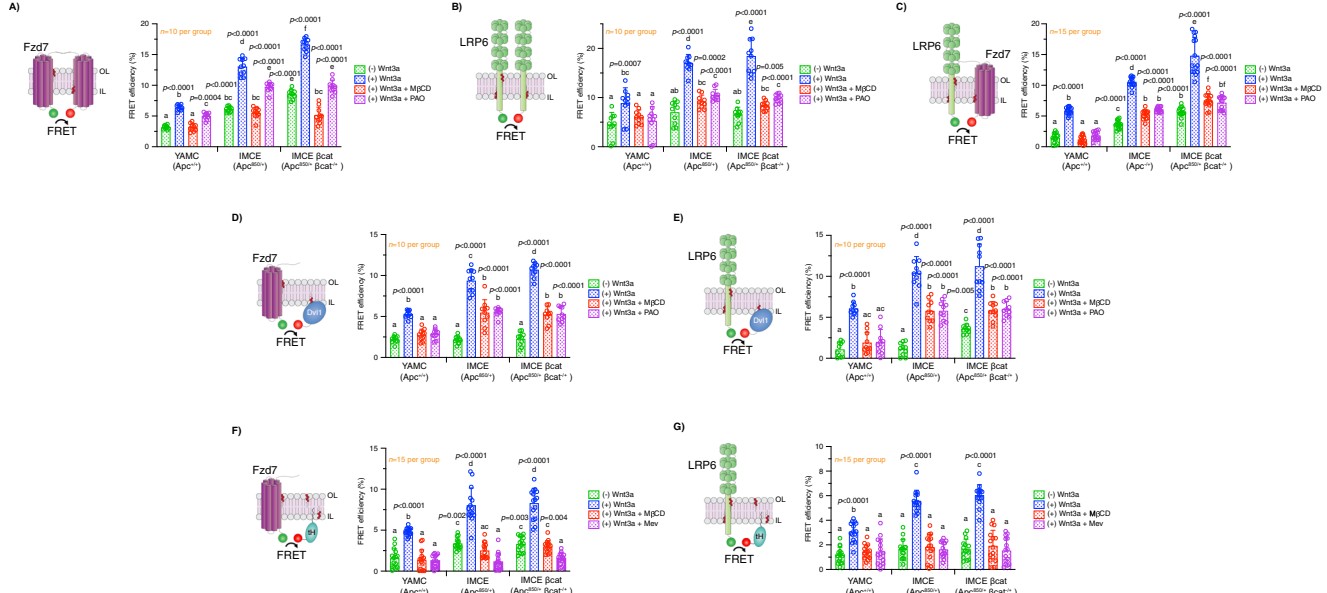

**Fig. 7 | Oncogenic APC enhances macromolecular interactions within Wnt receptor nanoscale signaling platforms.** For in vitro FLIM-FRET experiments, cells co-expressing EGFP- and mCherry-tagged **A** Fzd7 or **B** LRP6 or **C** EGFP-tagged LRP6 and mCherry-tagged Fzd7 were used to perform homo- and hetero-clustering FLIM-FRET analyses, respectively. To examine the effect of oncogenic APC on the interactions between Dvl1 and Wnt receptors, cells co-expressing EGFP-tagged **D** Fzd7 or **E** LRP6 and mCherry-tagged Dvl1 were used to perform FLIM-FRET. To examine the effect of oncogenic APC on plasma membrane Wnt receptor localization, cells co-expressing EGFP-tagged **F** Fzd7 or **G** LRP6 and tH-RFP were used to perform FLIM-FRET analyses. For FLIM-FRET experiments, YAMC, IMCE, and IMCE βcat cells were pre-treated with mevastatin (5 μM, 24 h), MβCD (10 mM, 30 min), or phenylarsine oxide (PAO) (20 μM, 30 min) and washed, as indicated. Subsequently, cells were incubated with Wnt3a-conditioned media or control media without Wnt3a for 30 min, washed, fixed, and imaged. The apparent FRET efficiency was calculated from FLIM data averaged per FOV (mean ± SD, from n = 10–15 FOVs containing 3–5 cells each were examined per condition, exact n value is shown in each graph). For all experiments, statistical significance was determined by two-way ANOVA and post Tukey's multiple comparison test. Different letters indicate significant differences between WT APC (control) and mutant APC/treatment groups (experimental) (P < 0.05). Source data are provided as a Source data file.

## Oncogenic APC remodels the structure and organization of Wnt receptor nanoassemblies

It is well known that LRP5/6 and Fzd require lipid raft localization and nanoclustering (100–200 nm) within Wnt macromolecular condensates for efficient signaling and the stabilization of βcat[20,21,42,76]. Based on the notion that oncogenic APC leads to overactivation of Wnt signaling, a hallmark of CRC[14], we hypothesized that oncogenic APC-induced plasma membrane perturbations disrupt the structure and organization of Wnt receptor nanoassemblies. Thus, we assessed the degree of Wnt receptor homo- and hetero-nanoclustering in cultured colonocytes expressing fluorescently-labeled LRP6 and Fzd7 using FLIM-FRET. In Wnt receptor homo-clustering experiments, Wnt3a-stimulated colonocytes expressing oncogenic APC exhibited increased Fzd7-Fzd7 and LRP6-LRP6 FRET efficiency compared to WT APC (Fig. 7A, B). This is noteworthy, since previously published data suggest that Wnt signal transduction requires an amplification step, which depends on LRP6 homodimerization[10]. Furthermore, oncogenic APC enhanced Fzd7 homo-clustering in unstimulated colonocytes while LRP6 homo-clustering remained unaffected. In hetero-clustering experiments, unstimulated and Wnt3a-stimulated colonocytes expressing oncogenic APC displayed increased LRP6-Fzd7 FRET efficiency when compared to WT APC (Fig. 7C). It is noteworthy, that the magnitude of LRP6-Fzd7 FRET efficiency of unstimulated colonocytes expressing oncogenic APC was similar in comparison to Wnt3a-stimulated cells expressing WT APC. To examine the role of key Wnt signaling lipid effectors in Wnt receptor organization, MβCD and PAO were used to disrupt the levels of free cholesterol and PI(4,5)P₂, respectively. In these experiments, both MβCD- and PAO-treated Wnt3a-stimulated colonocytes decreased LRP6 and Fzd7-associated homo- and hetero-clustering (Fig. 7A–C). In addition, we examined the effects of various truncated APC proteins on LRP6-Fzd7 hetero-clustering interactions utilizing our human CRC cell line models. We found

that all three APC truncations enhanced LRP6-Fzd7 nanoclustering (Fig. S10E), consistent with IMCE and IMCE βcat cells (Fig. 7C and Fig. S10F). Moreover, our data indicate that the proximity (lower FRET calculated distance) between Fzd7 and LRP6 at the plasma membrane was increased by oncogenic APC (Fig. S9H) when compared to WT APC.

To further investigate the effect of oncogenic APC on Wnt-associated proteolipid condensate structure and organization, we examined the interactions between Wnt receptors and Dvl. As expected, FLIM images from Wnt3a-stimulated cultured colonocytes expressing EGFP-tagged LRP6 or Fzd7 and mCherry-Dvl1 exhibited an increased FRET efficiency when compared to their unstimulated counterparts (Fig. 7D, E). In addition, LRP6- and Fzd7-Dvl1 FRET efficiency associated with IMCE and IMCE βcat cells was higher in comparison to YAMC control in the presence of Wnt3a. In contrast, unstimulated colonocytes displayed low LRP6- and Fzd7-Dvl1 FRET efficiency, consistent with Dvl1 localizing exclusively to the cytosolic space in the absence of receptor activation. Strikingly, LRP6-Dvl1 interactions showed a slight increase in unstimulated IMCE βcat cells (highest Wnt signaling dysregulation) when compared to all other unstimulated conditions tested (Fig. 7D, E). Consistent with these results, our data indicate that the proximity (lower FRET calculated distance) between Dvl1 and Wnt receptors at the plasma membrane was increased by oncogenic APC (Fig. S9I, S9J) when compared to WT APC. In these experiments, both MβCD- and PAO-treated Wnt3a-stimulated colonocytes decreased Dvl1 interactions with LRP6 and Fzd7 (Fig. 7D, E).

We next investigated whether the increase in Wnt receptor clustering induced by oncogenic APC was, in part, due to the enhanced recruitment of receptor molecules into Wnt signaling condensates ("signaling hot spots"). For this purpose, the interactions between Wnt receptors and tH (lipid raft marker) were assessed using FLIM-FRET.

FLIM images of Wnt3a-stimulated colonocytes exhibited increased Fzd7-tH and LRP6-tH FRET efficiency compared to unstimulated colonocytes (Fig. 7F, G). Furthermore, Fzd7- and LRP6-tH FRET efficiency associated with oncogenic APC-expressing cells was higher in comparison to WT APC control in the presence of Wnt3a. Notably, oncogenic APC induced an increase in tH-Fzd7 FRET efficiency in unstimulated colonocytes compared to WT APC, while LRP6 remained unaffected. In these experiments, both MβCD- and mevastatin-treated Wnt3a-stimulated colonocytes decreased Fzd7-tH and LRP6-tH FRET efficiency (Fig. 7F, G).

**Oncogenic APC induces changes in plasma membrane hierarchical organization of Wnt proteolipid condensates in vivo**

In recent years, numerous studies have revealed that multiple proteins and lipids which localize to the plasma membrane, compartmentalize into defined regions known as nanoclusters[93–95]. Furthermore, nanocluster formation has been described to be an essential step involved in cellular signaling from the plasma membrane, thus making nanoclustering a central feature of plasma membrane organization[76]. Nanoclustering is also an important process that influences the formation of macromolecular condensates, likely via enhancing membrane phase separation and nanodomain stability. This is noteworthy, since we have observed that oncogenic APC modulates plasma membrane biophysical and biochemical properties, e.g., increases plasma membrane free cholesterol and rigidity. In addition, we have correlated these changes to the dysregulation of Wnt receptor clustering and their interactions with key lipid and protein effectors of the Wnt pathway. However, since plasma membrane organization is influenced by many factors including lipid saturation[57,96], cholesterol composition[76,81] and cytoskeletal interactions[97], its modulation may not be entirely correlated to protein clustering in the plasma membrane. Therefore, to confirm the effect of oncogenic truncated APC-induced plasma membrane dysregulation on Wnt receptor clustering, we used super-resolution microscopy to examine the structure and organization of key components of Wnt biological condensates (Fig. 8A). Initially, we measured the size and distribution of LRP6-Fzd7 nanoassemblies in PDOs using stochastic optical reconstruction microscopy (STORM) and Voronoi analysis (Fig. 8B and Fig. S11A). This fluorescence-based microscopy technique routinely achieves a spatial resolution of 20–30 nm[98,99], suitable for resolving Wnt nanoclusters (100–200 nm). PDOs were fixed as described previously[79] and fluorescently-labeled primary antibodies against Wnt receptors were employed to quantify the area as well as the size frequency distribution of LRP6-Fzd7 nanoclusters. The specificity of these antibodies was validated in preliminary experiments (Fig. S11B, S11C). Expression of oncogenic truncated APC in Crispr APC and CRC-PDOs significantly increased Wnt receptor cluster area compared to normal PDOs (Fig. 8C, E). In order to further elucidate the effect of oncogenic truncated APC on Wnt receptor plasma membrane organization, we assessed the frequency distribution of Wnt receptor clusters of varying size. Our examination revealed that oncogenic truncated APC induced an increase in the frequency of relatively larger clusters (Fig. 8D, F). We also examined the changes in cluster size and frequency distribution of Wnt nanoclusters in mouse single colonocytes derived from crypts isolated from AfGC (APC 580) homo and GC (WT APC) mice using a set of verified LRP6 and Fzd7 antibodies (Fig. S11D–S11F). Expression of oncogenic APC in mouse colonocytes significantly increased Wnt receptor cluster area as well as in the frequency of relatively larger clusters when compared to WT APC (Fig. S11G–S11J), similar to the results found for the PDOs (Fig. 8C–F). Notably, as observed in the case of LRP6-Fzd7 cluster size as well as plasma membrane free cholesterol and rigidity, the magnitude of dysregulation associated with Wnt receptor cluster size and the appearance of larger Wnt receptor clusters in mouse crypt-derived colonocytes increased over time following oncogenic APC expression.

To further investigate the details surrounding the disruption of Wnt proteolipid condensates by oncogenic truncated APC, we examined the global distribution and signaling status of Wnt receptors. Wnt signaling activation requires nanoclustering, which in turn depends on the arrangement of Wnt receptors into condensed signaling "hotspots"[76,100]. We hypothesized that canonical Wnt signaling activation will lead to an increase in the total number of Wnt receptor molecules present within clusters, i.e., increased density, as well as the total number of clusters present in each cell and their respective density. Furthermore, we hypothesized that oncogenic truncated APC will promote biochemical properties of Wnt condensates. To quantify changes in single molecule and cluster organization, we employed single-color single-molecule localization microscopy data, i.e., STORM and a cluster detection method, density-based spatial clustering of applications with noise (DBSCAN), along with a customized co-localization analysis. Consistent with our FLIM-FRET (clustering) and STORM (cluster size) results, oncogenic truncated APC increased the total number of Fzd7 and LRP6 single molecules found within a cluster (Fig. 8G, H) as well as the absolute density of Fzd7 and LRP6 molecules per individual cluster area (Fig. 8I, J). In addition, oncogenic truncated APC promoted the recruitment of Fzd7 and LRP6 molecules into clustered domains, as we were able to detect an increase in the percentage of clustered vs non clustered localized receptor single molecules (Fig. 8K, L). Finally, oncogenic truncated APC promoted the formation of Fzd7 and LRP6 clusters (Fig. 8M, N) as well as Fzd7-LRP6 cellular cluster density (Fig. 8O, P).

Finally, to confirm that the structure and organization of key members of the Wnt condensate signaling machinery are modulated by oncogenic truncated APC, we measured the size and distribution of Dvl1, an essential Wnt signaling-related cytosolic protein, in PDOs using stochastic optical reconstruction microscopy (STORM) and Voronoi analysis. PDOs were fixed and fluorescently-labeled using a primary antibody against Dvl1 to quantify the area as well as the size frequency distribution of Dvl1 nanoassemblies (Fig. S12A). Expression of oncogenic truncated APC in Crispr APC and CRC-PDOs significantly increased Dvl1 cluster area compared to normal PDOs (Fig. S12B). In addition, our examination revealed that oncogenic truncated APC induced an increase in the frequency of relatively larger Dvl1 clusters (Fig. S12C). Notably, consistent with the STORM experiments involving Fzd7-LRP6 nanoassemblies, oncogenic truncated APC increased the total number of Dvl1 single molecules found within a cluster (Fig. S12D), the absolute density of Dvl1 molecules per individual cluster area (Fig. S12E), promoted the recruitment of Dvl1 molecules into clustered domains (Fig. S12F), increased the number of Dvl1 clusters (Fig. S12G) and increased the density of Dvl1 cellular clusters (Fig. S12H). Overall, these data confirm that oncogenic truncated APC, due to its documented reduced scaffolding activity, dysregulated LRP6-Fzd7-Dvl1 condensate dynamics, in part, by increasing the incorporation of cytosolic Dvl1, translocation of LRP6, Fzd7, and Dvl1 molecules into clusters, and promoting the formation and stabilization of relatively large Wnt nanoassemblies.

**Free cholesterol is sufficient to reshape the organization of Wnt receptors and dysregulate downstream Wnt signaling activation**

Since oncogenic APC mediates the loss of plasma membrane free cholesterol homeostasis, thereby altering the plasma membrane localization of key Wnt members and increasing lipid raft-dependent interactions between Wnt receptor and their effectors, we next investigated the role of cholesterol in oncogenic APC-induced plasma membrane dysregulation and aberrant Wnt signaling in vivo. For this purpose, we employed the *Drosophila* intestinal (midgut/hindgut) model (Fig. 9A). The fly midgut and hindgut display functional and morphological similarities to the mammalian small and large intestine, respectively[101]. Moreover, similar to the mammalian intestine, the *Drosophila* midgut and hindgut are comprised of a monolayer

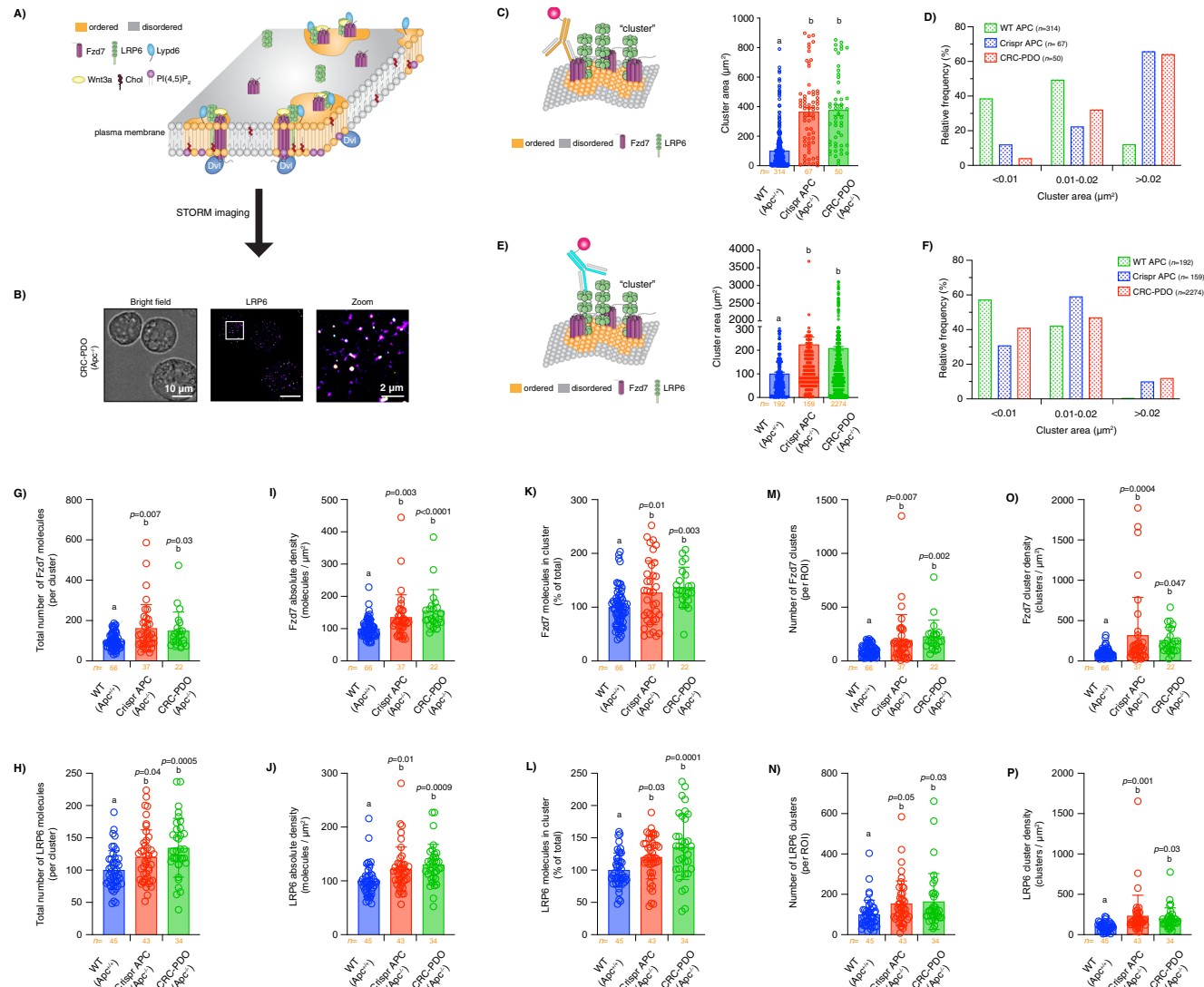

**Fig. 8 | Oncogenic APC alters the structure and organization of key protein constituents of the Wnt condensate signaling machinery.** For in vivo STORM imaging experiments, isolated single colonocytes from PDOs were fixed and labeled with primary monoclonal rat Fzd7 or mouse LRP6 antibody fluorescently labeled with Alexa Fluor 647. **A** Model of the formation of Wnt proteolipid condensates in ordered plasma membrane nanodomains examined via STORM imaging. **B** Representative bright field and Voronoi images of isolated single colonocytes from CRC-PDOs labeled with primary anti-LRP6-AF647. Quantitative analysis of Fzd7 and LRP6 **C**, **E** cluster area, **D**, **F** cluster area relative frequency, **G**, **H** total number of receptor molecules inside clusters, **I**, **J** receptor molecule absolute density, **K**, **L** percentage of receptor molecules forming part of clustered regions, **M**, **N** total number of receptor clusters, and **O**, **P** cellular receptor cluster density in isolated single colonocytes from PDOs, respectively. Cluster area was calculated from STORM data averaged per region of interest (ROI) and the respective relative frequency was calculated from individual cluster distribution data (mean ± SD, from $n = 50$–$2274$ ROIs, exact $n$ value is shown below each bar). Data associated with the number of single receptor molecules, receptor clusters, and their density was calculated from raw fluorescence intensity images converted to text (.txt) $x$–$y$ coordinate files using Clus-Doc (mean ± SD, from $n = 22$–$66$ ROIs, exact $n$ value is shown below each bar). Different letters indicate significant differences between WT APC (control) and mutant APC groups (experimental) ($P < 0.05$). Source data are provided as a Source data file.

epithelium that is replenished regularly by ISCs, which give rise to all intestinal cell types (Fig. 9A)[102]. The presence of ISCs within the intestine epithelium of this versatile genetic model organism allows for the use of genetic tools to assay Wnt pathway events associated with LRP5/6 and Fzd nanoclustering in vivo. Thus, the *Drosophila* adult gut is a powerful model to study signaling mechanisms regulating stem cell maintenance, dysfunction, and tumorigenesis, including aberrant Wnt signaling[103,104]. Initially, we established our ability to regulate the levels of free cholesterol in the *Drosophila* intestine. The level of free cholesterol in the fly gut was modulated exogenously by feeding flies diets with varying cholesterol concentrations. Since *Drosophila* is a sterol auxotroph, tissue cholesterol levels can be rigorously controlled via the diet[105,106]. Cholesterol-enriched diets of varying composition increased free cholesterol (as well as esterified cholesterol) in the

intestinal tissue in a dose-dependent manner (Fig. 9B, C). Notably, cholesterol feeding increased filipin III staining throughout the *Drosophila* midgut epithelium (Fig. 9D). Using these dietary conditions, we performed FLIM-FRET on *Drosophila* intestinal stem/progenitor cells from flies expressing EGFP- or mCherry-labeled humanized LRP6 (hLRP6) or Fzd7 (hFzd7) under the control of the upstream activating sequence (UAS) and temperature sensitive stem cell driver escargot(esg)-Gal4^TS to examine the effect of increased free cholesterol on Wnt receptor nanoclustering. Dietary cholesterol increased hLRP6, hFzd7, and hLRP6-hFzd7 FRET efficiency in *Drosophila* intestinal stem/progenitor cells in a dose-dependent manner (Fig. 9E–G). Consistent with in cellulo and in vivo experiments (Figs. 7 and 8, Figs. S10 and S11), the level of free cholesterol-induced dysregulation was correlated with the degree of Wnt receptor homo- and hetero-clustering.

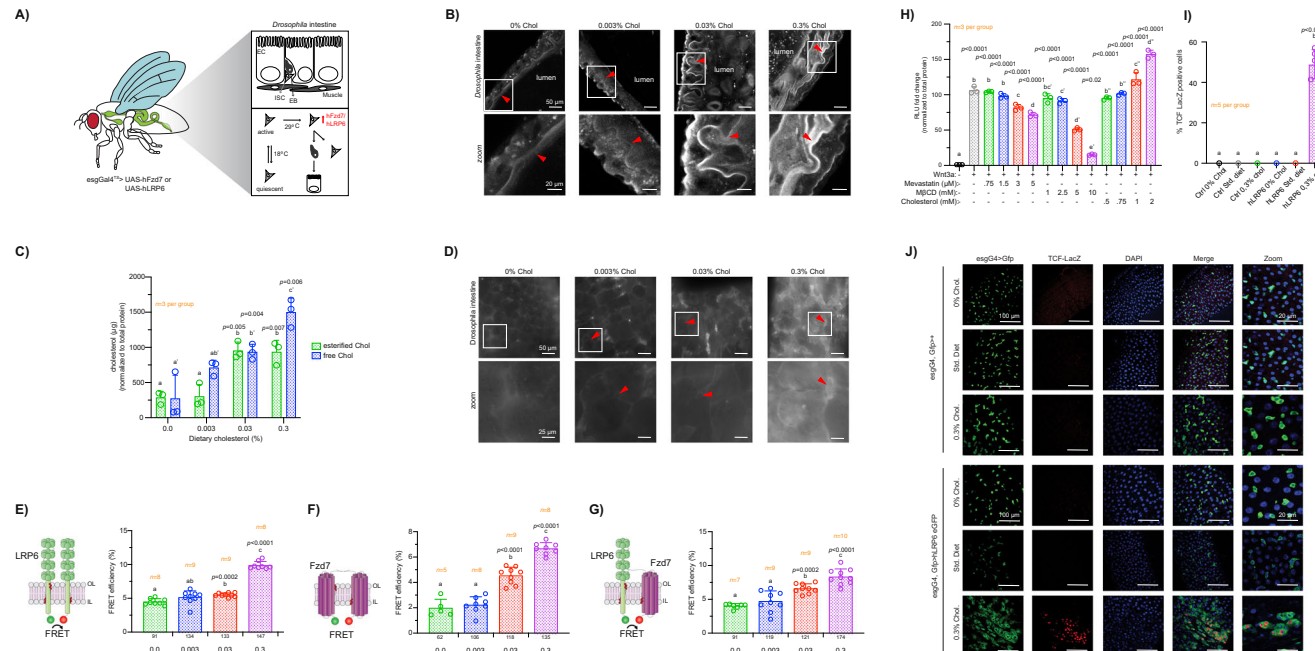

**Fig. 9 | In vivo effect of cholesterol modulation on Wnt receptor organization and βcat activation. A** *Drosophila* midgut-hindgut intestinal tissue model. ISCs/progenitor cells express humanized hFzd7 or hLRP6 under control of the UAS and esg-Gal4ᵀˢ. **B** Filipin III-stained midgut from *Drosophila* fed various cholesterol diets (red arrow, intestinal lumen). **C** Quantification of total cholesterol from *Drosophila* midgut. Cholesterol was calculated from luciferase luminescence data using the Amplex™ Red cholesterol assay and normalized to total protein (mean ± SD, from *n* = 3 independent biological replicates). **D** Filipin III fluorescence distribution of *Drosophila* intestinal epithelium (red arrow, ISCs). Effects of cholesterol on Wnt receptor organization. Flies co-expressing EGFP- and mCherry-tagged **E** LRP6 or **F** Fzd7 or **G** EGFP-LRP6 and mCherry-Fzd7 were used to perform FLIM-FRET in flies fed various cholesterol diets. FRET efficiency was calculated from FLIM data (mean ± SD, from *n* = 5–10 guts, ROIs analyzed provided below each bar, *n* value is shown in each graph). **H** Quantitative analysis of free cholesterol-induced βcat activation. 3T3 LL cells were pre-treated with mevastatin (5 µM, 24 h), MβCD (10 mM, 30 min), and MβCD-cholesterol (cholesterol) (10 mM, 30 min), and incubated with control or Wnt3a-conditioned media for 24 h. Luciferase luminescence was measured using a Luciferase Assay System kit. Luciferase luminescence fold change was normalized to total protein (mean ± SD, from *n* = 3 independent biological replicates). **I** Quantitative analysis of Wnt signaling activation. The percentage of TCF-LacZ+ cells is shown. Error bars represent *n* = 5 independent biological replicates (mean ± SD, -100 cells analyzed per group). **J** Qualitative analysis of cholesterol-induced Wnt activation. TCF-LacZ+ cells activity in *Drosophila* posterior midguts from control (*w¹¹¹⁸*; esgGal4ᵀˢ, UAS-GFP; TCF-LacZ) and hLRP6-expressing ISCs (*w¹¹¹⁸*; esgGal4ᵀˢ, UAS-GFP, UAS-hLRP6; TCF-LacZ). Flies were feed a cholesterol free, standard (Std. Diet) or high cholesterol diet for 5 days. ISCs, GFP; nuclei, DAPI. Statistical significance determined by **C** two-way ANOVA or **E**–**I** one-way ANOVA and post hoc Tukey's test. Different letters indicate significant differences between treatment groups (*P* < 0.05). Representative images and scale bars are provided for microscopy data. Enterocytes, ECs; visceral muscle cells, muscle; intestinal stem cells, ISCs; enteroblasts, EB. Source data file provided.

Our dietary observations indicate that loss of free cholesterol homeostasis alone is sufficient to induce increased tight interactions between LRP6 and Fzd7 and disrupt Wnt receptor organization. Therefore, we further elucidated the significance of free cholesterol-induced Wnt receptor disruption and the role of free cholesterol in aberrant Wnt signaling on a WT APC background. For this purpose, we employed a Wnt pathway reporter cell line (3T3 LL) expressing firefly luciferase under the control of the TCF/LEF promoter system. The TCF/LEF promoter system recruits βcat and allows transcription of βcat-related genes enabling the quantification of βcat activity and Wnt signaling activation. To quantify βcat activity, 3T3 LL (Apc +/+) cells were incubated with varying concentrations of free cholesterol and the resulting changes in luciferase expression (Wnt signaling activation) were determined using a quantitative luciferase activity assay. Free cholesterol dysregulation induced increased luciferase activity in Wnt3a-stimulated 3T3 cells in a dose-dependent manner compared to control (no exogenous free cholesterol) (Fig. 9H). In contrast, two distinct cholesterol-lowering drugs, i.e., mevastatin and MβCD, suppressed luciferase expression (Fig. 9H). In addition, reduction of plasma membrane free cholesterol with mevastatin inhibited βcat activity in a mutant APC background (IMCE) (Fig. S13A). As expected, inhibition of βcat activity was accompanied by a decrease in free cholesterol (filipin III) and membrane rigidity (Di-4) (Fig. S13B, S13C).

We further examined the role of free cholesterol in aberrant Wnt signaling in vivo on a WT APC background. For this purpose, we employed a Wnt signaling activity fly model co-expressing a ubiquitous Wnt reporter comprised of the LacZ gene under the control of a six pair TCF-Helper (6TH) binding site promoter system[107] and intestinal stem/progenitor cell-specific hLRP6-GFP expression under the control of the temperature sensitive esg-Gal4ᵀˢ system. This Wnt reporter was shown to be activated in intestinal stem/progenitor cells within the *Drosophila* midgut in response to changes in Wnt activation. Interestingly, hLRP6-GFP expression targeted to the larval eye imaginal disc by GMR-Gal4 resulted in adult eye phenotypes similar to a gain of function allele of Wingless (Wg) and Armadillo (Arm, encoding *Drosophila* β-catenin homolog)[108,109] (Fig. S14). Moreover, silencing the *Drosophila* Wnt transcription factor TCF (dTCF) rescued this hLRP6-induced eye phenotype, suggesting that the phenotype is induced by an overactivation of Wnt signaling (Fig. S14). To further assess the effect of free cholesterol on Wnt signaling activation in *Drosophila* intestinal stem/progenitor cells, TCF-reporter control flies (TCF-LacZ; esgG4ᵀˢ, GFP>+) or hLRP6-expressing flies (TCF-LacZ; esgG4ᵀˢ, GFP>hLRP6 eGFP) were fed diets with varying concentrations of cholesterol (low, standard and high). Remarkably, we found that Wnt reporter activation was induced in intestinal stem/progenitor cells only in flies fed a high cholesterol concentration diet, when Wnt signaling is activated by hLPR6 expression, which suggests that cholesterol supplementation can enhance Wnt signaling in vivo in intestinal stem/progenitor cells (Fig. 9I, J).

Finally, to confirm that the aberrant activation of βcat by mutant APC drives the documented loss of plasma membrane homeostasis, including the dysregulation of cholesterol, we tested the effect of aberrant activation of βcat on plasma membrane cholesterol and rigidity in the context of a WT APC background. In order to perform these experiments, cultured YAMC colonocytes (Apc+/+) were treated with an inhibitor of GSK3β, CHIR99021. This chemical compound has been shown to inhibit the β-catenin destruction complex thus leading to aberrant stabilization of βcat[110,111]. Our data indicate that CHIR99021 incubation increased both plasma membrane cholesterol and rigidity (Fig. S15). As expected, co-incubation with MβCD antagonized this effect. Collectively, these results demonstrate that the stabilization of βcat increases plasma membrane free cholesterol and rigidity.

## Discussion

The cell plasma membrane acts as a nexus merging extra- and intra-cellular components, which together enable many of the fundamental cellular signaling processes associated with CRC, including aberrant Wnt signaling[76]. From a functional perspective, Wnt factors organize to form specialized plasma membrane domains, i.e., signalosomes or nanoclusters[39,42]. Dysregulation of one or more of these components or the highly dynamic structures they form can promote oncogenic Wnt signaling[76]. Previous studies have identified APC status (WT vs mutant) as a critical determinant of CRC risk[15]. Here, we describe an intricate Wnt signaling-associated mechanism involving oncogenic truncated APC and the loss of plasma membrane cholesterol homeostasis, which drives aberrant Wnt signaling and CRC tumorigenesis. Furthermore, we show that oncogenic truncated APC induces alterations in the structure and organization of Wnt signaling nanoassemblies (biomolecular condensates) and their nearby membrane environment. Finally, our findings indicate that free cholesterol alone can recapitulate some of the phenotypes associated with oncogenic truncated APC in cellulo and in vivo. Together, these findings directly link the interplay between free cholesterol in the plasma membrane, Wnt biomolecular condensate machinery, and aberrant Wnt signaling in the context of cancer.

Historically, efforts have focused on elucidating the downstream effect of "loss of APC" in the intestine[15,63]. Based on these studies, it is evident that the coincident aberrant stabilization of βcat and changes in RNA expression of βcat-related genes induced by loss of APC, perturbs many cellular processes in the intestine such as differentiation, migration, proliferation, and apoptosis, consequently leading to the maintenance of APC mutant cells displaying "stem-like properties". The latter, a trademark of CRC, facilitate and promote CRC tumorigenesis[61,62]. Importantly, our results demonstrate that onco-genic truncated APC dysregulates plasma membrane free cholesterol homeostasis, which consequently directly modulates Wnt receptor spatiotemporal dynamics and downstream signaling. This is note-worthy, since the upstream activation of Wnt receptors at the plasma membrane has been shown to be required for CRC development, even in the presence of downstream oncogenic truncated APC[8–13]. We believe that the chronic increase in plasma membrane cholesterol (20–230% change among orthogonal models) and rigidity observed in mutant APC cells and their GMPVs in the absence of stimulation with Wnt3a ligand, highlight the important role of constitutive activation of Wnt signaling in these CRC models when compared to "healthy" cells. We also found that multiple APC truncations increased the interaction between Wnt receptors, i.e., LRP6 and Fzd7, and free cholesterol without Wnt3a stimulation. This is yet another key piece of evidence indicating the relevance of constitutive Wnt signaling in CRC.

In the context of CRC, early onset mutations of the first *Apc* allele, are mostly, if not always, followed by inactivation of the second allele (LOH)[43]. In the vast majority of CRC cases, this first mutation event leads to the expression of a truncated APC gene product. Notably, our findings demonstrate that oncogenic truncated-APC-dependent loss

of plasma membrane homeostasis might occur very early during CRC development. Consistent with this notion, the dysregulation of plasma membrane biochemical and biophysical properties documented herein occurred not only in the context of mutated homozygous (Apc−/−) but also in heterozygous (Apc−/+) models. These discoveries are consistent with previous published data showing that the expression of Wnt signaling target genes is significantly enhanced in cells and organoids derived from human familial adenomatous polyposis (FAP) patients carrying different heterozygous APC trun-cation mutations (FAP1 and FAP2)[112]. In contrast, a different cell model (FAP3) derived from a patient carrying a single APC mutation, resulting in a complete lack of APC protein expression, did not recapitulate the effects observed in the FAP1 and FAP2 models highlighting the importance of truncated APC. Moreover, organoids grown from human mutant FAP cells displayed a cyst-like growth pattern. Interestingly, this cyst-like morphology is consistent with previous findings showing that the growth pattern of specific orga-noids changes from branched to cyst-like structures as a result of increased Wnt signaling[113,114]. In agreement with these findings, other phenotypes associated with the Wnt signaling pathway and cancer are altered in the heterozygous APC mutant background. For exam-ple, a single mutated *Apc* allele was shown to increase cell pro-liferation, stem cell number in crypts, crypt fission, and upregulate neurogenic genes[112,115]. The latter has been reported to occur in prostate, head and neck, pancreatic and colon cancer. From a mechanistic dominant negative "gain of function" perspective, truncated mutant APC proteins can dimerize with full-length WT APC in Apc−/+ cells, resulting in the functional inactivation of full-length WT APC proteins[116,117]. Altogether, these data support the notion that multiple cancers are initiated by an APC mutation event.

Activating mutations in the Wnt pathway have been shown to dysregulate cellular cholesterol in CRC models[118,119]. Thus, it does not come as a surprise that several studies have reported that cancer cells exhibit higher levels of total cellular cholesterol and cholesterol-enriched lipid rafts compared to healthy cells[76]. Here we provide mechanistic findings that serve as the initial steps in deciphering the effects of mutant APC on cellular free cholesterol homeostasis. Our global transcriptome analyses suggest that oncogenic truncated APC-induced loss of cholesterol homeostasis is partly driven by alterations in genes encoding effectors associated with cholesterol uptake, efflux, synthesis, intracellular trafficking and metabolism, e.g., *Abca1*, *Abcg1*, *Apol11a*, *Cyb5r3*, *Dhcr7*, *Ebp*, *Fdft1*, *Fdps*, *Ggps1*, *Insig1(2)*, *Lrp8*, *Lpcat2*, *Mvk*, *Mvd*, *Olr1*, *Pmvk Scarf1*, *Scap*, *Srebf2*, and *Soat2*. In addition, we show a directional gene dysregulation consistent with the accumula-tion of free cholesterol in colonocytes. This is noteworthy, since selective targeting of the de novo cholesterol biosynthesis pathway in cells expressing oncogenic truncated APC is considered a potential anti-cancer therapeutic strategy[120]. Interestingly, many of these genes have been reported to facilitate key steps involved in CRC tumor-igenesis and/or act as potential CRC therapeutic targets[56,57,120–126]. Some of the aforementioned and alternative genes have also been linked to other types of cancer and processes associated with lipid metabolism, e.g., atherosclerosis[127–131].

From a biophysical perspective, our findings demonstrate that oncogenic truncated APC induces alterations in plasma membrane free cholesterol localization in Lo domains, resulting in transbilayer asymmetry and clustering. These findings also indicate that several of the biophysical features of Wnt signaling membrane domains are uniquely altered by oncogenic truncated APC, including plasma membrane rigidity and Lo domain stability. This is noteworthy, since disruption of cholesterol-enriched rafts can significantly improve the responsiveness to anti-cancer therapies[76]. Concordant with our find-ings, select anti-cancer drugs can also alter the protein content of these specialized plasma membrane domains[132]. Therefore, oncogenic truncated APC-mediated dysregulation of plasma membrane free

cholesterol and Wnt-associated raft components might play an important role during the early stages of CRC development.

From a biochemical perspective, the mechanisms linking loss of plasma membrane homeostasis and aberrant Wnt signaling-driven tumorigenesis in the intestine remain to be elucidated. Highly relevant to plasma membrane homeostasis and the CRC biology field, it is now recognized that the geometry of biological membranes is directly linked to signal processing capability[76,133]. Notably, Lo-Ld membrane domain structures and organization are precisely modulated by specific lipid-protein interactions, e.g., nanoclustering. Furthermore, nanoclusters give rise to functional lateral plasma membrane signaling domains, e.g., Wnt signalosomes, whose organization is highly cholesterol-dependent. Our findings suggest that oncogenic truncated APC-induced loss of plasma membrane cholesterol homeostasis leads to profound alterations in Wnt receptor nanocluster structure such as a robust increase in the translocation of cholesterol, LRP6, Fzd7, and Dvl1 into Wnt condensates. We also observed enhanced interactions between LRP6 and Fzd7 as well as increased Wnt receptor nanocluster size in the intestine. In addition, the frequency of nanocluster formation associated specifically with relatively larger Wnt clusters was augmented. These findings substantiate the notion that dysregulation of Wnt receptor nanoassemblies, along with the impairment of APC scaffolding activity, promote aberrant Wnt signaling in the intestine. Intertwined with this emerging scenario, previous studies describe the ability of plasma membrane lipids to aid in the formation of proteolipid membrane "hot spots" reminiscent of Lo domains that facilitate signaling events[93,134,135], including downstream stabilization of βcat[42,91].

Select lipid pools such as cholesterol, PI(4,5)P$_2$, PS, and phosphatidic acid (PA) are key structural components of Wnt nanoassemblies which reside in Lo regions. These dynamic lipids, in part due to their unique membrane-modulatory properties, can support interactions between Wnt nanocluster components and recruitment of important effectors required for the stabilization of activated Wnt biological condensates, which in turn may promote aberrant Wnt signaling[76]. These findings support a potentially highly relevant link between oncogenic truncated APC-induced (cholesterol-mediated) membrane rigidity and enhanced Wnt receptor nanoclustering. Indeed, there is evidence demonstrating that protein clustering and rigidity at the membrane can interactively modulate one another. For example, the tendency of protein molecules to organize into nanoclusters has the effect of rearranging the plasma membrane into specialized dynamic compartments thus altering membrane undulation and bending rigidity[136]. Furthermore, formation of relatively larger clustered protein lattices can lower lipid diffusion rates, a characteristic observed in Lo domains when compared to Ld domains, which can increase interactions between protein clusters and their neighboring lipids[136]. The latter was confirmed in our FLIM-FRET findings. Conversely, membrane rigidity or lipid packing can modulate protein nanoclustering. For example, increased membrane order of a specific membrane region caused by lipid-lipid interactions between cholesterol and highly saturated lipids can lead to clustering of resident cholesterol molecules and increased membrane disorder of the domain surroundings[137]. In turn, clustered cholesterol molecules preferentially accommodate protein molecules with high affinity towards this lipid, while excluding proteins exhibiting low cholesterol affinity. Consequently, the formation of protein "islands" in these rigid cholesterol-enriched domains is observed[137]. Our observations that oncogenic truncated APC-induced loss of plasma membrane homeostasis dramatically enhanced both free cholesterol and PI(4,5)P$_2$ interactions with LRP6, Fzd7, and Dvl1 are consistent with these previous findings.

Surprisingly, our data suggest that LRP6 and Fzd7 display distinct sensitivity towards the dysregulation of plasma membrane cholesterol. From a mechanistic point of view, previous studies have shown that free cholesterol is enriched around LRP5/6 molecules in the absence of Wnt3a and, following Wnt3a stimulation, Fzd translocates to this cholesterol-enriched microenvironment through LRP5/6 binding[32]. Concurrently, cytosolic Dvl is recruited to the LRP5/6-Wnt-Fzd active complex through recognition of Fzd, plasma membrane IL free cholesterol, and PI(4,5)P$_2$ by the PDZ domain in Dvl. With regard to cholesterol, we demonstrate that this lipid displays relatively high and low interaction in the resting state (no Wnt3a ligand), with LRP6 and Fzd7, respectively, and that oncogenic truncated APC promotes this rearrangement. However, following Wnt3a stimulation both Wnt receptors display similar enhanced interaction with free cholesterol on a mutant APC background. These results are consistent with the notion that Wnt3a stimulation rearranges the localization of Wnt receptors at the plasma membrane to bring them in closer proximity to free cholesterol-enriched Wnt signaling hot spots. This scenario is further supported by previous studies showing that both suppression of LRP5/6 expression and depletion of free cholesterol reduce the Wnt3a-induced co-localization of Fzd and Dvl[32]. With respect to PI(4,5)P$_2$, our data demonstrate that interaction with LRP6 and Fzd7 is highly dependent on Wnt3a stimulation on an oncogenic truncated APC background. In agreement with our observations, prior studies have shown that Wnt3a can stimulate the synthesis of PI(4,5)P$_2$ and that PI4KIIα and phosphatidylinositol-4-phosphate 5-kinase type I (PIP5KI), two kinases required for PI(4,5)P$_2$ synthesis, are essential for the nuclear accumulation of βcat[138]. Interestingly, the mechanism regulating PI(4,5)P$_2$ synthesis appears to be dependent on Dvl, which has been shown to induce PIP5KI activity in a dose-dependent manner and coimmunoprecipitates with PIP5KI in cell lysates[87]. Interactions between PIP5KI and Dvl have been shown to be Wnt ligand-dependent. That being said, further research is required to investigate whether other anionic lipids such as PS and PA are disrupted by oncogenic truncated APC.

Interactions between members of Wnt biological condensates, which localize to lipid rafts in order to signal properly[20,24], were also modulated by oncogenic truncated APC. In the context of Wnt pathway activation, localization of LRP5/6 and Fzd to Lo domains potentiate Lypd6 cluster-dependent LRP5/6 phosphorylation[20] and cholesterol/PI(4,5)P$_2$-dependent Fzd binding of Dvl[32], which in turn stabilizes Wnt biological condensates and induces downstream signaling. Notably, our data suggest that loss of plasma membrane homeostasis enhances Wnt proteolipid condensate stability and activation. In the presence of Wnt3a, we observed increased Lypd6 clustering as well as increased interactions of Dvl1 with free cholesterol, PI(4,5)P$_2$, and Wnt receptors at the plasma membrane IL. Consistent with our findings, previous studies have shown that following formation of the Wnt-LRP5/6-Fzd complex and Wnt ligand-induced enrichment of free cholesterol in activated Wnt nanoclusters, cytosolic Dvl is recruited to the plasma membrane IL through its coincident recognition of Fzd, free cholesterol and PI(4,5)P$_2$[32]. Interactions between membrane free cholesterol and Dvl promote the scaffolding function of Dvl by prolonging interactions of Dvl with Fzd and other cytosolic proteins comprising Wnt biological condensates, e.g., Axin and Ck1γ, thereby facilitating Wnt proteolipid condensate activation and endocytosis. It is noteworthy that GPI-anchored proteins, like Lypd6, have also been described to aid in the recruitment of kinases at the IL of the plasma membrane[20]. Lastly, the binding of Dvl to free cholesterol at the plasma membrane IL has been proposed to follow a stoichiometric Langmuir-type pattern, which suggests that the activity of Dvl would respond spontaneously to localized changes of free cholesterol levels[32]. Therefore, it is possible that a chronic change in plasma membrane free cholesterol levels could modulate the function of Dvl and potentially other cytosolic components of the Wnt cluster machinery, e.g., Axin.

Overactivation of the Wnt signaling pathway and adenoma formation is the first step in CRC development[139]. Interestingly, restoration of APC function prompts rapid polyp regression and the

reemergence of a relatively healthier intestinal state, suggesting that colonic transformation is dependent on an optimal level of Wnt signaling activity[64]. Supporting this notion, the "just right" hypothesis states that the amount of aberrant Wnt signaling is titrated to induce CRC tumorigenesis[140]. Remarkably, our findings have shown that critical phenotypes, e.g., loss of cholesterol homeostasis and disruption of Wnt condensate structure, exhibit a dependence on the level of Wnt signaling dysregulation in colonic crypts (double mutant allele Apc 580/580 > single mutant allele Apc 580/+ > WT Apc +/+) and cultured colonocytes (Apc 850/+ βcat −/+ > Apc 850/+ > Apc +/+). Most significantly, exogenous intervention with cholesterol-lowering drugs, e.g., statins, were able to rescue the phenotype back to WT levels. Thus, it is reasonable to propose that loss of plasma membrane homeostasis could serve as a potential drug target to modulate the level of aberrant Wnt signaling outside a "just right" state in CRC as well as key signaling pathways involved in other detrimental human pathologies[141–144]. In future studies, it would be of interest to further elucidate how oncogenic truncated APC alters cholesterol homeostasis and its levels at the plasma membrane in CRC patients, including truncations related to poor CRC prognosis.

Our study has multiple strengths. First, to comprehensively assess the role of oncogenic truncated APC in reshaping plasma membrane cholesterol, proteolipid hierarchical organization and downstream Wnt signaling, we used mouse, fly and human orthogonal models. Second, we combined total internal reflection fluorescence (TIRF) with direct stochastic optical construction microscopy (STORM) and widefield fluorescence lifetime imaging microscopy combined with fluorescence resonance energy transfer (FLIM-FRET) for the analysis of proteolipid membrane nanoclustering within Wnt macromolecular condensates. Third, we demonstrate that free cholesterol alone can recapitulate some of the phenotypes associated with oncogenic truncated APC-induced loss of plasma membrane homeostasis. Fourth, we conducted experiments to show that stabilization of βcat increases plasma membrane free cholesterol and rigidity, thus linking downstream to upstream Wnt signaling and further highlighting the non-autonomous nature of mutant APC in CRC. Finally, from a translational perspective, since the APC "gatekeeping" gene is a key regulator of Wnt signaling and ~90% of all colon cancer cases are associated with defects in the Wnt signaling pathway, our observations support the development of plasma membrane-targeted therapies to reduce cancer risk.

Our study also has some limitations. First, since free cholesterol can be transported through vesicular and non-vesicular mechanisms from the plasma membrane to other organelles in a dynamic and bidirectional fashion, there is a need to systematically determine to what extent intracellular trafficking and endomembranes are affected by mutant APC and how these interact with the plasma membrane. Second, further research is required to assess the therapeutic feasibility of using strategies to target plasma membrane cholesterol hierarchical organization to reduce aberrant Wnt signaling and cancer risk. Finally, the effects of cooperative oncogenic driver gene mutations were not assessed.

To conclude, we describe a critical role for mutant APC in shaping plasma membrane cholesterol-enriched Lo domains, membrane rigidity, Wnt receptor dynamic nanoassemblies, and their interactions with key lipid and protein effectors within Wnt condensates. These findings corroborate the concept that plasma membrane cholesterol and plasma membrane-associated Wnt signaling are essential drivers of CRC development in the presence of mutant APC. To our knowledge, this is the first evidence directly linking the interplay between mutant APC, the levels of free cholesterol in the plasma membrane, membrane order, aberrant Wnt signaling, and cholesterol homeostatic pathways in cancer. We also document the unique ability of exogenous cholesterol to disrupt plasma membrane homeostasis and drive Wnt signaling autonomously in *Drosophila* intestine and cultured cells. In

contrast, cholesterol-lowering drugs suppress these phenotypes. A mechanistic model for the effects of oncogenic truncated APC-driven dysregulation of plasma membrane biochemical and biophysical properties in relation to downstream Wnt signaling is shown in Fig. 10. Altogether, these findings suggest that therapeutic approaches aimed at restoring plasma membrane cholesterol homeostasis and proteolipid spatiotemporal dynamics could prove an effective strategy to reduce cancer risk.

## Methods

### Mice genetics, husbandry, diet, and study design
All experiments utilizing mice were approved and conducted in rigorous accordance with the Texas A&M University Institutional Animal Care and Use Committee (IACUC 2023-0050) and conformed to NIH guidelines. To generate a conditional colonic-targeted mutant APC mouse model (Fig. 2), C57 black 6 (C57BL/6) CDX2P-CreERT2 mice (IMSR Cat# JAX_022390, RRID:IMSR_JAX:022390) were crossed with C57BL/6 Lgr5-EGFP-IRES-CreERT2-APC^loxP/+ mice[145] to generate GC (Apc +/+), AGC (Apc−/+), and AfGC (Apc−/−) C57BL/6 mice. GC, AGC, and AfGC mice were housed in cages in a temperature- (68° to 79° F; set point 73° F) and humidity- (50–70%) controlled animal facility with a 12 h light/dark cycle and fed standard chow. For all in vivo experiments, C57BL/6 mice were weaned at 3 weeks of age, acclimated, and fed a semi-purified modified AIN-76A rodent diet with 10% (w/w) safflower oil, 21% casein, 43% sucrose, 16% cornstarch, 5% Solka Floc FCC200 (cellulose), 3.7% AIN-76 mineral mix, 0.1% AIN-76 vitamin mix, 0.3% methionine, and 0.2% choline bitartrate (Research Diets Inc, D03092902) for 2 wks prior to induction of oncogenic APC580D in the colon by injection of tamoxifen (Fig. 4). C57BL/6 mice were provided with fresh diet every three days. All animals had access to food and water at all times. After 16–24 weeks of age (4–6 months), C57BL/6 mice were intraperitoneally injected once per day for four consecutive days with 100 mg/kg tamoxifen (Sigma-Aldrich, T5648) dissolved in corn oil (carrier) to induce expression of oncogenic APC580D, fed fresh diet (every three days) and weighted every single day until termination. Mice were terminated by $CO_2$ asphyxiation followed by cervical dislocation at the times indicated in each study. The guidelines associated with tumor induction in rodents by the Institutional Animal Care and Use Committee (IACUC) stipulates that the maximal tumor volume should not exceed 2000 mm³. In our studies involving C57BL/6 mice polyposis, none of the colonic polyps exceeded 2000 mm³.

### Cell culture
Conditionally immortalized young adult mouse colonic epithelium (YAMC), Immortomouse × MIN colonic epithelium (IMCE) and IMCE β-catenin (IMCE βcat) isogenic cell lines were seeded, incubated with RPMI-1640 media (ThermoFisher, 21870076) supplemented with 1% GlutaMAX-I (ThermoFisher, 35050061), 5% Fetal Bovine Serum (FBS) (HyClone, SH30070.03), 0.1% ITS Premix Universal Culture Supplement (Corning, 354351), 1% Penicillin-Streptomycin (PenStrep) (ThermoFisher, 15140148) and 10% IFN-γ (Sigma, 11276905001) (IFN-γ was added freshly to media immediately prior to use) (complete mouse RPMI-1640) under permissive conditions, 33 °C with 5% $CO_2$, and grown to 70–90% confluency after 48 h. HCT116Δ, HCT116, SW480, and DLD1 human epithelial colorectal cancer cells were seeded, incubated with high glucose RPMI-1640 (ATCC, 30-2001) with L-glutamine and HEPES supplemented with 10% FBS and 1% PenStrep (complete human RPMI-1640) at 37 °C with 5% $CO_2$ and grown to 70–90% confluency after 48 h. HT29 cells were seeded, incubated with McCoy's 5A media (Sigma, M8403) supplemented with 10% Fetal Bovine Serum (FBS) (HyClone, SH30070.03), 1% GlutaMAX-I and 1% PenStrep (ThermoFisher, 15140148) (complete McCoy's 5A). In the case of the commercially available Leading Light® Wnt reporter cell line (3T3 LL) (Enzo, ENZ-61002-0001), cells were seeded, incubated with DMEM

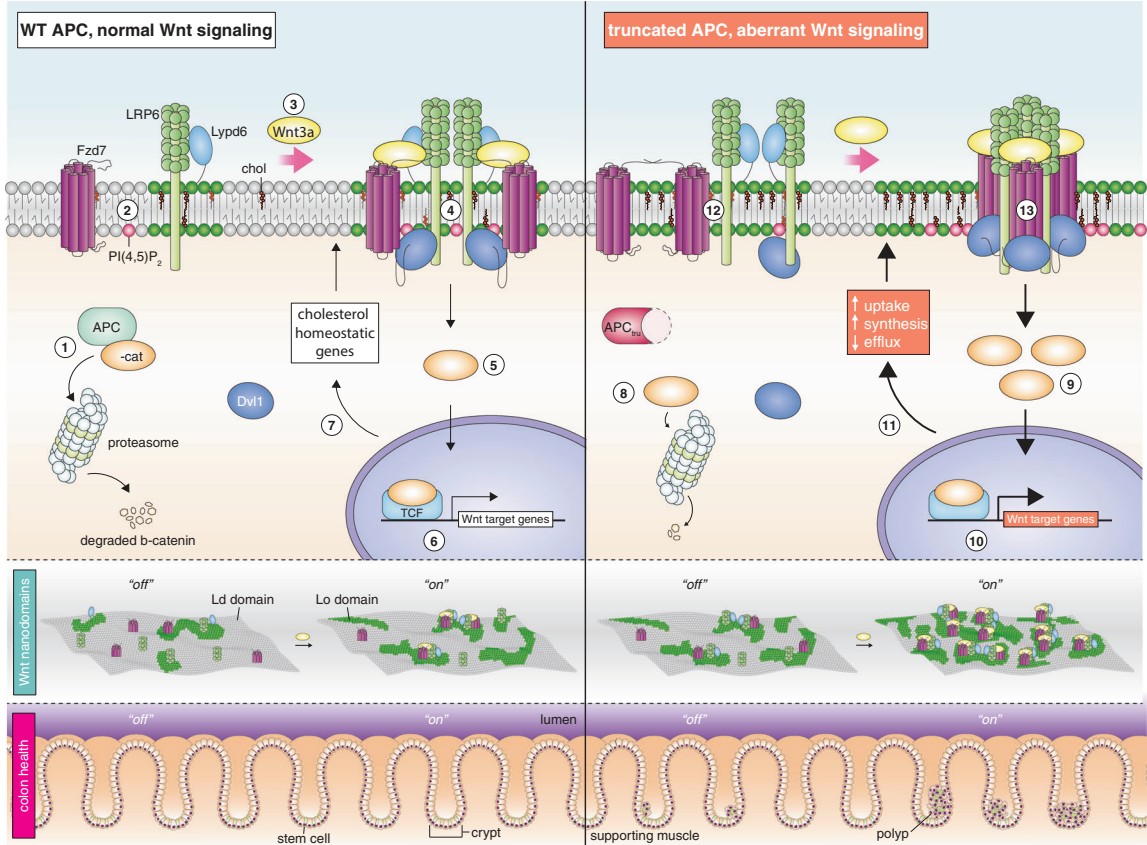

**Fig. 10 | Link between oncogenic APC, loss of plasma membrane homeostasis, and colon cancer development.** The plasma membrane serves as a nexus integrating extra- and intracellular Wnt pathway modulators, which by means of their specific organization at the plasma membrane play an essential role in the homeostatic maintenance of the colonic crypt. (**1**) WT APC tightly regulates the levels of stabilized βcat by facilitating its degradation via the proteasome, (**2**) turning off upstream Wnt signaling activation. (**3**) In the presence of Wnt3a ligand, the canonical Wnt signaling pathway is activated. (**4**) Activation involves increased localization of IL free cholesterol and PI(4,5)P$_2$, Wnt receptor/effector clustering and translocation of Wnt-associated cytosolic proteins, e.g., Dvl1, at the plasma membrane. (**5**) Consequently, Wnt condensates drive an increase in the levels of stabilized βcat, which in turn, regulates the (**6**) transcription of Wnt target genes, (**7**) including those involved in maintaining membrane cholesterol homeostasis. Together, these processes ensure a normal/healthy colonic epithelium. In contrast, mutation of *Apc* leads to the expression of truncated APC protein, resulting in the dysregulation of key Wnt signaling-associated cellular steps. Firstly, truncated APC elicits (**8**) a decrease in βcat degradation, which leads to its (**9**) aberrant stabilization. (**10**) The increase in stabilized βcat modulates the transcription of (**11**) Wnt target genes, including those involved in cholesterol uptake, synthesis, and efflux. (**12**) The loss of cholesterol homeostasis increases the levels of plasma membrane cholesterol and rigidity, which in turn alters the spatial temporal dynamics of lipid rafts (Lo domains) as well as lipid raft localization and interactions of Wnt signaling-associated receptors and lipid/protein effectors within Wnt condensates in an unstimulated state (no Wnt3a). (**13**) In the presence of Wnt3a, truncated APC exacerbates oncogenic Wnt signaling by increasing IL cholesterol, PI(4,5)P$_2$ levels, Lo domain stability, and the number of Wnt receptors/effectors at the plasma membrane. This dysregulates Wnt receptor/effector spatial temporal dynamics and nanoclustering, thereby promoting feedforward activation of aberrant βcat signaling and tumorigenesis. "Off", absence of Wnt3a ligand; "On", presence of Wnt3a ligand.

(ThermoFisher, 10569010) supplemented with 1% PenStrep and 10% FBS (complete DMEM) at 37 °C with 5% CO$_2$ and grown to 70–90% confluency after 48 h. To grow HAP1 cell lines, these were seeded, incubated with Iscove's modified Dulbecco's medium (IMDM), 10% FBS, and 1% PenStrep (complete IMDM). YAMC and IMCE cell lines were kindly provided by Robert H. Whitehead (Ludwig Cancer Institute, Melbourne, Australia). The IMCE βcat cell line was kindly provided by Lynn M. Matrisian (Department of Cancer Biology, Vanderbilt University School of Medicine, Nashville, TN). YAMC, IMCE, and IMCE βcat cell lines were authenticated by STR profiling and an interspecies contamination test (CellCheck) performed at IDEXX BioResearch. HCT116Δ (Horizon Discovery, HD 104-009), HCT116 (ATCC, CCL-247), SW480 (ATCC, CCL-228), DLD1 (ATCC, CCL-221), and HT29 (ATCC, HTB-38) cell line characterization was performed by vendors via STR profiling. All cell lines employed in our experiments tested negative for mycoplasma bacteria as assessed by a universal mycoplasma detection kit (30-1012K; American Type Culture Collection, Manassas (ATCC), VA).

To remove cholesterol from the plasma membrane of cell lines using methyl-beta-cyclodextrin (MβCD), seeded cells were washed with DPBS and incubated with 10 mM MβCD (unless otherwise indicated) at 33 °C with 5% CO$_2$ for 30 min. After incubation, cells were washed with DPBS (2x) and utilized in experiments as dictated by protocols. To enrich the plasma membrane of cell lines with cholesterol, cells were incubated with a water soluble MβCD:cholesterol complex (Sigma-Aldrich, C4951). For this purpose, previously seeded cells were washed with DPBS, incubated with varying concentrations of complexed cholesterol at 33 °C with 5% CO$_2$ for 3 h, and washed with DPBS to remove free MβCD. Subsequently, cells were utilized in experiments as dictated by protocols.

For transfection and expression of FLIM-FRET probes, plasmids were mixed with Lipofectamine™ 3000 reagent and P3000 reagent (ThermoFisher, L3000001) in OPTI-MEM media (ThermoFisher, 31985070) and incubated at room temperature for 10 min. The DNA complex was added to previously seeded colonic cells (YAMC, IMCE, and IMCE βcat) in an 8-well coverglass slide and cells were kept at 33 °C

for 24 h. After 24 h, cells were washed DPBS (2x), fixed in 4% paraformaldehyde (PFA), and washed with cold DPBS to remove fixative and imaged.

## Generation of giant plasma membrane vesicles (GPMVs)

To examine the effects of oncogenic APC on plasma membrane biochemical and biophysical characteristics, giant plasma membrane vesicles (GPMVs) were generated as described previously[50,51]. This membrane model retains most of the plasma membrane original native membrane diversity and components while allowing an examination of its properties without the potential interference of membranes from other cellular organelles. Briefly, colonic cell lines (YAMC, IMCE, and IMCE βcat) were seeded in a 6-well plate using sterile cell culture techniques, incubated with complete RPMI 1640 at 33 °C, 5% $CO_2$ and grown to 80–90% confluency. Subsequently, cells were washed with DPBS (2x) and GPMV buffer (2x, 10 mM HEPES, 150 mM NaCl, 2 mM $CaCl_2$, pH 7.4). To induce vesicle formation, cells were incubated with vesiculation buffer (GPMV buffer + 25 mM PFA, 2 mM DTT) at 37 °C with 5% $CO_2$ atmospheric pressure for 2 h. GPMVs were readily visible at ×20 magnification as free-floating spheres displaying a dark contrast (plates were gently skaken to locate vesicles). Following GPMV formation, supernatants were collected into a low-retention microcentrifuge tube and spun down at $500 \times g$ for 5 min to pellet cells/debris. To produce a relatively pure GPMV solution, the supernatant containing GPMVs was transferred to a clean low-retention microcentrifuge. To generate mouse crypt-derived GPMVs, colonic crypts were washed with DPBS (2x) and GPMV buffer (2x), placed in 6-well plates, and incubated with vesiculation buffer (10 mM HEPES, 100 mM NaCl, 2 mM $CaCl_2$, 2 mM dithiothreitol (DTT), 25 mM PFA) for 2 h at 37 °C. Following incubation, colonic crypts were fixed with 4% PFA and stained with 50 μg/mL filipin III (Sigma, SAE0087) or alternatively with 1 μM Di4 to measure plasma membrane free cholesterol and membrane order, respectively. Imaging was performed using a Ti2-E with an A1R scanner module and A1 camera confocal microscope using a 1.4 numerical aperture ×100 Plan Apo oil objective. Image fluorescence analysis was performed using Fiji (ImageJ).

## Measurement of temperature dependence on GPMV phase separation

To examine the effect of oncogenic truncated APC on temperature-dependent phase separation, cell-derived GPMVs were generated as described above, and stained with 5 μg/mL FAST DiI (ThermoFisher, D7756), a Ld domain marker. Approximately, 25–50 μL of a solution containing fluorescently-labeled GPMVs was placed inside an in-house constructed imaging chamber[51] composed of a 24 × 60 mm coverslip atop a BSA-coated 18 × 18 mm coverslip held together by silicone vacuum grease. The temperature of the GPMV solution was modulated using a CL-100 single-channel bipolar temperature controller (Warner Instruments, USA) fitted to an imaging chamber (Warner Instruments, USA) and an LCS-1 heat exchanger (Warner Instruments, USA). The temperature of the GPMV solution was monitored at the image chamber site using an external TA-29 cable with bead thermistor. For data acquisition, the temperature was set below 15 °C and modulated in small increments up to 40 °C. The GPMV solution was allowed to equilibrate for 10 min in between temperature setting changes. Images were acquired using a Nikon Ti2-E confocal microscope with an A1R scanner module and A1 camera using a 1.3 numerical aperture ×40 Plan Apo oil objective and processed using NIS-Elements AR 5.41.02 software.

## Organoid culture

Patient-derived cells from healthy or CRC individuals obtained from colon biopsies were embedded in growth factor-reduced Matrigel (Corning, 356231). Following Matrigel polymerization, embedded cells were overlaid with normal human organoid (Advanced DMEM/F12

(Gibco, 12634010) supplemented with 1% GlutaMAX-I (Gibco, 35050-061), 1% PenStrep (Gibco, 15140-122), 10 mM HEPES, 50 ng/mL EGF (Life Technologies, PHG0311), 1 μM N-acetyl-l-cysteine (Sigma, A9165), 1X N2 (Life Technologies, 17502-048), 1X B27 (Life Technologies, 12587-010), 10 nM [Leu15]-Gastrin 1 (Sigma, G9020), 10 mM Nicotinamide (Sigma, N3376), 500 nM A83-01 (Tocris, 2939), 3 μM SB202190 (Sigma, S7067), 100 μg/mL primocin (Invivogen, #ant-pm), 10 μM Y-27632 (Sigma, Y0503), 50% Wnt conditioned medium) or CRC human organoid media (Advanced DMEM/F12 supplemented with 1% GlutaMAX-I, 1% PenStrep, 10 mM HEPES, 50 ng/mL EGF, 100 ng/mL hNoggin (PeproTech, 120-10C), 1 μM N-acetyl-l-cysteine, 1X N2, 1X B27, 10 nM [Leu15]-Gastrin 1, 10 mM Nicotinamide, 500 nM A83-01, 3 μM SB202190, 10 nM prostaglandin $E_2$, 100 μg/mL primocin, 10 μM Y-27632) media, respectively. Y-27632 was withdrawn from the organoid media 2 days after plating and continued to be grown into PDOs in their respective media. During organoid growth, the media was changed every 2 days. The oligonucleotide sequences of Crispr-Cas9 primers and reaction protocols employed herein are available in the literature[66].

## Fluorescence-assisted cell sorting of colonic stem cells

Mouse colons were removed, washed with cold PBS without calcium and magnesium (PBS$^{-/-}$), everted on a disposable mouse gavage needle (Instech Laboratories), and incubated in 15 mM EDTA in PBS$^{-/-}$ at 37 °C for 35 min as previously described[146]. Subsequently, following transfer to chilled PBS$^{-/-}$, crypts were mechanically separated from the connective tissue by rigorous vortexing. For intestinal stem cell isolation, crypt suspensions were dissociated to individual cells with 0.25% Trypsin-EDTA containing 200 U/ml DNase. Cell suspensions were then filtered through a 40-μm mesh and GFP-expressing Lgr5$^+$ stem cells were collected using a Bio-Rad S3e Cell Sorter and processed using the ProSort 1.6 software (Fig. S16). Dead cells were excluded by staining with propidium iodide. Sorting purity was routinely examined and over 95%. Sorted stem cells were collected in sorting-collection medium (1X B27, 1X N2, 1 μM N-acetylcysteine, 10 μM Y-27632 in Live Cell Imaging Solution (LCIS, ThermoFisher, A14291DJ)), counted and aliquots prepared for subsequent experiments.

## In cellulo free cholesterol measurement via Filipin III microscopy

To investigate the effect of oncogenic APC on plasma membrane free cholesterol, seeded cells were washed with DPBS, fixed in 4% PFA at RT, and washed again with cold DPBS to remove fixative. Subsequently, fixed colonocytes and their derived GPMVs were incubated with 50 μg/mL filipin III (Sigma, SAE0087) in DPBS for 45 min on ice in the dark, washed with cold DPBS to remove unbound filipin III and imaged as described[147]. In some cases, cells were co-incubated with a solution of filipin III (50 μg/mL) and CellMask™ DeepRed PM stain (1x). Fluorescence imaging was performed with a 1.3 numerical aperture ×40 Plan-Fluor oil objective mounted on a wide field Nikon Eclipse microscope or an Amnis FlowSight® flow cytometer equipped with the Inspire for FlowSight 200.1.680.0 collection software (Luminex, Austin, TX) and processed with NIS-Elements AR 5.41.02 or IDEAS® 6.2 software, respectively. Filipin III is a naturally fluorescent polyene antibiotic that specifically binds to free cholesterol but not to esterified sterols[48]. To generate fluorescence images, filipin III was excited at 405 nm and the fluorescence emission intensity was recorded using an appropriate DAPI filter set (microscopy) or across a channel corresponding to 430–505 nm wavelength range (flow cytometry). Cellular morphology was examined using differential interference contrast (DIC) microscopy. All microscopy image analysis was performed using ImageJ and a custom-built macro that generates a plasma membrane mask to analyze filipin III fluorescence intensity. Flow cytometry data were analyzed using the image data exploration and analysis software IDEAS® 6.2.

## Cholesterol starvation assay

YAMC (APC +/+), IMCE (APC 850/+) and IMCE βcat (APC 850/+ βcat −/+) cell lines were initially maintained in RPMI-1640 media supplemented with 1% GlutaMAX-I, 5% Fetal Bovine Serum (FBS), 0.1% ITS Premix Universal Culture Supplement, 1% Penicillin-Streptomycin (PenStrep) and 10% IFN-γ (IFN-γ was added freshly to media immediately prior to use) (complete mouse RPMI-1640) under permissive conditions, 33 °C with 5% CO$_2$, and grown to 70–90% confluency after 48 h. Subsequently, cells were trypsinized by a 5 min incubation with 0.05% trypsin-EDTA (Gibco, 25300-054), washed, counted using an automated cell counter (Countess II, Invitrogen, USA) and 200,000 cells seeded overnight to allow cell attachment. Cells were washed with DPBS, and incubated with regular FBS supplemented medium (1.4 mg/dL cholesterol, Hyclone USA) or, alternatively, with RPMI-1640 supplemented with lipoprotein depleted FBS (0.04 mg/dL cholesterol, 35 times lower than regular FBS; Kalen Biomedical LLC, USA). Cells were grown for 24 h or, alternatively, 72 h until 80% confluency, trypsinized, counted, and fixed with 4% PFA. Free cholesterol was quantified using filipin III as described above. Plasma membrane filipin III fluorescence was measured using an Amnis FlowSight® flow cytometer (Luminex, Austin, TX).

## In cellulo measurement of plasma membrane rigidity/order via Di4 microscopy

To elucidate the effect of oncogenic APC on plasma membrane rigidity, previously seeded cells and their derived GPMVs were incubated with 5 μM (confocal microscopy) or 1 μM (imaging flow cytometry) Di-4-ANEPPDHQ (Di4) (ThermoFisher, D36802) in Leibovitz's L-15 medium (L-15) and imaged immediately by flow cytometry or confocal microscopy in order to avoid dye internalization. Fluorescence images were acquired with a 1.15 numerical aperture ×40 plan apochromat oil objective mounted on a Leica TCS SPEII RYBV automated DMi8 confocal microscope (Wetzlar, Germany) or an Amnis FlowSight® flow cytometer (Luminex, Austin, TX) and processed with LAS X 3.5.7.23225 or IDEAS® 6.2 software, respectively. Di4, a solvatochromic dye, provides a rapid evaluation of the plasma membrane lipid environment, e.g., lipid phase order/rigidity by allowing the generation of generalized polarization (GP) images[68,148]. To produce generalized polarization (GP) images, Di4 was excited at 488 nm and the fluorescence emission intensity of Di4 was recorded across two different channels representing ordered (O: 500–580 nm) and disordered (D: 620–700 nm) wavelength ranges for confocal microscopy or ordered (O: 480–560 nm) and disordered (D: 640–745 nm) wavelength ranges for flow cytometry to allow the examination of ordered and disordered membrane domains. The same laser power and detector settings were used for every experiment. For confocal microscopy images, processing was conducted using Fiji/ImageJ (National Institutes of Health) software, with a custom-built macro to convert microscopy images from Leica (.lif) to TIFF (.tif) files and a publicly available GP analysis plugin to generate GP data images. Flow cytometry data was analyzed using the image data exploration and analysis software IDEAS® 6.2. The GP value (relative membrane rigidity) was calculated using the following equation:

$$GP = \frac{Intensity_{ordered} - Intensity_{disordered}}{Intensity_{ordered} + Intensity_{disordered}} \tag{1}$$

## In vivo measurement of plasma membrane rigidity/order and free cholesterol

In order to perform studies in live colonic crypts, derived single colonocytes and isolated Lgr5$^+$ stem cells, oncogenic APC was induced for the indicated times and mice were terminated. Immediately after termination, the colon was rapidly removed, flushed with ice-cold DPBS, and a longitudinal section of tissue (~1/3 colon) was immediately fixed in 4% paraformaldehyde (PFA; Electron Microscopy Sciences, 15713-S) for hematoxylin and eosin (H&E) staining (histopathology analysis). The remaining colonic tissue was utilized for crypt, single colonocyte and stem cell studies. To examine the effect of oncogenic APC on plasma membrane rigidity, live colonic crypts and their derived single colonocytes were isolated as described previously[146,149]. Following single colonocyte isolation, GFP-high (stem cell) and GFP-low (daughter cell) cell populations were selectively collected using a Bio-Rad S3e Cell Sorter (Hercules, California, USA)[149]. GFP-high (Lgr5$^+$) cells were used for all stem cell experiments. Following isolation, mouse colonic crypts, their derived single colonocytes and Lgr5$^+$ stem cells were incubated with 5 μM (confocal microscopy) or 1 μM (imaging flow cytometry) Di4 (ThermoFisher, D36802) and immediately imaged with a 1.15 numerical aperture ×40 plan apochromat oil objective mounted on a Leica TCS SPEII RYBV automated DMi8 confocal microscope (Wetzlar, Germany) or an Amnis FlowSight® flow cytometer (Luminex, Austin, TX). Generalized polarization (GP) images were obtained as described above.

To elucidate the effect of oncogenic APC on plasma membrane free cholesterol, live colonic crypts were isolated, fixed in 4% PFA, washed with cold DPBS to remove fixative, incubated with 50 μg/mL filipin III (Sigma, SAE0087) in DPBS for 45 min on ice in the dark, washed with cold DPBS to remove unbound filipin III and imaged with a 1.3 numerical aperture ×40 Plan-Fluor oil objective mounted on a wide field Nikon Eclipse microscope. Images were processed using NIS-Elements AR 5.41.02 software. To generate filipin III images, crypts stained with filipin III were excited at 405 nm and the fluorescence emission from filipin III intensity was recorded using an appropriate DAPI filter set. All image analysis was performed using ImageJ (ImageJ, RRID:SCR_003070).

## RNA isolation and sequencing

**Cell lines.** YAMC (Apc +/+), IMCE (Apc 850/-), and IMCE βcat (Apc 850/+ βcat−/+) cell lines were cultured until ~80% confluent under permissive conditions (33 °C and 5% CO$_2$) in RPMI 1640 medium, no glutamine (21870076; Gibco), supplemented with 5% fetal bovine serum (FBS, HyClone), 2 mM GlutaMAX (Gibco), 5 μg/mL insulin, 5 μg/mL transferrin, 5 ng/mL selenious acid (Corning), and 5 IU/mL of murine interferon-γ (Roche). Three separate experiments were conducted for each cell line.

**Mice.** AfGC (Apc 850/850), AGC (Apc 850/+), and GC (Apc+/+) mice, were fed a 5% corn oil diet, 2 weeks prior to colonic induction of oncogenic Apc by injection of tamoxifen (100 mg/kg, four times) and termination after 3 (AfGC and GC) or 10 (AGC) weeks (n = 4 per group). RNA from cell lines and mouse colonic crypts was extracted using a RNAeasy Micro Kit (Qiagen) followed by DNase treatment using an DNA-free DNA Removal Kit (Invitrogen) in accordance with the manufacturer's protocol. RNA quality was assessed using an Agilent Bioanalyzer (Agilent Technologies). Only samples with an RNA quality of RIN ≥ 8 were used. RNA concentration was measured using spectrophotometry. Samples were subsequently sequenced using a TruSeq Illumina Stranded mRNA kit (Illumina). Sequencing was performed on a NextSeq 500 (Illumina). Data analysis was conducted following removal of all genes with counts per million values <1.

**Human.** Raw data was downloaded from the Gene Expression Omnibus (GEO: http://www.ncbi.nlm.nih.gov/geo); 36 samples (normal "uninvolved" colon and primary CRC) were generated from 18 CRC patients[150].

## RNAseq analysis

As described previously[151], RNA-seq data were normalized using the upper-quartile method with publicly available R software, EdgeR 3.42.4. In addition, EdgeR-robust in several contrasts was used to

identify differentially expressed genes. To account for multiple testing, the Benjamini-Hochberg (BH)[152] false discovery rate (FDR) procedure was utilized and genes were considered to be DE if the corresponding corrected p-values were less than 0.05. A gene target list based on prior data collected from cholesterol, sphingolipid, and fatty acid metabolic literature (~290 genes) was used for volcano plots (see Supplemental Table 1 for details). Gene set enrichment analysis (GSEA 4.1.0, UC San Diego, Broad Institute) analysis was also performed to identify enriched biological pathways and enrichment plots. Based on GSEA, differentially expressed genes related to the WNT signaling pathway were used for differential expression analysis.

### Western blotting
Colonocytes derived from cell lines or PDO were washed with DPBS, lysed with RIPA lysis buffer containing 50 mM Tris-HCl, 150 mM NaCl, 1 mM EDTA, 0.1% NP-40, 0.5% sodium deoxycholate, 0.1% sodium dodecyl sulfate, 2.5 mM sodium pyrophosphate, 10 mM β-mercaptoethanol supplemented with protease inhibitor cocktail (Sigma) and alkaline phosphatase inhibitor cocktail (Thermo-Scientific). Lysates were incubated at 37 °C for 45 min and subjected to standard SDS-PAGE procedure. SDS-PAGE gels were then washed with 20% ethanol for 12 min. Protein transfer was performed using a Trans-Blot Turbo transfer system (Bio-Rad, USA) and western blotting was subsequently performed using the following primary antibodies: anti-APC for mouse colonocytes (1:1000, OP44, EMD Millipore); anti-APC for human colonocytes (1:5000, ab40778, Abcam); anti-Abca1 (1:1000, 96292, Cell Signaling); anti-Abcg1 (1:1000, ab52617, Abcam); anti-Lrp8 (1:500, PA1-16913, Invitrogen), anti-Scarf1 (1:1000, PA5-115870, Invitrogen); anti-βcat (1:2000, 610154, BD Biosciences); anti-GAPDH (1:2000, sc-365062, Santa Cruz) and anti-β-actin (1:2000, 4970S, Cell Signaling). Secondary anti-rabbit (1:20,000, 5220-0480, Seracare) or anti-mouse (1:20,000, 5450-0011, Seracare) conjugated to horseradish peroxidase was used to detect primary antibodies. Signal was detected using ECL substrate (Bio-Rad, USA), imaged with the Bio-Rad Chemidoc System, and processed with the ImageLab Touch 2.3.0.07 software. Protein bands were quantified using Image Lab 6.0 (Bio-Rad, USA).

### Drosophila stocks and husbandry
The following strains were obtained from the Bloomington Drosophila Stock Center: w1118, GMRGl4 (#1104), 6xTCF-Helper-LacZ (#68167), and TubGal80ts (#7017). UAS-dTCF^RNAi (#3014) was obtained from the VDRC Stock Center. esg-Gal4 was kindly provided by Dr. Shigeo Hayashi. UAS-LRP6-GFP, UAS-LRP6-mCh, UAS-Fzd7-GFP, and UAS-Fzd7-mCh flies were generated for this study. All flies were backcrossed ×10 into the w1118 background, with continued backcrossing every 6–8 months to maintain isogenecity. All flies were reared on standard yeast- and cornmeal-based diet at 25 °C and 65% humidity on a 12 h light/dark cycle, unless otherwise indicated. The standard lab diet (cornmeal-based) consisted of 14 g Agar/165.4 g Malt Extract/41.4 g Dry yeast/78.2 g Cornmeal/4.7 mL propionic acid/3 g Methyl 4-Hydroxybenzoate/1.5 L water. Drosophila diet containing cholesterol was formulated by adding cholesterol to a potato (cholesterol-depleted) based diet as follows. Mashed potato flake (50 g) was mixed with 200 mL of preservative solution (1 L water, 5 mL of 12% Nipagen in ethanol, and 5 mL of propionic acid). Cholesterol (Sigma) was mixed with a preservative solution to produce a final concentration of 0.3% (m/v), 0.03%, and 0.003%, immediately prior to mixing with mashed potato flake. No cholesterol was added to the 0.0% cholesterol diet.

### Generation of transgenic Drosophila
UAS-LRP6-GFP, UAS-LRP6-mCh, UAS-Fzd7-GFP and UAS-Fzd7-mCh flies were generated by PCR amplification of the human cDNA sequences of LRP6 or Fzd7, tagged (3′) with GFP or mCherry from the respective cDNA plasmids with specific primers, and subsequently

cloned into the pUASt-attp plasmid (DGRC, #1419). The pUASt-LRP6-GFP and pUASt-Fzd7-GFP plasmids were injected into w1118; attp40 embryos and the plasmids pUASt-LRP6-mCh and pUASt-Fzd7-mCh were injected into w1118; attp2 embryos with a phiC31 integrase helper plasmid (Rainbow Transgenic Flies).

### Conditional expression of UAS-linked transgenes in Drosophila
For fluorescence resonance energy transfer (FRET) experiments, the esgGal4 driver was combined with a ubiquitously expressed temperature-sensitive Gal80 inhibitor (tubGal80ts). Crosses and flies were kept at 18 °C (permissive temperature), and 5-day-old females were fed the experimental cholesterol diets for 5 days, and then shifted to 29 °C for 2 days to allow for expression of the transgenes before analysis. For experiments utilizing the transgenic Wnt reporter, crosses and flies were kept at 18 °C (permissive temperature) and 5-day-old females were fed the experimental cholesterol diets for 2 days, and subsequently shifted to 29 °C for 5 days to allow for expression of the transgenes before analysis.

### Drosophila filipin staining
Intact midguts were dissected and fixed at room temperature for 30 min in fixative solution (100 mM glutamic acid, 25 mM KCl, 20 mM MgSO$_4$, 4 mM sodium phosphate, 1 mM MgCl$_2$, and 4% formaldehyde). A solution of 50 μg/mL filipin (Sigma, SAE0087) in PBS was used to stain cholesterol (1 h at RT). The midguts were then incubated in a 50% glycerol/PBS solution at 4 °C for 2 h to reduce autofluorescence.

### Drosophila immunostaining and microscopy
Intact midguts were dissected and fixed at RT for 30 min in 100 mM glutamic acid, 25 mM KCl, 20 mM MgSO$_4$, 4 mM sodium phosphate, 1 mM MgCl$_2$, and 4% formaldehyde. All subsequent incubations were performed in PBS, 0.5% BSA, and 0.1% Triton X-100 at 4 °C. The following primary antibody was used: mouse anti-LacZ (40-1a; 1:100) from Developmental Studies Hybridoma Bank. Fluorescent secondary antibodies (1:500) were obtained from Jackson Immunoresearch (Donkey Alexa Fluor® 488 AffiniPure Donkey Anti-Mouse, 715-545-150; Cy™3 AffiniPure Donkey Anti-Mouse IgG, 715-165-150; Donkey anti-mouse Alexa Fluor 647, 715-605-150). Hoechst (1:1.000) was used to stain DNA. Confocal images were collected using a Nikon Eclipse Ti confocal system (utilizing a single focal plane) and processed using the NIS-Elements AR 5.41.02 software.

### Fluorescence lifetime imaging microscopy combined with fluorescence resonance energy transfer (FLIM-FRET)
For in cellulo FLIM experiments, previously transfected cultured colonocytes were washed with DPBS, fixed in 4% PFA and 0.2% glutaraldehyde for 15 min, and washed with DPBS (3x). To elucidate the effect of APC on plasma membrane protein and lipid organization, cells expressing EGFP-tagged protein alone or co-expressing both EGFP-tagged and mCherry-tagged proteins were excited at 488 nm and the EGFP fluorescence lifetime was measured using a Lambert Instrument FLIM unit attached to a wide field Nikon Eclipse microscope and processed using LI-FLIM 1.2.26 software. EGFP was sinusoidally excited by a modulating 3-Watt 497 nm light-emitting diode (LED) at 40 MHz under epi-illumination. A solution of fluorescein (1 μmol/L) was used as a lifetime reference standard. Cells were imaged with a 1.3 numerical aperture ×40 Plan-Fluor oil objective using an appropriate GFP filter set. The phase and modulation were determined from 12 phase settings using the manufacturer's software. This analysis resulted in an image where the fluorescence lifetime of EGFP was determined and assigned to each pixel. The color scale on each pixel represents the fluorescent lifetime, which equates to the level of interaction between the EGFP- and mCherry-tagged proteins. Lifetime (phase) values were pooled and averaged from regions of interest drawn on individual cells. Each experiment was replicated at least

3 times. Statistical analysis was performed using one-way ANOVA. The fluorescence lifetime imaging microscopy combined with fluorescence resonance energy transfer (FLIM-FRET) method is highly favored over intensity-based FRET measurements because fluorescence lifetime is an intrinsic property of the fluorescent molecule and is generally insensitive to weak signal, excitation source, and variations in the donor–acceptor ratio[153].

For *Drosophila* FLIM experiments, adult flies expressing EGFP/mCherry constructs in gut stem cells were dissected in DPBS and fixed 20 min in 4% PFA before mounting in Mowiol medium. To perform FLIM experiments, cells expressing EGFP-tagged protein alone or co-expressing both EGFP-tagged and mCherry-tagged proteins were excited at 488 nm and the EGFP fluorescence lifetime was measured using a Lambert Instrument FLIM unit attached to a wide field Nikon Eclipse microscope, as described previously.

The percentage of the apparent FRET efficiency ($E_{app}$) was calculated using the measured lifetimes of each donor–acceptor pair ($\tau_{DA}$) and the average lifetime of the donor-only ($\tau_D$) samples as previously described[154]:

$$E_{app} = \left(1 - \frac{\tau_{DA}}{\tau_D}\right) \times 100 \qquad (2)$$

The apparent distance between donor–acceptor pair ($r$) was calculated using our fluorescence lifetime data and the following equations[154]:

$$E_{app} = \frac{R_0^6}{r^6 + R_0^6} \times 100 \qquad (3)$$

$$\tau_{DA} = \tau_D \times \frac{R_0^6}{r^6 + R_0^6} \qquad (4)$$

where $R_0$ is the Förster radius where half of the energy between donor–acceptor pair is transferred ($R_0$ for the EGFP-mCherry pair = 5.4 nm[155]).

## Super-resolution stochastic optical reconstruction microscopy (STORM)

To investigate the effect of oncogenic APC on Wnt receptor cluster structure, mouse primary single colonocytes (~100,000 cells) were suspended in live cell imaging solution (LCIS) (Invitrogen, A14291DJ) and transferred to poly-D-lysine (Gibco, A3890401) coated 8-well coverglass slides (Cellvis, C8-1.5H-N) and allowed to attach for 30 min on ice. Following attachment, cells were fixed with 250 μL of ice-cold 4% PFA-PEM buffer for 15 min on ice. Following fixation, cells were washed with ice-cold DPBS (2x). To fluorescently label colonocytes for nanocluster analysis using STORM, colonocytes cells were blocked with 5% BSA-DPBS for 30 min at RT and, after aspirating solution, incubated with 125 μL of 10 μg/mL of primary rat IgG2B LRP6 (1:20, R and D Systems Cat# MAB2960, RRID:AB_2139440) or anti-LRP6-AF647 (1:20, R and D Systems, Cat# FAB1505R) or rat IgG2A Fzd7 (1:50, R and D Systems Cat# MAB1981-100, RRID:AB_2247464) or anti-Fzd7-AF647 (1:50, R and D Systems, Cat# FAM1981R) or anti-Dvl1-AF647 (1:20, Santa Cruz Biotechnology, Cat# sc-8025 AF647) antibody in 1% BSA-DPBS for 1 h at RT. In the case of primary non-fluorescent antibodies, cells were rinsed with 1% BSA-DPBS (2x), DPBS (2x), blocked with 5% donkey serum-DPBS for 30 min at RT and incubated with 250 μL of 5 μg/mL Donkey Anti-Rat CF568 (1:400, Biotium; 20092) secondary antibody in 1% donkey serum-DPBS for 1 h at RT. Following incubation with secondary or primary fluorescently-labeled antibodies, colonocytes were washed with 1% donkey serum-DPBS (2x), DPBS (2x), fixed with 4% PFA-PEM for 15 min at RT, and washed with DPBS (2x).

To prepare samples for STORM imaging, a cysteamine (MEA) and glucose oxidase (Glox) buffer was prepared as described previously[156]. Briefly, TN buffer (50 mM Tris (pH 8.0) and 10 mM NaCl), an oxygen scavenging system (0.5 mg/mL glucose oxidase (Sigma, G2133), 40 μg/mL catalase (Sigma, C40) and 10% (w/v) glucose) and 10 mM MEA (Sigma, 30070) were mixed. MEA was stored as a solid at 4 °C and prepared fresh as a 1 M stock solution in 0.25 N HCl. This stock solution was kept at 4 °C and used within 2 wks of preparation. STORM imaging and analysis was performed using a Nikon N-STORM 4.0 Super-Resolution System. Samples were imaged at RT by direct STORM on an inverted N- STORM 4.0 Super-Resolution System (Nikon Ti, Japan) equipped with an Apochromat TIRF ×100 oil immersion objective with an 1.49 numerical aperture and a quad-band pass dichroic filter (Nikon, 97335). Images were acquired in highly inclined and laminated optical sheet (HILO) microscopy mode and focal drift was prevented with hardware autofocusing (Nikon Perfect Focus System 4). CF568 was pushed into the dark state using the 561 nm (125 mW) laser line at maximum laser intensity. Drift correction was performed using NIS-Elements AR ProEM-nSTORM 5.41.02 software. Data were acquired with a field-of-view (FOV) 256 × 256 pixels (160 nm pixel size), at 8.5 ms frame rate for 10,000 frames with an EMCCD camera (Princeton Instruments, ProEM-HS 512BX3). The sparsity of single molecules per frame was controlled with 405 nm laser (100 mW) at 20–60% of total power. Single-molecule localization analysis was performed using NIS-Elements AR ProEM-nSTORM 5.41.02 software. The Maximum Width, which is the maximum possible width a spot of some intensity in the image to be identified as a molecule by STORM Analysis was set to 400 nm. To exclude the identification of two molecules in close proximity, fluorescent spots whose ratio of elongation in the X and Y direction larger than 1.3 were rejected as a single molecule. Lateral drift was compensated for using autocorrelation-based drift correction with NIS-Elements AR 5.41.02 software which corrects drift based on tracking the entire set of molecules.

For nanocluster measurements, nanoclusters were identified by Voronoi analysis using NIS-Elements AR 5.41.02 software (NIS-Elements, RRID:SCR_014329) which utilizes 3D Delaunay triangulation[157]. Briefly, Voronoi creates clusters from the molecules present in the current image with the specified maximum distance between the adjacent molecules and the specified minimum number of molecules in the emerging cluster. The maximum distance was set at 50 nm, and minimum number of molecules at 10. Masked images of the clusters (gray scale) were converted to binary using a custom-built macro and exported to ImageJ/Fiji to generate ROIs. Clusters size was examined using the analyze particle function, within a custom-built macro, to quantify clusters larger than 0.001 μm².

To quantify single molecule and cluster density parameters, single-molecule localization microscopy (SMLM) data were analyzed using MATLAB (9.13.0.2193358, MathWorks) and Clus-Doc[158], a software written in MATLAB for detection of clusters and extraction of clustering and single molecule information adapted to utilize Nikon image files. Briefly, raw fluorescence intensity Nikon image files were exported as a text file (.txt), generating tables containing the $x$–$y$ particle coordinates of each molecules detected during data acquisition. To retrieve information on single molecules and individual clusters, DBSCAN analysis[158,159] was used to identify individual clusters. The software detects clusters using a propagative method which links points belonging to the same cluster based on two parameters; the minimum number of neighbors $\varepsilon$ ($\varepsilon = 3$) in the radius $r$ ($r = 20$ nm).

## Quantification of Wnt signaling activation

The 3T3 mouse fibroblast Wnt reporter cell line LEADING LIGHT® (3T3 LL) was used to determine the effects of different plasma membrane modulators on Wnt signaling activation. Briefly, previously seeded 3T3 LL cells were treated with membrane altering agents, washed with DPBS to remove compounds, and incubated with complete DMEM.

After incubation, the luminescence intensity from luciferase was quantified in a BMG LABTECH CLARIOstar microplate reader and processed with the MARS 4.2 software by employing the Luciferase Reporter Assay System (Promega, E1500).

### LRP6 and Fzd7 siRNA knockdown and fluorescent labeling to analyze antibody specificity

For non-fluorescently labeled primary antibodies, previously seeded colonocytes (8-well coverglass slide) were washed with DPBS, and the appropriate delivery mix (Accell siRNA Delivery Media, Horizon Discovery, B-005000-500) containing Accell mouse Fzd7 (5 nmol; Horizon Discovery, E-041631-00-0005) or LRP6 (5 nmol; Horizon Discovery, E-040651-00-0005) or non-targeting (5 nmol; Horizon Discovery, D-001910-10-05) siRNA SMARTpool was added to each well and incubated at 33 °C with 5% $CO_2$ for 96 h. After 96 h, cells were incubated with 0.05% trypsin-EDTA (Gibco, 25300-054) for 5 min at 37 °C, mixed with 5% FBS-RPMI mouse complete media and transferred to a 1.7 mL tube. The cell solution was centrifuged at $300 \times g$ for 5 min using a benchtop microcentrifuge 5424 (Eppendorf, Germany). Following centrifugation, the supernatant was aspirated and the cell pellet resuspended in 1.25X PEM buffer (100 mM PIPES, 6.25 mM EGTA, 2.5 mM $MgCl_2$, pH 6.8). For fluorescently labeled primary antibodies, previously seeded HAP1 parental and KO cell lines were washed with DPBS, incubated with 0.05% trypsin-EDTA for 5 min at 37 °C, mixed with 5% FBS-IMDM complete media and transferred to a 1.7 mL tube. The cell solution was centrifuged at $300 \times g$ for 5 min using a benchtop microcentrifuge 5424 (Eppendorf, Germany). Following centrifugation, the supernatant was aspirated and the cell pellet resuspended in 1.25X PEM buffer (100 mM PIPES, 6.25 mM EGTA, 2.5 mM $MgCl_2$, pH 6.8). To fix cell samples, a solution of 4% PFA in PEM buffer was added and incubated with cells for 20 min on ice. Following cell fixation, the solution was centrifuged at $500 \times g$ for 5 min and the supernatant aspirated. To fluorescently label Wnt receptors, cells were washed with DPBS, blocked with 5% goat serum in DPBS for 1 h, washed, and incubated with LRP6 and Fzd7 primary antibodies overnight. After labeling with primary antibodies, cells were imaged using an Amnis FlowSight® flow cytometer or washed with DPBS, blocked with 5% goat serum in DPBS for 30 min at RT, washed with DPBS and incubated with goat anti-rat AF647 (Invitrogen, A21247) secondary antibody for 1 h at RT, washed with DPBS and imaged using an Amnis FlowSight® flow cytometer.

### Reporting summary

Further information on research design is available in the Nature Portfolio Reporting Summary linked to this article.

## Data availability

The RNAseq data from this publication have been deposited in the Gene Expression Omnibus (GEO) database under accession number GSE234522 . Source data are provided with this paper.

## Code availability

The following macros are available in The Chapkin Lab repository (https://github.com/chapkinlab/Mutant-APC-reshapes-plasma-membrane-Nat-Comms): Binary to Tif file converter for STORM analysis, Cluster size particle analysis, PM mask generator and filipin III analysis for microscopy images, Lif to Tif file converter for Di4 analysis, and RNA expression analysis. In addition, source codes for the underlying functions and graphical user interface (GUI) application *ClusDoc* are available at the authors' Git repository (https://github.com/PRNicovich/ClusDoC).

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

## Acknowledgements

We thank Dr. Brad Weeks for histological assessment of mouse tissue sections and Dr. Gus A. Wright for valuable insight associated with flow cytometry experimental design and data processing. This work was supported by the CPRIT Regional Center of Excellence in Cancer Research, the Allen Endowed Chair in Nutrition & Chronic Disease Prevention, Texas A&M AgriLife Institute for Advancing Health through Agriculture, and the National Institutes of Health (R35-CA197707, R.S.C.; RO1-CA244359, R.S.C.; P30-ES029067, R.S.C.; R37CA259363, J.R.). A. Erazo-Oliveras is a recipient of a Diversity Supplement Award (R35 CA197707-S1) and a National Academies of Sciences, Engineering, and Medicine Ford Foundation Postdoctoral Fellowship.

## Author contributions

A.E.-O., M.M.-V., M.M., V.T., J.K., J.R., and R.S.C. designed the studies and wrote the manuscript. A.E.-O., M.M.-V., M.M., V.T., J.M.R-R., E.K., R.C.W., M.L.S., X.W., K.K.L., J.S.G., and D.A.M. performed the experimental work.

## Competing interests

The authors declare no competing interests.
