## [Peer Review File · Nature Communications]

Mutant APC reshapes Wnt signaling plasma membrane nanodomains by altering cholesterol levels via oncogenic β -cateninREVIEWER COMMENTS

Reviewer #1 (Remarks to the Author):

Erazo-Oliveras et al. studied the distribution and function of cholesterol in CRC cells carrying APC mutations. They found that an elevated level of cholesterol is present in the plasma membrane of these cells, which promotes oncogenic Wnt- β -catenin signaling by facilitating the assembly of Wnt signalosomes in the plasma membrane. Overall, this is an interesting and technically excellent study that reports important new findings potentially linking cellular cholesterol to CRC pathogenesis. However, genetic inconsistency in experimental systems and potential flaws in data interpretation raise concerns about the physiological relevance and significance of their results and the validity of their conclusion. For the report to be further considered for publication in this journal, the following major points must be addressed.

1. The authors found that cholesterol levels at the plasma membrane are higher in oncogenic CRC cells and tissue samples than in healthy counterparts and that the high cholesterol levels are linked to disruption in expression of proteins involved in cellular cholesterol homeostasis. This is an important finding that lays the foundation for this study. Overall, the fact that consistent data on the elevated cholesterol level were obtained from a wide range of cells, ranging from cultured cells to colonic crypts to CSCs, can be viewed as a main strength of this part of work. As elaborated by the authors, >90% of APC mutation in CRC results in double-allele APC truncation. And emerging data suggests that the truncated APC may have both loss-of-function and gain-of-function properties, enabling its unique oncogenic activity. Thus, one should employ CRC cell and tissue models harboring diverse APC truncations to obtain results of great physiological relevance and significance. Total deletion and point mutations of APC or double mutation of APC and β -catenin are very rare in CRC. Without any rationalization or justification, the authors used various cell types harboring many mutations except the most commonly observed double allelic APC truncations. For instance, they used cells derived from a heterozygous deletion mutant and a double mutant of APC and β -catenin in their gene dose experiments. They also used a mouse cell model with an APC point mutation (see below). It is very likely that APC deletion, truncation and point mutation exert completely different effects on the plasma membrane cholesterol and downstream signaling. Also, it is expected that β -catenin mutation has distinctively different effect on the plasma membrane cholesterol. Thus alternate and mixed use of a APC truncation mutant and a point mutation (as well as an APC-truncation- β -catenin double mutant) may yield very complex results that cannot be interpreted in a straightforward manner. It also seems premature to reach any conclusion about cholesterol-CRC connection by comparing *Apc* $-/+ \beta$ cat $-/+$ and *Apc* $-/+$ in their gene-dose experiment. Physiologically more relevant comparison of *Apc* $-/-$ and *Apc* $+/+$ is absent and that of *Apc* $-/+$ and *Apc* $+/+$ did not reveal major differences in Fig 1E. Performing *in vivo* imaging using colonic crypts and Lgr5+ CSCs to verify the imaging data from cell models is an excellent approach. Again, however, the authors used the crypts and CSCs from mice harboring a point mutation of APC (D580S) instead of APC truncation or deletion. This makes it difficult to directly compare *in vitro* data and *in vivo* data. The authors did compare the healthy and cancerous colonic tissue samples (from

CRC patients very likely carrying APC truncations), which could have produced convincing data, but unfortunately, they only performed comparative transcriptome analysis, not cholesterol quantification for these samples, weakening their argument. They should use cell and tissue models with physiologically more relevant and consistent genetic backgrounds (APC truncations). They should also perform cholesterol measurements for the patient-derived healthy and cancerous colon tissues.

2. Then authors investigated the functional consequences of cholesterol enrichment in the plasma membrane. Specifically, they systematically measured the effects of escalated plasma membrane cholesterol levels on membrane rigidity and molecular interactions (lipid-protein interaction and protein-protein interactions) at the plasma membrane by fluorescence-based quantitative methods using cell models harboring wild type and mutant APC. Overall, these comprehensive quantitative analyses on molecular interactions are excellent approaches that yielded an impressive array of biophysical data. Beside the problem associated with the APC mutations used in this study (see above), there is a major concern about the focus of this part of investigation. It has been reported that intermediate level of constitutive activation of Wnt signaling is achieved in oncogenic APC CRC, which is important for survival of these CRC cells. Thus, it is important to systematically compare the constitutive Wnt signaling activities in unstimulated CRC cells carrying oncogenic vs. wild type APC. However, this study mainly focused on comparison of b-catenin signaling activity in unstimulated vs. Wnt-stimulated oncogenic APC CRC cells. This approach led the authors to conclude that APC-truncated CRC cells have amplified Wnt-mediated b-catenin signaling compared to APC-WT cells. This may be true but it ignores other studies reporting that APC-truncated CRC cells have elevated constitutive b-catenin signaling activity. Even when the data in unstimulated APC-WT vs APC-mutated cells & tissues are presented, they are described in a relatively casual and qualitative manner without clear and consistent trends that the data fall short of providing key mechanistic insight into how APC mutation affects various steps in the Wnt-b-catenin signaling to allow these cells to gain survival advantages over APC-WT cells. Finally, the mechanistic link between the increased membrane rigidity and signalosome formation by APC mutations was not clearly described.

3. There are also technical concerns about the cholesterol quantification systems. The authors did not compare different methods and systems used in the study and rationalize their selective use in various experiments, making it difficult to directly compare data from different experiments. For example, they used the GPMV system for cholesterol estimation for some, but not all, parts of the study. The GPMV system serves as an excellent model membrane for membrane domain studies but may not be an ideal system for cholesterol quantification due to its well-documented limitation, i.e., a loss of transbilayer asymmetry. The author should explain why the system was used over the unperturbed plasma membrane. Similarly, although filipin has been extensively used to track intracellular cholesterol especially for NPC research, it is known to have many limitations because of its membrane-disrupting activity, binding to GM1, as well as rapid photobleaching. Why was filipin used in some experiments and a PFO-D4-based probe (D4H) in others? Since GM1 may be differentially enriched in the plasma membranes of different cells and tissues, the authors should preclude the potential contribution of GM1 to their signals by using another cholesterol probe, such as a PFO-D4 domain construct, in all experiments. If this is not feasible, they should at least provide an explanation.

Reviewer #2 (Remarks to the Author):

The Wnt pathway is an ancient signaling pathway that plays critical roles in animal development and, when improperly regulated, has a major impact on many human diseases. In the case of the latter, the best studied is the role of Wnt signaling in sporadic colorectal cancer, where activating mutations in the pathway due to mutations in the tumor suppressor, APC, are present in the vast majority of cases (>80%).

The prevailing model of Wnt signaling proposes the formation of three large molecular weight complexes that are involved in regulating Wnt signal transduction, including the cytoplasmic β -catenin "destruction complex," the nuclear "enhanceosome," and the plasma membrane "signalosome." In the case of the signalosome, which has been proposed to consist of oligomerized Wnt receptor-ligand complexes (and the cytoplasmic protein, Dvl), the driving force for assembly of the complex remains unclear. A previous study provides evidence for the role of free cholesterol in promoting Wnt/ β -catenin signaling in *Xenopus* embryos and cultured mammalian cells (Sheng et al., 2014); the proposed mechanism is via facilitating membrane receptor assembly in a process involving, in part, cholesterol binding by Dvl.

In the current manuscript by Erazo-Oliveras et al., using colonic epithelial cells from mice with wild-type and truncated APC, the authors tested the link between oncogenic APC mutations and plasma membrane-free cholesterol homeostasis. Analyses were performed primarily through the use of filipin III (a fluorescent cholesterol binder), confocal microscopy, FRET, flow cytometry, and the *Drosophila* intestinal model. Differential gene expression studies indicate that genes involved in cholesterol homeostasis are upregulated in APC truncated mutants. The authors concluded that "oncogenic APC" increases membrane cholesterol by transcriptional regulation of genes involved in cholesterol homeostasis and thereby promoting assembly of Wnt receptors complexes and enhancing Wnt ligand signaling.

The novelty of the manuscript is the hypothesis that APC can regulate Wnt signaling by altering plasma membrane cholesterol homeostasis. The effect of APC truncation on altering free plasma cholesterol levels is consistent with their observations on increased plasma membrane fluidity, increased FRET efficiency, and increased "nanoassemblies" of Wnt receptor complexes.

General critiques:

Evidence is not provided that APC's role in altering plasma membrane cholesterol homeostasis is direct, as opposed to its already published role in activating Wnt receptor signalosome assembly and signaling. Similarly, these studies do not provide insight into how APC truncations (or, acutely, normal Wnt receptor activation) lead to a localized increase in plasma membrane cholesterol.

1) Other than differential expression, no analysis was performed to confirm that upregulated cholesterol genes in their screen are increased at the protein level and actually contributed to increased plasma cholesterol levels (via knockdown or knockout studies). This is a major weakness because this is the basis for the authors' model.

2) The title is somewhat misleading. Based on the proposed model, one would expect that the effect of APC on cholesterol homeostasis is not unique to APC mutations but would be observed in cells with other activating mutations in the Wnt pathway- something that the authors did not test.

3) Perhaps more interesting are acute changes in membrane cholesterol in response to Wnt ligand treatment. These changes (within 30 min) are unlikely to be due to a transcriptional response. In fact, many experiments demonstrated changes only in the presence of Wnt3a, which makes one question the significance of the transcriptional model.

Other comments:

1) Fig S1C – These are not isogenic lines. HCT116 contains an activating β -catenin mutation and, hence exhibits active Wnt signaling. Isoforms of HCT116 containing wild-type β -catenin and truncated APC have been reported and would be particularly useful for comparison.

2) Fig. 1A-C Why are free levels of cholesterol greater for the Apc $-/+$ compared to Apc $+/+$? Apc $-/+$ has not previously been shown to demonstrate increased Wnt signaling as assessed by increased β -catenin levels or activation of Wnt target genes, so why would these cells have elevated cholesterol?

3) What is the level of plasma membrane cholesterol in oncogenic β -catenin $-/+$ and $+/+$ cells? If elevated, wouldn't this provide evidence that Wnt signaling transcriptionally regulates genes in cholesterol homeostasis? One could also demonstrate this by treating WT cells with the GSK3 inhibitor, CHIR90221, which will stabilize β -catenin.

4) Fig. 2E. What is the evidence that increased colonic/cecal weight is due to the proliferation of the epithelium? Some histology would be useful.

5) Does the increased plasma membrane cholesterol in APC mutants (and upon Wnt3a treatment) only affect Wnt receptors? If the increased plasma cholesterol is localized to plasma membrane domains containing Wnt receptors, it should not affect other signaling pathways. This is important to test.

6) Fig 6D. Quantification of signal intensity in ISCs is necessary. Scale bar is marked as 50um in the figure, but 500um in the figure legend.

7) Fig 6E-G. What is the level of overexpression of the human Wnt receptors by comparison to the endogenous Drosophila Wnt receptors? The overexpression approach would likely lead to artifactual conclusions regarding receptor multimerization. Since this is a Drosophila model, why test human receptors and not Drosophila Wnt receptors?

8) Fig 6I. This panel is not referred to in the text.

-Quantitation of Wnt reporter activation in the different conditions is necessary.

-Controls that show the reporter is specific for Wnt target gene activation in the intestine are necessary.

-Does the expression of the Wnt reporter in the intestine increase in wild-type flies that are fed cholesterol?

-Does overexpression of human LRP6 result in an increase in Wnt reporter expression by comparison to the wild-type control?

-Does overexpression of human Fzd7 increase Wnt reporter expression by comparison to the wild-type control?

9) Fig S8. The phenotype shown for overexpression of LRP6-GFP in the retina is not specific for overactivation of Wnt signaling, but instead could result from any number of perturbations that increase apoptosis or disrupt retinal differentiation. Loss of APC results in the apoptosis of photoreceptors during mid-pupation without a marked reduction in size of the retina, in contrast with the LRP6 overexpression phenotype shown.

In summary, this is an interesting study (albeit the writing is a bit inaccessible) that further confirms the role of cholesterol in regulating Wnt receptor activation. Unfortunately, the authors have not rigorously demonstrated their central model that truncations in APC contribute to increased plasma cholesterol levels via a transcriptional mechanism and that these changes are specific for APC mutations.

Reviewer #3 (Remarks to the Author):

In the article called Orthogonal model analyses reveal a novel role of mutant APC in reshaping cholesterol-dependent Wnt nanocluster structure-function and feedforward amplification of oncogenic β -catenin by Erazo-Oliveras and colleagues, the authors describe a connection between oncogenic signaling of APC, plasma membrane organization and colorectal cancer.

The manuscript thoroughly assessed the role of cholesterol homeostasis in different CRC models, including cellular assays, mouse models to test cholesterol homeostasis, FRET assays for interaction, and an in vivo fly model. The paper is clearly written, the techniques are consistent across the different models, and the figures are clean and of high quality. The article claims to be the first to link free cholesterol in the plasma membrane, WNT signaling, and the mevalonate pathway in cancer.

The structure of the text could be improved, but given the relevance of the findings, I would consider it for publication if the major comments are addressed and the text is amended.

Major comments

General comment. The authors narrow the importance of cholesterol only to cancer signaling, while this is a fundamental and very important process in all pathways and across all life forms. A mention of the importance of cholesterol in other fields like neuroscience, immunology, and development would enrich the text. A specific benefit here would be putting the changed lipid organization in the context of changed signaling. The work from the Dotti and co-workers that describes insulin receptor activation upon cholesterol loss comes to mind, but others might also be considered. Likewise, for cancer, the change in lipid composition and synthesis and its effects on treatment could be worth including, putting the membrane at the heart of the discussion of multiple altered cellular responses.

Figure 1. Although the use of Filipin III staining is widely accepted to assess Chol content, the plasma membrane is a tricky organelle to be imaged. Thus, most of the signal coming from Filipin III is probably from intracellular organelles. The authors address this point by imaging instead GPMVs. However, disruption of the cell may perturb the distribution of the lipids itself. Alternatively, a fluorometric cholesterol assay, as done for the fly model, could also be applied for the cell model to more accurately reveal if the cholesterol is coming from the PM or from the internal organelles, being differentiated by the different timings of the assay.

Figure 3. The concept of lipid rafts has significantly evolved since its initial statement, with the scientific community even debating its sole existence. Nowadays, we would say that there is a conciliation of ideas which acknowledge the need of Chol molecules in determined regions of membranes as a support for the association of other proteins, therefore the "raft" idea, but with other mechanisms as well coordinating these associations, such as the tetraspanin web or the actin cytoskeleton. The main problem with the lipid raft hypothesis is that it is based on experiments prone to artifacts. Lipids are quite motile molecules and very susceptible to changes in temperature; thus, samples measured at room temperature or on ice will mostly have patches of clumped Chol, while the same probe at 37°C (physiological temp) will be very hard to sample given its high motility. In these lines, fixation with PFA

has been reported inefficient when it comes to lipids, needing the adding of at least 0.1% glutaraldehyde to decrease the movement. Also, a pre-fixation at 37°C would reveal Chol domain even more reliable.

Another comment regarding the D4H probe, since the beginning, it was established that the concept of lipid raft concerns only the outer leaflet of the plasma membrane. The inner leaflet has a completely different organization and cholesterol concentration (much less than the outer). The authors do mention in a later timepoint the differences in cholesterol levels in both leaflets, without correcting the initial statement of probing for "lipid rafts" using D4H in the cytosol for previous panels.

Figure 4. panel E, on the one hand, the legend mentions measuring PI(4,5)P₂, while the figure displays probing with D4H without binding to cholesterol. The figure legend then proceeds to mention a "EGFP-tagged...", no noun present. On the other hand, the text of the cited figure mentions a PLC ratio shown for Figure 4E. This needs to be fixed.

Figure 5. The use of secondary antibodies for STORM is known to induce artificial clusters. Therefore, to clearly assess cluster formation, a labeled primary antibody, nanobody or aptamer is recommended.

Minor comments

- Pg.5 first paragraph, the phrase "...while retaining the functionality of the plasma membrane" should be removed as this is not true. The cytoskeleton is one of the primordial subjects for plasma membrane organization, the moment it is disrupted; the functionality is gone.
- Figure 1, D the gene *Plcb1* is both in the unique and in the shared genes. If mistaken, the number of genes should also be adjusted.
- Table S1. The subcategories in both panels should follow the same order
- P values need to be displayed instead of meaningless letters
- The name *Apc*^{-/+} should be consistent throughout the manuscript, including legends.
- Figure S3. Legend of panel A concludes different findings than the cited text.
- There is no materials and methods describing FACS of Figure S3 panel B
- Figure 2. Panel G should be the final one as it presents a new CRC model, while H is derived from the mice as in previous panels.
- Figure 3. Panel G is not mentioned in the figure legend

Reviewer #1

Erazo-Oliveras et al. studied the distribution and function of cholesterol in CRC cells carrying APC mutations. They found that an elevated level of cholesterol is present in the plasma membrane of these cells, which promotes oncogenic Wnt- β -catenin signaling by facilitating the assembly of Wnt signalosomes in the plasma membrane. Overall, this is an interesting and technically excellent study that reports important new findings potentially linking cellular cholesterol to CRC pathogenesis. However, genetic inconsistency in experimental systems and potential flaws in data interpretation raise concerns about the physiological relevance and significance of their results and the validity of their conclusion. For the report to be further considered for publication in this journal, the following major points must be addressed.

*1. The authors found that cholesterol levels at the plasma membrane are higher in oncogenic CRC cells and tissue samples than in healthy counterparts and that the high cholesterol levels are linked to disruption in expression of proteins involved in cellular cholesterol homeostasis. This is an important finding that lays the foundation for this study. Overall, the fact that consistent data on the elevated cholesterol level were obtained from a wide range of cells, ranging from cultured cells to colonic crypts to CSCs, can be viewed as a main strength of this part of work. As elaborated by the authors, >90% of APC mutation in CRC results in double-allele APC truncation. And emerging data suggests that the truncated APC may have both loss-of-function and gain-of-function properties, enabling its unique oncogenic activity. Thus, one should employ CRC cell and tissue models harboring diverse APC truncations to obtain results of great physiological relevance and significance. Total deletion and point mutations of APC or double mutation of APC and β -catenin are very rare in CRC. Without any rationalization or justification, the authors used various cell types harboring many mutations except the most commonly observed double allelic APC truncations. For instance, they used cells derived from a heterozygous deletion mutant and a double mutant of APC and β -catenin in their gene dose experiments. They also used a mouse cell model with an APC point mutation (see below). It is very likely that APC deletion, truncation and point mutation exert completely different effects on the plasma membrane cholesterol and downstream signaling. Also, it is expected that β -catenin mutation has distinctively different effect on the plasma membrane cholesterol. Thus alternate and mixed use of a APC truncation mutant and a point mutation (as well as an APC-truncation- β -catenin double mutant) may yield very complex results that cannot be interpreted in a straightforward manner. It also seems premature to reach any conclusion about cholesterol-CRC connection by comparing *Apc* $^{-/+}$ β cat $^{-/+}$ and *Apc* $^{-/+}$ in their gene-dose experiment. Physiologically more relevant comparison of *Apc* $^{-/-}$ and *Apc* $^{+/+}$ is absent and that of *Apc* $^{-/+}$ and *Apc* $^{+/+}$ did not reveal major differences in Fig 1E. Performing *in vivo* imaging using colonic crypts and *Lgr5*⁺ CSCs to verify the imaging data from cell models is an excellent approach. Again, however, the authors used the crypts and CSCs from mice harboring a point mutation of APC (D580S) instead of APC truncation or deletion. This makes it difficult to directly compare *in vitro* data and *in vivo* data. The authors did compare the healthy and cancerous colonic tissue samples (from CRC patients very likely carrying APC truncations), which could have produced convincing data, but unfortunately, they only performed comparative transcriptome analysis, not cholesterol quantification for these samples, weakening their argument. They should use cell and tissue models with physiologically more relevant and consistent genetic backgrounds (APC truncations). They should also perform cholesterol measurements for the patient-derived healthy and cancerous colon tissues.*

To address the concerns related to the use of colorectal cancer (CRC) cell lines and tissue models harboring diverse APC truncations, we have performed free cholesterol and membrane order measurements on an expanded series of authentic human adenocarcinoma cell lines harboring truncating APC mutations¹, i.e., **HT-29 (853 and 1555)**, **SW480 (13338)** and **DLD-1 (1427)** (see revised **Figures 1, 2 and 4**). The indicated amino acid positions refer to the length of the various truncated forms of APC expressed in these CRC cell lines. Notably, SW480 and DLD1 each display a truncating mutation on one allele and both subsequently underwent loss of heterozygosity of their second allele, whereas HT29 cells carry a different truncating mutation per allele. Furthermore, we also performed cholesterol and membrane order experiments employing patient-derived

organoids (PDOs). More specifically, **Crispr-modified human colonic organoids** carrying a double APC truncating mutation² as well as **tumor derived human organoids from CRC patients** expressing a truncated APC were utilized (see revised **Figures 3 and 4**). Consistent with our previous data, we observed an increase in the levels of plasma membrane cholesterol and rigidity in all models expressing truncated APC when compared to their wildtype (WT) APC counterpart (see revised **Figures 2-4**).

We apologize for not clearly describing the mutant truncated APC models employed in our experiments. We believe that this particular concern raised by the reviewer stems from a misunderstanding regarding the actual nature of the APC protein expressed in colonocytes that were either grown in culture or derived from live mouse crypts. We have clarified this point in the manuscript as shown in revised **Figure 1**. As reviewer #1 correctly discussed, we employed cell lines expressing truncated APC (**850**). However, this mutation is not “uncommon” since a very similar truncation has been reported in a well-established human adenocarcinoma cell line, i.e., HT-29 (**853**) and across colorectal cancer datasets, including the Cancer Genome Atlas (**849 and 853**) (**Figure 1A**). Nonetheless, as described above, in point #1, we have expanded the orthogonal models employed, including a series of human CRC models displaying different APC truncations.

Moreover, reviewer #1 states that we utilized “a mouse cell model with an APC point mutation”. This statement is incorrect. The APC Δ 580S system is a well-established mouse model, which we cite in references 62 and 69. As described by Hinoi *et al.*, our mouse model carries an *Apc* allele in which *loxP* sites, flanking exon 14 (*Apc* Δ 580S), were targeted by Cre recombinase (Cre) following induction with tamoxifen. Cre-mediated recombination deletes exon 14 in the *Apc* gene sequence creating a sequence shift in the normal genome reading frame, thus creating a frameshift mutation in codon 580, which in turn produces a premature chain termination codon. This termination codon results in the expression of a truncated polypeptide that is 605 aa in length of which only the first 580 aa of the sequence correspond to the normal *Apc* protein. We believe that this reviewer would have had a more favorable opinion of our work if he/she had been aware of this fact. Therefore, both *in cellulo* and *in vivo* data as well as their comparisons described in our manuscript are relevant and credible. We have included an analysis of the proteins expressed in each model (**Figure 1C**) that were employed in this manuscript and added a scheme to **Figure 1** to more clearly drive home the notion that we employed mutant APC truncation models throughout our manuscript.

Additionally, we have included additional data describing differentially expressed genes across orthogonal (cell lines, mouse and human clinical CRC) models in order to contrast the effects of *Apc* mutation on cholesterol homeostasis. See revised **Figure 2** and **Supplementary Tables 1 and 2** for details.

Finally, as mentioned above, we have performed additional cholesterol, membrane order and FLIM-FRET measurements employing various CRC models expressing APC truncations, which are physiologically relevant. Refer to revised **Figures 2, 3, 4, S7A, S7C and S7E** for results.

2. Then authors investigated the functional consequences of cholesterol enrichment in the plasma membrane. Specifically, they systematically measured the effects of escalated plasma membrane cholesterol levels on membrane rigidity and molecular interactions (lipid-protein interaction and protein-protein interactions) at the plasma membrane by fluorescence-based quantitative methods using cell models harboring wild type and mutant APC. Overall, these comprehensive quantitative analyses on molecular interactions are excellent approaches that yielded an impressive array of biophysical data. Beside the problem associated with the APC mutations used in this study (see above), there is a major concern about the focus of this part of investigation. It has been reported that intermediate level of constitutive activation of Wnt singling is achieved in oncogenic APC CRC, which is important for survival of these CRC cells. Thus, it is important to systematically compare the constitutive Wnt signaling activities in unstimulated CRC cells carrying oncogenic vs. wild type APC. However, this study mainly focused on comparison of b-catenin signaling activity in unstimulated vs. Wnt-stimulated oncogenic APC CRC cells. This approach led the authors to conclude that APC-truncated CRC cells have amplified Wnt-mediated b-catenin signaling compared to APC-WT cells. This may be true but it ignores other studies reporting that APC-truncated CRC cells have elevated constitute b-catenin signaling activity. Even when the data in unstimulated APC-WT vs APC-mutated cells & tissues are presented, they are described in a relatively casual and qualitative manner without clear and consistent trends that the data fall short of providing key mechanistic insight into how APC mutation affects various steps in the Wnt-b-catenin signaling to allow

these cells to gain survival advantages over APC-WT cells. Finally, the mechanistic link between the increased membrane rigidity and signalosome formation by APC mutations was not clearly described.

To address concerns related to the APC truncations used in our study and the focus of our lipid-lipid, protein-protein and lipid-protein interaction experiments, we performed cholesterol clustering (lipid-lipid interactions), Wnt receptor heteroclustering (protein-protein) and cholesterol-Wnt receptor (lipid-protein interactions) interaction experiments, i.e., FLIM-FRET, employing unstimulated CRC cells expressing oncogenic truncated APC or wild type (WT) APC (see revised **Figure S7**). This was complemented by a systematic analysis of our results in the manuscript text indicating that multiple plasma membrane-associated interactions between Wnt pathway receptors and their effectors were enhanced in the presence of oncogenic APC on a “unstimulated” background when compared to WT APC.

Furthermore, we have more clearly described our cholesterol clustering (lipid-lipid interactions), cholesterol-Wnt receptor (lipid-protein interactions), Wnt receptor heteroclustering (protein-protein interactions) and Wnt receptor-effector (upstream protein-downstream protein interactions) interaction experiments. Specifically, we have highlighted the major trends and insights that support how truncated APC affects various processes involved in Wnt pathway-associated membrane signaling and downstream activation of β -catenin, which allow cells expressing oncogenic APC to gain survival advantages over WT APC-expressing cells. See point #4 for more details.

Additionally, an expanded description of the link between oncogenic APC-mediated increased Wnt receptor heteroclustering and enhanced membrane rigidity has been provided in the revised manuscript text.

3. There are also technical concerns about the cholesterol quantification systems. The authors did not compare different methods and systems used in the study and rationalize their selective use in various experiments, making it difficult to directly compare data from different experiments. For example, they used the GPMV system for cholesterol estimation for some, but not all, parts of the study. The GPMV system serves as an excellent model membrane for membrane domain studies but may not be an ideal system for cholesterol quantification due to its well-documented limitation, i.e., a loss of transbilayer asymmetry. The author should explain why the system was used over the unperturbed plasma membrane. Similarly, although filipin has been extensively used to track intracellular cholesterol especially for NPC research, it is known to have many limitations because of its membrane-disrupting activity, binding to GM1, as well as rapid photobleaching. Why was filipin used in some experiments and a PFO-D4-based probe (D4H) in others? Since GM1 may be differentially enriched in the plasma membranes of different cells and tissues, the authors should preclude the potential contribution of GM1 to their signals by using another cholesterol probe, such as a PFO-D4 domain construct, in all experiments. If this is not feasible, they should at least provide an explanation.

To address the technical concerns related to the quantification of cholesterol and the systems/assays employed to perform these experiments, we initially measured free cholesterol in giant plasma membrane vesicles (GPMVs) derived from mouse crypts using filipin III (**Figure 3G**). We subsequently measured plasma membrane rigidity employing the same system (**Figure 4C**). Consistent with our previous GPMV findings, oncogenic APC increased the level of plasma membrane cholesterol as well as plasma membrane rigidity. As suggested, in the revised manuscript, we have now utilized GPMVs to examine the effect of APC on plasma membrane cholesterol levels in all respective parts of our study in order to allow for better comparisons between experiments.

Additionally, reviewer #1 raised a valid concern regarding the use of GPMVs as a lipid bilayer system to quantify plasma membrane cholesterol. Specifically, this reviewer firstly points out that the loss of lipid transbilayer asymmetry is associated with GPMVs and thus, concludes that this perceived limitation decreases the significance/validity of quantifying cholesterol using this system. At the same time, the reviewer then suggests that “intact” membranes would provide a better system to perform these experiments when compared to the GPMV system. We respectfully disagree with the reviewer’s first concern regarding loss of lipid asymmetry. Specifically, we strongly believe that loss of lipid transbilayer asymmetry in this particular experimental scenario, does not interfere in any way with our ability to determine the level of total cholesterol in GPMVs. Particularly, since filipin III cannot discriminate between the bilayer outer and inner leaflet, it enabled us to report on the total

level of cholesterol localized on both leaflets in the plasma membrane, which was the main goal of this experiment. Furthermore, at no point did we attempt to link the reported amount of cholesterol molecules detected by filipin III to their exact location on the plasma membrane, e.g., exofacial vs cytofacial leaflet, ordered vs disordered domains. Finally, regarding the second comment, the reviewer failed to acknowledge some key experimental details associated with the use of an “unperturbed” membrane model. Specifically, details associated with our flow cytometry and fluorescence microscopy experiments, which allowed us to examine and quantify the effects of APC truncations on plasma membrane cholesterol in whole cultured colonocytes (**Figures 2A, 4H, 4I, 4K, S1 and S9B**), mouse crypts (**Figure 3C and 3F**) and sorted stem cells (**Figure 3H**).

Reviewer #1 conveyed an important question regarding the selective use of filipin III and D4H, a fluorescently-labeled free cholesterol probe, for specific experiments: “*Why was filipin used in some experiments and a PFO-D4-based probe (D4H) in others?*”. We apologize for not clearly describing the rationale behind our experimental design. In the case of filipin III experiments, this dye was employed since it allows for the rapid, efficient and relatively specific examination of total plasma membrane free cholesterol (inner + outer leaflet) (**Figures 2 and 3**). In addition, we utilized fluorescently-labeled D4H to investigate the effect of oncogenic APC specifically on the interactions between cholesterol and Wnt receptors as well as key lipid and protein effectors (**Figures 4 and 5**). Due to our ability to recombinantly express this probe in the cytosol of cultured colonocytes and perform FLIM-FRET, we reported on these very close proximity interactions taking place at the inner leaflet of the plasma membrane while excluding the outer leaflet. Some of these interactions included Wnt3a-dependent changes on transbilayer asymmetry of plasma membrane cholesterol (TAPMC), inner leaflet cholesterol clustering and the interactions between inner leaflet cholesterol and Wnt receptors, among others. Based on the literature, inner leaflet cholesterol and its changes triggered by Wnt ligand stimulation allow cholesterol to serve as a signaling lipid, crucial for its modulation of cellular events associated with signal transduction and clustering during Wnt signaling. Thus, we did not arbitrarily choose to use different cholesterol probes for the analysis of cholesterol homeostasis, rather it was an experimentally driven strategic choice based on the specific needs of each study and the tested hypotheses.

In order to address the reviewer’s concern regarding various limitations associated with filipin III microscopy experiments, i.e., “*membrane-disrupting activity, binding to GM1, rapid photobleaching*”, we employed the Amplex™ Red cholesterol assay in order to corroborate our previous cholesterol analyses. Importantly, the findings obtained from these experiments showed that oncogenic APC drives an increase of free cholesterol in mouse cultured colonocytes (**Figure S1C**) and their derived GPMVs (**Figure S1D**), consistent with findings from our filipin III experiments. Notably, the Amplex™ Red cholesterol assay is based on an enzymatic reaction that targets the conversion of cholesterol and couples a reaction byproduct and Amplex™ Red to the stoichiometric conversion of the latter into a fluorescently-active molecule, i.e., resorufin, with high specificity, thus enabling the quantitative analysis of cholesterol levels without the inherent limitations mentioned above associated with filipin III.

Reviewer #2

The Wnt pathway is an ancient signaling pathway that plays critical roles in animal development and, when improperly regulated, has a major impact on many human diseases. In the case of the latter, the best studied is the role of Wnt signaling in sporadic colorectal cancer, where activating mutations in the pathway due to mutations in the tumor suppressor, APC, are present in the vast majority of cases (>80%).

*The prevailing model of Wnt signaling proposes the formation of three large molecular weight complexes that are involved in regulating Wnt signal transduction, including the cytoplasmic b-catenin “destruction complex,” the nuclear “enhanceosome,” and the plasma membrane “signalosome.” In the case of the signalosome, which has been proposed to consist of oligomerized Wnt receptor-ligand complexes (and the cytoplasmic protein, Dvl), the driving force for assembly of the complex remains unclear. A previous study provides evidence for the role of free cholesterol in promoting Wnt/ β -catenin signaling in *Xenopus* embryos and cultured mammalian cells (Sheng et al., 2014); the proposed mechanism is via facilitating membrane receptor assembly in a process involving, in part, cholesterol binding by Dvl.*

In the current manuscript by Erazo-Oliveras et al., using colonic epithelial cells from mice with wild-type and truncated APC, the authors tested the link between oncogenic APC mutations and plasma membrane-free cholesterol homeostasis. Analyses were performed primarily through the use of filipin III (a fluorescent cholesterol binder), confocal microscopy, FRET, flow cytometry, and the Drosophila intestinal model. Differential gene expression studies indicate that genes involved in cholesterol homeostasis are upregulated in APC truncated mutants. The authors concluded that "oncogenic APC" increases membrane cholesterol by transcriptional regulation of genes involved in cholesterol homeostasis and thereby promoting assembly of Wnt receptors complexes and enhancing Wnt ligand signaling.

The novelty of the manuscript is the hypothesis that APC can regulate Wnt signaling by altering plasma membrane cholesterol homeostasis. The effect of APC truncation on altering free plasma cholesterol levels is consistent with their observations on increased plasma membrane fluidity, increased FRET efficiency, and increased "nanoassemblies" of Wnt receptor complexes.

General critiques:

Evidence is not provided that APC's role in altering plasma membrane cholesterol homeostasis is direct, as opposed to its already published role in activating Wnt receptor signalosome assembly and signaling. Similarly, these studies do not provide insight into how APC truncations (or, acutely, normal Wnt receptor activation) lead to a localized increase in plasma membrane cholesterol.

1) Other than differential expression, no analysis was performed to confirm that upregulated cholesterol genes in their screen are increased at the protein level and actually contributed to increased plasma cholesterol levels (via knockdown or knockout studies). This is a major weakness because this is the basis for the authors' model.

To address this concern, we performed protein expression analysis targeting a select group of gene products displaying a key role in cholesterol homeostasis. We show that these proteins are dysregulated, consistent with the previous changes documented at the RNA level (**Figure 2Q**). Therefore, we conclusively show that these genes and their protein products contribute to the observed oncogenic APC-dependent dysregulation of cholesterol homeostasis.

2) The title is somewhat misleading. Based on the proposed model, one would expect that the effect of APC on cholesterol homeostasis is not unique to APC mutations but would be observed in cells with other activating mutations in the Wnt pathway- something that the authors did not test.

We believe that the aberrant activation of β -catenin by mutant APC, to a great extent, drives the documented loss of plasma membrane homeostasis described in this manuscript, including the dysregulation of cholesterol. The title explicitly reflects this notion and, most importantly, highlights the dependence of downstream aberrant Wnt signaling on oncogenic APC without alluding to other untested potential mechanisms pointed out by the reviewer. Furthermore, an investigation on the effects of other activating mutations in the Wnt pathway is clearly beyond the scope of this work. In an attempt to address the reviewer's concerns, we (i) slightly modified the title and (ii) have tested the effect of aberrant β -catenin activity on plasma membrane cholesterol and rigidity on a WT APC background. In order to perform these experiments, cultured colonocytes were treated with an inhibitor of GSK3 β , CHIR99021. This chemical compound has been shown to inactivate the β -catenin destruction complex thus leading to aberrant stabilization of β -catenin. Our data indicate that CHIR99021 incubation drives the increase of both plasma membrane cholesterol and rigidity (**Figure S11**).

3) Perhaps more interesting are acute changes in membrane cholesterol in response to Wnt ligand treatment. These changes (within 30 min) are unlikely to be due to a transcriptional response. In fact, many

experiments demonstrated changes only in the presence of Wnt3a, which makes one question the significance of the transcriptional model.

The changes associated with mutant APC in the presence of Wnt3a ligand in cultured colonocytes that were documented in our manuscript are indeed striking. The reviewer notes the membrane cholesterol changes observed as “*unlikely to be due to a transcriptional response*” due to the fact that they were shown to take place 30 min post Wnt3a ligand stimulation. This statement is somewhat arbitrary since it was not substantiated by any evidence. In fact, transcriptional responses following extracellular stimulus and subsequent activation of signaling events, including transcriptional changes, have been shown to occur within a broad time scale. While some of these transcriptional changes fall within a time scale of hours following stimulation, numerous transcriptional responses can be induced very rapidly within a few minutes³⁻⁷. Notably, it has been shown that Wnt3a ligand can activate β -catenin nuclear accumulation and the transcription of key Wnt pathway-related genes, e.g., cyclin D1, as early as 15 minutes⁸. Therefore, transcriptional responses within a 30-minute time frame driven by β -catenin following Wnt3a stimulation are, more than likely, achievable.

We would also like to further clarify that, based on our experimental design, not all membrane cholesterol changes documented in our manuscript were directly attributed to transcriptional changes mediated by aberrant β -catenin activity. In the case of the examination of the effects of truncated APC on the levels of plasma membrane free cholesterol (**Figures 2 and 3**), rigidity (**Figure 4**) and differential gene expression of cholesterol homeostasis-related genes (**Figure 2**), these studies were performed under dynamic equilibrium or steady state conditions without supplementation of exogenous Wnt ligand. Under these conditions, the time component does not influence the results observed and thus reflect the effects of aberrant transcriptional responses as a consequence of the mutation of APC. In the case of the cholesterol-associated experiments where changes were observed only in the presence of Wnt3a, which were fewer (2 cases) in comparison to changes that were observed both in the presence and absence of Wnt3a (4 cases), including changes of free cholesterol levels in the plasma membrane inner leaflet and raft domains, many of these changes can be easily explained to be the result of very rapid and dynamic processes dependent strictly on exogenous stimulus, which have been described previously in the literature⁹⁻¹². For example, under steady state conditions (absence of Wnt3a ligand) the levels of cellular plasma membrane free cholesterol in the cytofacial leaflet have been reported to be ~12-fold lower than that of the exofacial leaflet^{11,12}. This is known as the transbilayer asymmetry of PM cholesterol (TAPMC). However, an increase in inner leaflet free cholesterol has been shown to be triggered by Wnt3a in a dose and time-dependent manner^{11,12}. Moreover, this reported increase in inner leaflet cholesterol displays a quantitative correlation with the decrease in outer leaflet cholesterol suggesting the rapid redistribution of cholesterol between leaflets in response to Wnt3a. Notably, these changes have been directly linked to the initiation of other key Wnt signaling activities, including downstream recruitment of Dvl to the plasma membrane inner leaflet^{11,12}. The latter is involved in the formation of Wnt signaling nanoclusters. Therefore, it is clear that although very few phenotypes described in our manuscript show exclusively Wnt3a dependence (the majority were displayed in both the absence and presence of this ligand), the effect of transcriptional responses driven by aberrant β -catenin signaling can take place within the time scale of our experiments and, thus, are supported by our model. We have modified our putative model scheme to include these clarifications (**Figure 8**).

Other comments:

1) Fig S1C – These are not isogenic lines. HCT116 contains an activating β -catenin mutation and, hence exhibits active Wnt signaling. Isoforms of HCT116 containing wild-type β -catenin and truncated APC have been reported and would be particularly useful for comparison.

To address the concerns related to the use of “true” isogenic lines, we have examined the levels of plasma membrane free cholesterol and rigidity in numerous cancer cell line models and have compared them to HCT116 cells expressing WT APC and WT β -catenin (HCT116 Δ) (**Figures 2 and 4**)

2) *Fig. 1A-C Why are free levels of cholesterol greater for the Apc -/+ compared to Apc +/+? Apc -/+ has not previously been shown to demonstrate increased Wnt signaling as assessed by increased β -catenin levels or activation of Wnt target genes, so why would these cells have elevated cholesterol?*

We respectfully disagree with the reviewer's assessment regarding our data and point to previous studies in the literature where the effects of heterozygous expression of mutant APC on Wnt signaling and other phenotypes have been described¹³⁻¹⁶. For example, it has been recently demonstrated by gene ontology pathway analysis that the expression of Wnt signaling target genes is significantly enhanced in cells and organoids derived from human familial adenomatous polyposis (FAP) patients carrying different heterozygous APC truncation mutations (FAP1 and FAP2)¹⁶. These findings strongly suggest that the expression of Wnt target genes increase in an Apc-/+ background, directly contradicting the reviewer's comment. Interestingly, a different cell model (FAP3) derived from a patient carrying a single APC mutation, which led to the deletion of the Apc gene and, consequently, a complete lack of APC protein expression, did not recapitulate the effects observed in the FAP1 and FAP2 models highlighting the importance of truncated APC¹⁶. Moreover, organoids grown from these cells, which express heterozygous truncated APC, displayed a cyst-like growth pattern. This cyst-like morphology is consistent with previous findings showing that the growth pattern of specific organoids changed from branched to cyst-like structures as a result of increased Wnt signaling activity^{17,18}. Consistent with these observations, other phenotypes associated with this signaling pathway and cancer are altered in the heterozygous APC mutant background. For example, a single mutated APC allele was shown to enhance cell proliferation, increase stem cell number in crypts, increase crypt fission and upregulate neurogenic genes^{15,16}. The latter has been reported to occur in prostate, head and neck, pancreatic and colon cancer. Altogether, these data show that a single truncated APC can promote crypt progression to cancer, consistent with the notion that multiple cancers are initiated by APC mutation.

From a mechanistic dominant negative "gain of function" perspective, truncated mutant APC proteins dimerize with full-length WT APC in APC heterozygous cells, resulting in the functional inactivation of full-length WT APC proteins^{19,20}. It is important to highlight that the first 170 amino acids in the APC protein sequence are sufficient for its homodimerization in-vitro²¹. These points have been added to the revised Discussion.

3) *What is the level of plasma membrane cholesterol in oncogenic β -catenin -/+ and +/+ cells? If elevated, wouldn't this provide evidence that Wnt signaling transcriptionally regulates genes in cholesterol homeostasis? One could also demonstrate this by treating WT cells with the GSK3 inhibitor, CHIR99021, which will stabilize β -catenin.*

Please refer to our point addressing reviewer's #2 comment #2 and **Figure S11** in the manuscript for details. Briefly, we show that activation of β -catenin by the GSK3 inhibitor, CHIR99021, in cells expressing WT APC increased plasma membrane cholesterol and rigidity when compared to untreated WT APC cells.

4) *Fig. 2E. What is the evidence that increased colonic/cecal weight is due to the proliferation of the epithelium? Some histology would be useful.*

To address this comment, hematoxylin and eosin (H&E) stained sections were examined using light microscopy by a board-certified veterinary pathologist in a blinded manner. Histological lesions were categorized and scored accordingly. Mutant APC increased colonic crypt length (**Figure 3E**) as well as the degree of inflammation, tissue injury, proliferation and mucin staining (**Figure S4A**), as previously reported in the literature.

5) *Does the increased plasma membrane cholesterol in APC mutants (and upon Wnt3a treatment) only affect Wnt receptors? If the increased plasma cholesterol is localized to plasma membrane domains containing Wnt receptors, it should not affect other signaling pathways. This is important to test.*

We confined our experiments to Wnt receptors, because of the well-established role of membrane cholesterol in regulating their localization in lipid rafts and their dependence on these domains during Wnt receptor activation, clustering and signaling. This point has been restated in the revised manuscript. Furthermore, an investigation on the effects of loss of plasma membrane homeostasis on other signaling pathways, besides the Wnt pathway, is clearly beyond the scope of this work. Nonetheless, we understand the significance and potential impact of this question and thus it is the current focus of ongoing work in our laboratory.

6) Fig 6D. Quantification of signal intensity in ISCs is necessary. Scale bar is marked as 50um in the figure, but 500um in the figure legend.

Due to the nature of performing the filipin III stain *in vivo* in dissected midguts, we cannot concurrently image/label intestinal stem cell and filipin III. Thus, we cannot effectively/rigorously quantitate membrane filipin III signal intensity in intestinal stem cells (cell-specifically) vs other epithelial cell types. However, we believe the overall changes in cholesterol levels in various diets (**Figure 7C**), coupled with the uniformity of filipin staining across the monolayer midgut epithelium (including intestinal stem cells, **Figure 7D**) highlights that increasing cholesterol in the diet leads to an increase in membrane cholesterol in intestinal stem cells.

The legend, related to the scale bar, has been corrected.

7) Fig 6E-G. What is the level of overexpression of the human Wnt receptors by comparison to the endogenous *Drosophila* Wnt receptors? The overexpression approach would likely lead to artifactual conclusions regarding receptor multimerization. Since this is a *Drosophila* model, why test human receptors and not *Drosophila* Wnt receptors?

The strength of the fly model (besides being able to perform FLIM-FRET imaging *in vivo*) is, in part, the targeting of the tagged Wnt receptors specifically to intestinal stem cells (a cell type with a degree of sensitivity to Wnt activation). Unfortunately, this requires over-expression systems. Although we agree that any overexpression approach can lead to artifactual conclusions, this experimental approach is complementary to the cell-based FLIM-FRET experiments (**Figure 6A-6C**) that show similar results.

The reason to use *Drosophila* in this particular context is to in fact humanize the fly in order to potentially learn about mammalian disease mechanisms, as opposed to learning about endogenous signaling by utilizing insect receptors. Thus, the potential mechanisms identified by humanizing the fly must be corroborated with mammalian/human disease approaches. Due to differences between Wnt signaling across taxa, it is likely that some of these signaling mechanisms (i.e., receptor multimerization) will be different depending on which receptors are being used. Therefore, we are not arguing that multimerization of the receptor in response to cholesterol is a conserved process to control Wnt signaling activity, but instead that this approach provides an *in vivo* tool to confirm putative cancer signaling mechanisms. As an example, in order to address the conservation of this mechanism in insects, we generated APC Mutant Clones (dAPC1 and dAPC2) in the *Drosophila* midgut. In this context, we did not observe any difference in tumor formation in the presence or absence of cholesterol, which suggests that this mechanism may be limited to organisms that can synthesize cholesterol *de novo* (**Rebuttal Letter Figure 1**, see below)

8) Fig 6I. This panel is not referred to in the text.

The panel is now referenced in the text. We have also included quantification data for Wnt-reporter activation in intestinal stem cells using various cholesterol diets with or without LRP6 activation (**Figure 7I**).

-Quantitation of Wnt reporter activation in the different conditions is necessary.

We have also included quantification data for Wnt-reporter activation in intestinal stem cells using various cholesterol diets with or without LRP6 activation (**Figure 7I**).

-Controls that show the reporter is specific for Wnt target gene activation in the intestine are necessary.

Additional controls highlighting that the reporter is not active in wild type animals (without activation of LRP6) in various cholesterol diets are now included (**Figure 7I**). Additionally, this reporter system has previously been described in detail in the paper cited in the text (M. Chang et al., *Current Biology*, 2008). Regarding the specificity in *Drosophila* intestinal stem cells, this reporter was validated in the study referenced above (C. Wang & al., *Dev. Biol.*, 2012), highlighting that the TCF-LacZ reporter is activated specifically in APC1 and APC2 mutant intestinal stem cells clones.

-Does the expression of the Wnt reporter in the intestine increase in wild-type flies that are fed cholesterol?

As described above, additional controls highlighting that this reporter system is not active in wild type animals (without activation of LRP6) in various cholesterol diets are now included (**Figure 7I**).

-Does overexpression of human LRP6 result in an increase in Wnt reporter expression by comparison to the wild-type control?

In **Figure 7I** and **7J**, we show that overexpression of human LRP6 in *Drosophila* intestinal stem cells is not sufficient to induce Wnt reporter activation (since the reporter is turned OFF in cholesterol-free and standard diet conditions). However, in the presence of high cholesterol, LRP6 is able to activate the Wnt reporter in intestinal stem cells. In *Drosophila*, Arrow, the homologue of hLRP6, plays a minor role in the activation of Wnt signaling. This was shown *in vitro* using *Drosophila* S2 cells - overexpression of Arrow does not induce activation of Wnt reporters (L. Schweizer and H. Varmus, *BMC Cell Biology*, 2003). In this same study it was shown that overexpression of hLRP6 can activate Wnt signaling, similar to our findings in the developing retina and intestinal stem cells presented with high cholesterol.

-Does overexpression of human Fzd7 increase Wnt reporter expression by comparison to the wild-type control?

Based on various negative results, we found that human Fzd7 alone (unlike LRP6) is not able to activate Wnt signaling (**Rebuttal Letter Figure 2**, see below), as shown by the absence of the adult eye defect phenotype when Fzd7 expression was targeted to the larval eye imaginal disc by GMR-Gal4. In the contrary, expression of LRP6 resulted in an eye phenotype similar to a gain of function allele of wingless (Wg) and armadillo (Arm, encoding *Drosophila* β -catenin homolog) that mimic loss of APC^{22,23}. Since the receptors are overexpressed at similar levels, this may imply specificity in regulation of the pathway. In fact, individual receptors involved in hetero-receptor complexes in insects often show strong divergence in signal activation when overexpressed uniquely (although this is through overexpression and thus must be interpreted as such).

9) *Fig S8. The phenotype shown for overexpression of LRP6-GFP in the retina is not specific for overactivation of Wnt signaling, but instead could result from any number of perturbations that increase apoptosis or disrupt retinal differentiation. Loss of APC results in the apoptosis of photoreceptors during mid-pupation without a marked reduction in size of the retina, in contrast with the LRP6 overexpression phenotype shown. In summary, this is an interesting study (albeit the writing is a bit inaccessible) that further confirms the role of cholesterol in regulating Wnt receptor activation. Unfortunately, the authors have not rigorously demonstrated their central model that truncations in APC contribute to increased*

plasma cholesterol levels via a transcriptional mechanism and that these changes are specific for APC mutations.

We respectfully disagree with this comment. The eye phenotype shown in **Figure S10** is characteristic of Wnt activation. This phenotype was identified in *Drosophila* through a mutation called Glazed (Gla). Gla mutant adults are characterized by narrowed retina/eyes that are reduced in size, and the facets of these eyes coalesce into a gleaming, smooth sheet. The Gla mutation is an integration of a transposable element into the promoter region of *wg* gene (*Drosophila*'s Wnt) that leads to ectopic expression of Wg during retinal/eye development (E. Brunner et al., Dev. Biology, 1998).

We are not sure which study is being referred to by the reviewer regarding the loss of APC in the developing *Drosophila* eye. We believe the reviewer is referring to "The Adenomatous Polyposis Coli Tumor Suppressor and Wnt signaling in the Regulation of Apoptosis" from H. Benchabane and Y. Ahmed (Adv Exp Med Biol., 2011). Based on this study, the reviewer is correct regarding the fact that loss of APC1 does not induce a retina/eye phenotype. However, *Drosophila* activation of the Wnt pathway requires APC1 and APC2. Both APCs have an overlapping role which means silencing one is not enough to activate the Wnt pathway (C. Wang et al., Dev. Biol., 2012).

Reviewer #3:

In the article called Orthogonal model analyses reveal a novel role of mutant APC in reshaping cholesterol-dependent Wnt nanocluster structure-function and feedforward amplification of oncogenic β -catenin by Erazo-Oliveras and colleagues, the authors describe a connection between oncogenic signaling of APC, plasma membrane organization and colorectal cancer. The manuscript thoroughly assessed the role of cholesterol homeostasis in different CRC models, including cellular assays, mouse models to test cholesterol homeostasis, FRET assays for interaction, and an in vivo fly model. The paper is clearly written, the techniques are consistent across the different models, and the figures are clean and of high quality. The article claims to be the first to link free cholesterol in the plasma membrane, WNT signaling, and the mevalonate pathway in cancer. The structure of the text could be improved, but given the relevance of the findings, I would consider it for publication if the major comments are addressed and the text is amended.

Major comments

General comment. The authors narrow the importance of cholesterol only to cancer signaling, while this is a fundamental and very important process in all pathways and across all life forms. A mention of the importance of cholesterol in other fields like neuroscience, immunology, and development would enrich the text. A specific benefit here would be putting the changed lipid organization in the context of changed signaling. The work from the Dotti and co-workers that describes insulin receptor activation upon cholesterol loss comes to mind, but others might also be considered. Likewise, for cancer, the change in lipid composition and synthesis and its effects on treatment could be worth including, putting the membrane at the heart of the discussion of multiple altered cellular responses.

We have added several references highlighting the role of cholesterol in other processes, including that mentioned by the reviewer. Additionally, we have included a discussion regarding the role of cholesterol in cancer metastasis and its effects on cancer treatments.

Figure 1. Although the use of Filipin III staining is widely accepted to assess Chol content, the plasma membrane is a tricky organelle to be imaged. Thus, most of the signal coming from Filipin III is probably from intracellular organelles. The authors address this point by imaging instead GPMVs. However, disruption of the cell may perturb the distribution of the lipids itself. Alternatively, a fluorometric cholesterol assay, as done for the fly model, could also be applied for the cell model to more accurately reveal if the

cholesterol is coming from the PM or from the internal organelles, being differentiated by the different timings of the assay.

We respectfully disagree with this comment. The examination of plasma membrane cholesterol on whole cells was performed by imaging flow cytometry, which allows high-throughput quantification of cells while simultaneously generating single cell imaging data of each event recorded (**Figure S6**). Prior to quantifying filipin III cellular fluorescence, a series of cell masks are applied to every single recorded cell image, including a whole cell, plasma membrane (3 different thicknesses were applied) and cytosolic mask. This feature allows for the precise site-specific quantification of plasma membrane-associated fluorescence without interference from intracellular fluorescent signals. Moreover, with the use of specific controls, we confirmed that most of the filipin III signal originated from the plasma membrane. For example, incubation with M β CD, a cell impermeable cholesterol-depleting molecule that, due to its exogenous administration, can only extract cholesterol from the plasma membrane, decreased filipin III fluorescent signal by ~70-85% (**Figure 2B**). Nonetheless, in order to address the reviewer's concern and more robustly assess the changes in membrane cholesterol, we also examined cholesterol in various membrane models using the Amplex™ Red cholesterol assay (**Figure S1C** and **S1D**).

Figure 3. The concept of lipid rafts has significantly evolved since its initial statement, with the scientific community even debating its sole existence. Nowadays, we would say that there is a conciliation of ideas which acknowledge the need of Chol molecules in determined regions of membranes as a support for the association of other proteins, therefore the "raft" idea, but with other mechanisms as well coordinating these associations, such as the tetraspanin web or the actin cytoskeleton. The main problem with the lipid raft hypothesis is that it is based on experiments prone to artifacts. Lipids are quite motile molecules and very susceptible to changes in temperature; thus, samples measured at room temperature or on ice will mostly have patches of clumped Chol, while the same probe at 37°C (physiological temp) will be very hard to sample given its high motility. In these lines, fixation with PFA has been reported inefficient when it comes to lipids, needing the adding of at least 0.1% glutaraldehyde to decrease the movement. Also, a pre-fixation at 37°C would reveal Chol domain even more reliable.

To address this concern, several experiments associated with lipid, i.e., cholesterol membrane organization, and protein interactions were repeated using a more efficient fixation protocol that involves incubation of cells with 4% paraformaldehyde + 0.2% glutaraldehyde for 30 min at room temperature, as previously described in the literature²⁴. Consistent with our previous experiments using 4% PFA, these phenotypes displayed a dependence on mutant APC (**Figures 6H-6M** and **S7**).

Another comment regarding the D4H probe, since the beginning, it was established that the concept of lipid raft concerns only the outer leaflet of the plasma membrane. The inner leaflet has a completely different organization and cholesterol concentration (much less than the outer). The authors do mention in a later timepoint the differences in cholesterol levels in both leaflets, without correcting the initial statement of probing for "lipid rafts" using D4H in the cytosol for previous panels.

We respectfully disagree with the reviewer's comment that lipid rafts involve only the plasma membrane outer leaflet. The concept of interleaflet coupling suggests that despite the asymmetric composition between the outer and inner leaflets, coupling between the layers is surprisingly strong. This can be evidenced, for example, by recent experimental studies performed on phospholipid giant unilamellar vesicles showing that nanodomains formed in the outer layer are perfectly registered with those in the inner leaflet²⁵.

Figure 4. panel E, on the one hand, the legend mentions measuring PI(4,5)P2, while the figure displays probing with D4H without binding to cholesterol. The figure legend then proceeds to mention a "EGFP-tagged...", no noun present. On the other hand, the text of the cited figure mentions a PLC ratio shown for Figure4E. This needs to be fixed.

We have corrected the corresponding figure legend to address this concern.

Figure 5. The use of secondary antibodies for STORM is known to induce artificial clusters. Therefore, to clearly assess cluster formation, a labeled primary antibody, nanobody or aptamer is recommended.

In order to address this concern, we labeled single colonocytes derived from patient-derived organoids (PDOs) with a fluorescently-labeled primary antibody against LRP6 or Fzd7 (**Figure 6H-6M**). The new antibodies were characterized using HAP1 LRP6 or Fzd7-knockout cell lines (**Figure S8B** and **S8C**). Consequently, labeled colonocytes were imaged using STORM. We observed that mutant APC increased Wnt receptor cluster size and their frequency size distribution (**Figure 6J-6M**), consistent with our previous findings (**Figure S8D-S8J**).

Minor comments

- *Pg.5 first paragraph, the phrase "...while retaining the functionality of the plasma membrane" should be removed as this is not true. The cytoskeleton is one of the primordial subjects for plasma membrane organization, the moment it is disrupted; the functionality is gone.*

We have corrected the manuscript text to address this concern.

- *Figure1, D the gene Plcb1 is both in the unique and in the shared genes. If mistaken, the number of genes should also be adjusted.*

We have modified our illustrative depiction of these data (**Figure 2**) and thus corrected this typo.

- *Table S1. The subcategories in both panels should follow the same order*

The subcategories in both panels are arranged in alphabetical order. However, the cell lines described in each corresponding panel did not display exactly the same dysregulation of their genes and, consequently, did not display exactly the same dysregulation of their signaling pathways. As a consequence, the different cell lines do not share the same arrangement of subcategories and some pathways appear out of order or are simply absent (no significant change in RNA expression was observed).

- *P values need to be displayed instead of meaningless letters*

We respectfully disagree with this comment. Each letter, as indicated in each corresponding figure legend, indicate significant differences between treatment groups ($P < 0.05$) and thus are not meaningless. All of our graphs thoroughly display multiple comparisons between experimental conditions. This letter format not only allows us to show the presence or absence of significant changes but also the simultaneous comparison between any two specific conditions in a visually practical matter. Displaying each individual p-value would be otherwise impractical.

- *The name Apc-/+ should be consistent throughout the manuscript, including legends.*

We have changed our nomenclature as suggested in order to remain consistent throughout the entire manuscript.

- *Figure S3. Legend of panel A concludes different findings than the cited text.*

We are unsure as to which finding the reviewer is referring to. We show both in the text and figure that no changes in polyp formation were observed in heterozygous APC mice when compared to WT APC mice. It is possible that there was a misinterpretation regarding the wording displayed in the figure legend. Thus, we have rephrased the text to clarify any confusion.

- *There is no materials and methods describing FACS of Figure S3 panel B*

We apologize for this oversight. We have included all details pertaining to the examination of the effects of mutant APC on sorted stem cells.

- *Figure 2. Panel G should be the final one as it presents a new CRC model, while H is derived from the mice as in previous panels.*

We have modified the illustrative representation of this figure to describe the newly inserted data.

- *Figure 3. Panel G is not mentioned in the figure legend*

The figure legend text has been revised as suggested.

REVIEWER COMMENTS

Reviewer #1 (Remarks to the Author):

This reviewer expressed major concerns about 1) the lack of relevance and consistence of cell, tissue, and animal models, 2) the methods for cholesterol quantification, and 3) selection of model membrane systems. In the revised manuscript, Erazo-Oliveras et al. have satisfactorily addressed the technical and sample issues by additional measurements and clarification, which significantly improves the manuscript over the original one.

The authors have now clearly demonstrated that CRC cells and tissues with Apc truncation from diverse sources have higher cholesterol levels in their plasma membranes than Apc-WT cells, which is a major accomplishment. The main question is then how this elevated cholesterol is linked to high oncogenic activity of these CRC cells. The authors propose that elevated cholesterol in Apc-truncated CRC cells modulates Wnt receptor signaling nanocluster assemblies and their interactions with key lipid and protein effectors. This novel idea, which is in general consistent with their FLIM-FRET imaging data, also represents a major advancement in our understanding of signaling function of cellular cholesterol.

The real mechanistic question is how cholesterol achieves this feat and this critical mechanistic question still remains unanswered. The authors mainly resort to the cholesterol-mediated changes in membrane rigidity and lipid rafts. It is well documented that cholesterol modulates membrane rigidity but the authors should provide a more specific answer to the question as to how higher rigidity specifically alters molecular interactions in the Wnt signaling complex. Also, the authors attribute all cholesterol-mediated changes in molecular interactions in the Wnt signalosomes exclusively to changes in lipid rafts. This reviewer does not wish to revisit the “lipid raft controversy” but it should be noted that there are alternative mechanisms, such as the formation of biomolecular condensates (e.g., PMID: 30782412 and 29153704). Since the main focus of this work is elucidation of a new mechanism, however, the authors should perform more rigorous quantitative analysis of their data instead of vaguely linking all cholesterol-related phenomena to lipid rafts (and membrane rigidity). Finally, the question as to whether elevated cholesterol is important for constitutive or Wnt ligand-stimulated signaling activity still remains unclear. Collectively, I suggest the following revision to the authors to strengthen their arguments and improve the manuscript.

1. The authors have performed an impressive array of imaging measurements to provide quantitative information about the changes in cholesterol and molecular interactions in the plasma membrane. Yet they described correlations between cholesterol levels and molecular interactions mostly qualitatively and did not provide rigorous quantitative analysis, which renders their mechanistic interpretation less

credible and convincing. For example, they found that the increase in plasma membrane cholesterol in IMCE and IMCE-bcat cells led to the increase in co-localization of D4H and tH and explained this observation in terms of the changes in lipid raft domains. If they prefer to explain all their data in terms of lipid rafts, they should make additional efforts to establish that the lipid raft hypothesis provides the most plausible mechanism for their results. They could accomplish it by providing answers to the following simple questions. How can one quantify the change in lipid rafts? How is the cholesterol concentration quantitatively correlated with raft formation in their system? Is the observed increase in cholesterol concentration sufficient for significant change in lipid rafts? They can answer these questions with one or two sets of well-defined quantitative measurements using their sophisticated imaging methods.

2. The authors have provided more systematic data in response to my previous concerns about the differential effects of cholesterol on constitutive vs. ligand-stimulated Wnt signaling. Yet, this issue has not been fully resolved in the revised manuscript. For example, they found that cholesterol increased plasma membrane rigidity in CRC cultured cells (Fig 4B-4G) and many molecular interactions (e.g., Fig 4I and 4J and Fig 5) in the absence of Wnt3a but could detect a significant increase in D4H-tH FRET (Fig 4H) only in the presence of Wnt3a. Also, among those interactions that show cholesterol dependency in the absence of Wnt3a, some show a major difference between IMCE and YAMC whereas others exhibit the difference only between a more cholesterol-abundant IMCE bcat and YAMC. These results suggest that the effect of cholesterol on molecular interactions and clustering is controlled by a sophisticated mechanism, understanding of which would require more rigorous quantitative analysis. This point is also linked to the point #1, in that lipid raft hypothesis may not provide the most accurate answers to mechanistic questions.

Minor points:

1. In general, figures are difficult to understand because labels are often too small and of low resolution. Also, legends do not contain all necessary information about the labels and experimental conditions.

Reviewer #3 (Remarks to the Author):

The authors have responded to my criticisms and that of the other reviewers with further clarification and additional experimentation. However, I am not satisfied with several of their responses to my concerns, as outlined below.

1. To address my concern that upregulated cholesterol genes contribute to increased cholesterol, the authors performed immunoblots of a few proteins and showed observable changes in some of them. The authors stated that this result "conclusively shows that these genes and protein products contribute

to the observed oncogenic APC-dependent dysregulation of cholesterol homeostasis." To this reviewer, these noted changes are not dramatic. Importantly, knockdown studies, which were requested but not performed, are required to make this claim.

2. It seems odd that the authors state that testing other Wnt pathway activating (non-APC) CRC mutant lines (readily available from ATCC) is "beyond the scope of this work" as their model highlights altered cholesterol homeostasis (leading to Wnt receptor activation) is a specific feature of APC mutant cells.

3. The authors dismiss the possibility that the observed changes in cholesterol in response to Wnt3a are not due to gene transcription, given the rapid response (less than 30 min), because transcription of Wnt targets genes can be rapid. I believe the authors missed my point. The observable changes in cholesterol would require Wnt pathway activation, stabilization/nuclear localization of beta-catenin, transcription, translation, and alteration in regulator protein(s) activity to affect changes in free cholesterol. Thus, 30 minutes seem to me to be a particularly short frame for this type of response. Given that the authors strongly favor a transcription-mediated cholesterol homeostatic model, it will be essential to test that Wnt3a stimulation leads to increased cholesterol genes and proteins within the time frame (30 min) to demonstrate this convincingly.

4. In response to this reviewer's question as to why there is a greater level of free cholesterol in Apc -/+ compared to Apc +/+ cells, the authors cite studies showing that Apc -/+ cells exhibit Wnt activity. FAP is an inherited syndrome in which individuals have a heterozygous APC mutation. Transformation is initiated upon the acquisition of a second APC mutation and loss of function of the APC gene. Although it may be true that heterozygotes exhibit a certain level of Wnt signaling, possibly due to a dominant-negative mechanism (the mechanism itself remains murky and these cells are exquisitely sensitive to exogenous Wnt ligand stimulation), the authors fail to demonstrate that this is the actual situation in their studies. Thus, they should confirm that the heterozygous cells also show elevated Wnt target genes and changes in cholesterol regulatory genes.

5. The authors tested the GSK3 inhibitor, CHIR99021, on colonocytes and found that plasma membrane cholesterol levels rigidly increased (Fig S11). These results indicate that the observed changes in cholesterol are not specific to APC mutant cells or APC truncations but instead result from a more general response to Wnt signaling downstream of the destruction complex. As noted in the previous review, the manuscript as written attributes these effects specifically to APC truncations, which does not seem accurate and is therefore misleading.

6. If cholesterol staining precludes intestinal stem cell staining (Fig 7B,D), then the data shown do not support the conclusion that "cholesterol feeding led to an increase of filipin III staining in Drosophila ISCs" (page 16).

7. The TCF-LacZ reporter should not be referred to as a “ubiquitous Wnt reporter” (page 17). Wingless signaling in the adult intestine, as revealed by expression of Wingless and the activation of Wingless target genes, is highest at all the intestinal compartment boundaries and also present as a gradient in regions contiguous to compartment boundaries in the *Drosophila* adult intestine during homeostasis, including the midgut-hindgut boundary and posterior midgut (Tian 2016 PLOS Genetics 12: e1005822; Tian 2019 PLOS Genetics 15:e1008111). If this TCF-LacZ reporter had accurately displayed Wingless signaling in the adult gut, then the reporter would have displayed activity at all these regions. Since the authors of this manuscript found that the TCF-LacZ reporter is not active in wild-type intestines, the reporter is not accurately reporting Wingless signaling in the adult gut—not even at the highest levels of Wingless signaling. Similarly, the cited paper (Wang 2012, Dev Biology) analyzed TCF-LacZ reporter expression in the adult gut and found no activity in the epithelium of the middle midgut or posterior midgut near the midgut-hindgut boundary, indicating a major problem with accuracy of the reporter. Equally concerning, the reporter was not active at the dorsal-ventral boundary of the wing disc, as reported in both cited papers (Wang 2012, Dev Biology; and Chang 2008, Current Biology). Wingless is expressed highly at the dorsal-ventral boundary and critical in this region for specifying cell fate, calling into question whether this TCF-LacZ reporter accurately reports Wingless pathway activation. Therefore, the conclusions based on this reporter need confirmation using an independent method.

8. The *esg-Gal4* driver is not specific for ISCs, but also drives expression in enteroblasts. There are numerous incorrect “stem-cell specific” statements based on data from this driver in the text.

9. Neither the single *Apc* mutant nor the double *Apc* mutant phenotypes resemble the glassy surface reported in Fig S10, and therefore the conclusion that overexpression of LRP6 results in “adult eye phenotypes... that mimic loss of APC” is not correct. The statements that “loss of APC1 does not induce a retina/eye phenotype” and “Both APCs have overlapping role which means that silencing one is not enough to activate the Wnt pathway” are also not correct for the retina. In the retina, inactivation of *Apc1* singly or *Apc2* singly results in retinal phenotypes, but neither are similar to the eyes shown in Fig S10. Inactivation of *Apc1* singly results in apoptosis of all photoreceptors during mid-pupation and expansion of dorsal rim area fates. There is little change in retinal size and the surface is not glassy (Ahmed, 1998 Cell 93:1171; Lin, 2004 Development 131:2409; Benchabane, 2008 Development 135:936). Inactivation of *Apc2* singly results in mild expansion of the dorsal rim area (Benchabane, 2008 Development 135:936) with no change in retinal size and no glassy surface. *Apc1* and *Apc2* double mutant clones result in cell fate misspecification in the larval eye disc (Ahmed, 2002 Development 129:1751), and again, the retina is not glassy.

10. The model presented is a bit confusing. On one hand, the authors propose that mutant, truncated APC drives the change in levels of free cholesterol and membrane rigidity to promote receptor signalosome formation. On the other hand, the authors suggests that the mechanism is simply due to beta-catenin-mediated Wnt transcriptional activity (as evidence by their CHIR99021 data and implying

that it is not APC-mutant specific). Either model would be interesting and significant. However, the authors never directly tested these two possibilities. More concerning, the authors never performed experiments that tie together aberrant Wnt signaling and cholesterol homeostasis at the molecular level. What are the key/critical components of the cholesterol machinery that is regulated by the Wnt pathway to alter plasma cholesterol and rigidity? How do mutants of APC selectively regulate this process? Without answers to these questions, this paper represents a primarily descriptive paper.

As an aside, the authors refer to Wnt nanoclusters, which I assume is another name for Wnt signalosome complexes originally coined by Christof Niehrs (Bilic et al., 2007). If this is the case, renaming these Wnt receptor complexes Wnt nanoclusters seems arbitrary.

Reviewer #4 (Remarks to the Author):

In the revised version of the article called Novel role of mutant APC in reshaping plasma membrane cholesterol-dependent Wnt nanocluster structure-function and feedforward amplification of oncogenic β -catenin by Alfredo Erazo-Oliveras and colleagues, the authors thoroughly assessed the role of cholesterol homeostasis in different CRC models, including cellular assays, mouse models to test cholesterol homeostasis, FRET assays for interaction, and an in vivo fly model. In the present manuscript, the authors have addressed several of my main concerns. They elevated the discussion and added references. The use of Amplex Red cholesterol assay for confirming the cholesterol content and the change in fixation protocol, as well as the labeled primary antibody for the STORM experiment, is an improvement. Likewise, some minor comments have been addressed.

The article reads well and is an advancement for the field.

RESPONSE TO REVIEWERS' COMMENTS

Reviewer #1 (Remarks to the Author):

This reviewer expressed major concerns about 1) the lack of relevance and consistence of cell, tissue, and animal models, 2) the methods for cholesterol quantification, and 3) selection of model membrane systems. In the revised manuscript, Erazo-Oliveras et al. have satisfactorily addressed the technical and sample issues by additional measurements and clarification, which significantly improves the manuscript over the original one.

The authors have now clearly demonstrated that CRC cells and tissues with Apc truncation from diverse sources have higher cholesterol levels in their plasma membranes than Apc-WT cells, which is a major accomplishment. The main question is then how this elevated cholesterol is linked to high oncogenic activity of these CRC cells. The authors propose that elevated cholesterol in Apc-truncated CRC cells modulates Wnt receptor signaling nanocluster assemblies and their interactions with key lipid and protein effectors. This novel idea, which is in general consistent with their FLIM-FRET imaging data, also represents a major advancement in our understanding of signaling function of cellular cholesterol.

The real mechanistic question is how cholesterol achieves this feat and this critical mechanistic question still remains unanswered. The authors mainly resort to the cholesterol-mediated changes in membrane rigidity and lipid rafts. It is well documented that cholesterol modulates membrane rigidity but the authors should provide a more specific answer to the question as to how higher rigidity specifically alters molecular interactions in the Wnt signaling complex. Also, the authors attribute all cholesterol-mediated changes in molecular interactions in the Wnt signalosomes exclusively to changes in lipid rafts. This reviewer does not wish to revisit the "lipid raft controversy" but it should be noted that there are alternative mechanisms, such as the formation of biomolecular condensates (e.g., PMID: 30782412-Schaefer et al and 29153704 Gammons et al). Since the main focus of this work is elucidation of a new mechanism, however, the authors should perform more rigorous quantitative analysis of their data instead of vaguely linking all cholesterol-related phenomena to lipid rafts (and membrane rigidity). Finally, the question as to whether elevated cholesterol is important for constitutive or Wnt ligand-stimulated signaling activity still remains unclear. Collectively, I suggest the following revision to the authors to strengthen their arguments and improve the manuscript.

1. The authors have performed an impressive array of imaging measurements to provide quantitative information about the changes in cholesterol and molecular interactions in the plasma membrane. Yet they described correlations between cholesterol levels and molecular interactions mostly qualitatively and did not provide rigorous quantitative analysis, which renders their mechanistic interpretation less credible and convincing. For example, they found that the increase in plasma membrane cholesterol in IMCE and IMCE-bcat cells led to the increase in co-localization of D4H and tH and explained this observation in terms of the changes in lipid raft domains. If they prefer to explain all their data in terms of lipid rafts, they should make additional efforts to establish that the lipid raft hypothesis provides the most plausible mechanism for their results.

Response: We have expanded our discussion on membrane associated lipid raft-like domains to include "biomolecular cytoplasmic condensates" to provide additional perspective on the regulation of Wnt-beta catenin signaling. The conceptionally analogous organizing thermodynamic principal in cytoplasmic compartmentalization involves biomolecular

condensates, which concentrate molecules in the absence of an encapsulating membrane (lipid bilayer). Similar to membrane lipid raft-like domain self-assembly (organization), molecules, e.g., proteins and RNA, can freely diffuse in and out of “condensates” in the cytosol and tether to membranes through weak, multivalent interactions to drive dynamic liquid-liquid phase separation and the formation of mesoscale (nanometers to microns) non-membrane bounded structures (PMID: 35411084; PMID: 34349250; PMID: 30782412). We have provided additional new data demonstrating that the organization of a key Wnt cytosolic effector, i.e., Dvl1, which forms part of the Wnt signalosome, is dysregulated by oncogenic truncated APC (new **Figure S12**). In addition, we have inserted new plasma membrane “phase separation” data associated with the biochemical and biophysical characteristics of liquid disordered (Ld) and ordered (Lo) domains (new **Figure 5L**). Finally, we have expanded on our quantitative mechanistic investigation of the effects of oncogenic truncated APC on the interactions between molecules forming Wnt proteolipid condensates (new **Figure S9**). See the revised Discussion for details.

They could accomplish it by providing answers to the following simple questions. How can one quantify the change in lipid rafts?

Response: As noted above, our lipid raft related analyses include D4H (marks cholesterol enriched domains) and tH (marks liquid ordered-lipid raft-like domains) FLIM-FRET based experiments (**Figure 5H,I,K**). The FLIM-FRET method is highly favored over intensity-based FRET measurements because fluorescent lifetime is an intrinsic property of the fluorescent molecule and is generally insensitive to weak signal, excitation source, and variations in the donor acceptor ratio (PMID: 33515553; PMID: 26476105). The percentage of the apparent FRET efficiency (E_{app}) was calculated using the measured fluorescence lifetimes of each donor-acceptor pair (τ_{DA}) and the average fluorescence lifetime of the donor only (τ_D) samples. FRET occurs in a distance-dependent manner, generally in the length scale of 1–10 nm and thus provides exquisite quantitative insight into the nanoscale properties of plasma membrane liquid ordered-lipid raft-like domains. Furthermore, as discussed above, we have included new robust quantitative data describing the effects of oncogenic truncated APC on the interactions between molecules forming Wnt proteolipid condensates. As part of our revisions, we included a quantitative examination of resulting changes in the organization, i.e., distance between molecules (r , FRET distance), displayed by Wnt proteolipid receptors and effectors (new **Figure S9**). In summary, these analyses provide a quantitative measure of inner leaflet lipid raft topology and stability (PMID: 18068112). This point has been expanded upon in the revised manuscript.

To extend our membrane order (**Figure 5A-G**) and FLIM-FRET (**Figure 5H-K**) findings, we have inserted new data (new **Figure 5L**) using a Ld fluorescent marker (FAST Dil), which targets to cholesterol independent membrane domains in giant plasma membrane vesicles (GPMVs) (PMID: 22555243). Using confocal microscopy and a temperature-controlled imaging chamber, our findings support our mechanistic model (see revised **Figure 10**) that mutant APC reshapes the nanoscale organization of sterol-dependent and -independent lipid raft-like membrane domains.

How is the cholesterol concentration quantitatively correlated with raft formation in their system?

Response: Based on previous reports (PMID: 29769200; PMID: 26476105), the increase in tH cluster size (STORM experiments) and FRET efficiency between the lipid raft (tH) and cholesterol (D4H) markers observed in colonocytes expressing mutant APC (**Figure 5H,I,K** and **Figure S10**) can be interpreted either as an increase in the size and/or stability of the lipid raft-

like membrane domains. In addition, (in new experiments) the increase in the thermodynamic parameter miscibility transition temperature or T_{misc} in phase separation experiments using GPMVs (new **Figure 5L**) further supports our interpretation that oncogenic truncated APC increases the stability of Lo domains. We have inserted new text to clarify this point (see revised Discussion). The use of these lipid raft markers in a series of FLIM-FRET experiments demonstrate a novel role of mutant APC in reshaping the topology of lipid raft like membrane domains in colonocytes.

Is the observed increase in cholesterol concentration sufficient for significant change in lipid rafts? They can answer these questions with one or two sets of well-defined quantitative measurements using their sophisticated imaging methods.

Response: Based on our quantitative data (revised **Figures 5** and **6**), we have linked oncogenic APC-driven increases in size and/or stability of plasma membrane liquid ordered-lipid raft-like domains (e.g., tH and DH4) with the enhanced interaction between free cholesterol and Wnt receptors, (e.g., LRP6, Fzd7), Wnt activators/effectors, (e.g., PIP₂, Dvl1), increased Wnt receptor clustering and raft localization (**Figure 7**), enhanced Wnt proteolipid condensate formation (revised **Figure 8** and new **Figure S12**) and Wnt signaling activation (**Figures 9** and **S13**).

2. The authors have provided more systematic data in response to my previous concerns about the differential effects of cholesterol on constitutive vs. ligand-stimulated Wnt signaling. Yet, this issue has not been fully resolved in the revised manuscript. For example, they found that cholesterol increased plasma membrane rigidity in CRC cultured cells (Fig 4B-4G) and many molecular interactions (e.g., Fig 4I and 4J and Fig 5) in the absence of Wnt3a but could detect a significant increase in D4H-tH FRET (Fig 4H) only in the presence of Wnt3a. Also, among those interactions that show cholesterol dependency in the absence of Wnt3a, some show a major difference between IMCE and YAMC whereas others exhibit the difference only between a more cholesterol-abundant IMCE bcat and YAMC. These results suggest that the effect of cholesterol on molecular interactions and clustering is controlled by a sophisticated mechanism, understanding of which would require more rigorous quantitative analysis. This point is also linked to the point #1, in that lipid raft hypothesis may not provide the most accurate answers to mechanistic questions.

Response: We believe that the striking increase in plasma membrane cholesterol and plasma membrane rigidity observed in mutant APC cells and their GMPVs (**Figures 2, 4** and revised **Figure 5A-G**) in the absence of stimulation with Wnt3a ligand, highlights the important role of constitutive activation of Wnt signaling in these CRC models when compared to “healthy” cells expressing WT APC.

We also found that multiple APC truncations of different amino acid lengths increased the interaction between LRP6 and cholesterol (**Figure S10C**), consistent with IMCE and IMCE β cat cells (**Figures 6B** and **S10D**). Relative to our plasma membrane free cholesterol and rigidity experiments, the enhancement in LRP6-cholesterol interactions in CRC cultured cells occurred without Wnt3a stimulation. This is yet another key piece of evidence indicating the relevance of constitutive activation of Wnt signaling in this CRC model when compared to normal cells. Based on the premise that Wnt receptor activation (Wnt-stimulated signaling condition) is required for CRC development even in the presence of mutant APC (PMID:31211991; PMID:15542433; PMID:19773752; PMID:15054482), our novel findings support the targeting of the plasma membrane in APC mutated cells as a therapeutic strategy to suppress CRC. We have modified the Discussion to highlight these points.

Minor points:

1. In general, figures are difficult to understand because labels are often too small and of low resolution. Also, legends do not contain all necessary information about the labels and experimental conditions.

Response: Although we submitted high resolution (300 dpi) images in accordance with journal guidelines, it appears that the editorial office generated a low-resolution PDF file for the reviewers. The final publication will contain high resolution images. To avoid the creation of excessively large figure legends, we inserted some of the methodology/experimental details into the Methods section.

Reviewer #3 (Remarks to the Author):

The authors have responded to my criticisms and that of the other reviewers with further clarification and additional experimentation. However, I am not satisfied with several of their responses to my concerns, as outlined below.

1. To address my concern that upregulated cholesterol genes contribute to increased cholesterol, the authors performed immunoblots of a few proteins and showed observable changes in some of them. The authors stated that this result "conclusively shows that these genes and protein products contribute to the observed oncogenic APC-dependent dysregulation of cholesterol homeostasis." To this reviewer, these noted changes are not dramatic. Importantly, knockdown studies, which were requested but not performed, are required to make this claim.

Response: To address this concern, we have amended text in the manuscript to emphasize that our bulk RNA transcriptomic and proteomic data (**Figure S1, Table S1 and Table S2**) suggest that upregulated cholesterol related genes contribute "in part" to the increase in cellular free cholesterol levels. In addition, we have included RNA Seq data derived from APC mutant mice carrying only one *Apc* mutant allele showing no change in the expression of cholesterol related genes (new **Figure S2**) when compared to double mutant *Apc* mice. Interestingly, the heterozygous *Apc* mouse showed only a modest increase in plasma membrane cholesterol that required extended time when compared to the homozygous *Apc* mouse (**Figure 4F**). Furthermore, we have examined other important cholesterol homeostasis regulatory genes that display significant changes in their expression in a mutant APC dose-dependent manner (new **Figure S3**). Finally, (in new experiments) we cultured APC mutant and WT cell lines under lipoprotein depleted conditions (eliminates the effects of exogenous cholesterol uptake) to assess the contribution of exogenous cholesterol uptake and de novo cholesterol metabolism related genes in terms of the physiological relevance of the changes observed in the expression of cholesterol homeostatic genes driven by oncogenic truncated APC (new **Figure 3**).

2. It seems odd that the authors state that testing other Wnt pathway activating (non-APC) CRC mutant lines (readily available from ATCC) is "beyond the scope of this work" as their model highlights altered cholesterol homeostasis (leading to Wnt receptor activation) is a specific feature of APC mutant cells.

Response: The reviewer's proposed experiment is potentially complicated by the contribution of multiple colon cancer related oncogenic driver genes linked to cholesterol homeostasis as well as the potential role of mutant APC gain of function and dominant-negative mechanisms. Thus, we opted to assess whether aberrant activation of β cat drives the documented loss of plasma membrane homeostasis, including the dysregulation of cholesterol by testing the effect of

aberrant activation in the context of a WT APC background. For this purpose, as explained in the results section, cultured YAMC colonocytes (Apc+/+) were treated with an inhibitor of GSK3 β , CHIR99021. This compound has been shown to inhibit the β -catenin destruction complex thus leading to aberrant stabilization of β cat. Our data indicate that CHIR99021 incubation increased both plasma membrane cholesterol and rigidity (**Figure S15**). These results demonstrate that the stabilization of β cat increases plasma membrane free cholesterol and rigidity.

3. The authors dismiss the possibility that the observed changes in cholesterol in response to Wnt3a are not due to gene transcription, given the rapid response (less than 30 min), because transcription of Wnt targets genes can be rapid. I believe the authors missed my point. The observable changes in cholesterol would require Wnt pathway activation, stabilization/nuclear localization of beta-catenin, transcription, translation, and alteration in regulator protein(s) activity to affect changes in free cholesterol. Thus, 30 minutes seem to me to be a particularly short frame for this type of response. Given that the authors strongly favor a transcription-mediated cholesterol homeostatic model, it will be essential to test that Wnt3a stimulation leads to increased cholesterol genes and proteins within the time frame (30 min) to demonstrate this convincingly.

Response: We appreciate that other mechanisms may also contribute to intracellular cholesterol levels. Our findings suggest that mevalonate related pathways and other cholesterol homeostatic regulatory pathways are constitutively upregulated in part as a result of mutant APC status. See responses to point #2 of reviewer 1 and point #1 of reviewer 3 (above). The use of Wnt3a ligand was designed to further induce the transactivation and nanoclustering of the Wnt receptors.

4. In response to this reviewer's question as to why there is a greater level of free cholesterol in Apc -/+ compared to Apc +/+ cells, the authors cite studies showing that Apc -/+ cells exhibit Wnt activity. FAP is an inherited syndrome in which individuals have a heterozygous APC mutation. Transformation is initiated upon the acquisition of a second APC mutation and loss of function of the APC gene. Although it may be true that heterozygotes exhibit a certain level of Wnt signaling, possibly due to a dominant-negative mechanism (the mechanism itself remains murky and these cells are exquisitely sensitive to exogenous Wnt ligand stimulation), the authors fail to demonstrate that this is the actual situation in their studies. Thus, they should confirm that the heterozygous cells also show elevated Wnt target genes and changes in cholesterol regulatory genes.

Response: The mechanistic implications related to the effects of heterozygous APC mutation on free cholesterol status was not the focus of our study. Thus, we initially chose not to include these data. However, based on the concerns of the reviewer, we have included bulk RNA seq data obtained from Apc heterozygous cell lines and mice (see **Table S2A**, new **Figures S2** and **S3**).

5. The authors tested the GSK3 inhibitor, CHIR99021, on colonocytes and found that plasma membrane cholesterol levels rigidly increased (Fig S11). These results indicate that the observed changes in cholesterol are not specific to APC mutant cells or APC truncations but instead result from a more general response to Wnt signaling downstream of the destruction complex. As noted in the previous review, the manuscript as written attributes these effects specifically to APC truncations, which does not seem accurate and is therefore misleading.

Response: As noted in point #2 (Reviewer 3), our results (**Figure S13** and **S15**) suggest that mutant APC induced stabilization of β cat increases plasma membrane free cholesterol and rigidity. We have modified text in the manuscript to clarify this point.

6. If cholesterol staining precludes intestinal stem cell staining (Fig 7B,D), then the data shown do not support the conclusion that “cholesterol feeding led to an increase of filipin III staining in *Drosophila* ISCs” (page 16).

Response: We apologize for the confusion in the previous version. We used the word “preclude” to highlight that our experimental assays preclude the discrimination between intestinal stem cells membranes and enterocyte membranes within the midgut epithelium (we are analyzing all membranes in the midgut). However, this cholesterol staining doesn't preclude/eliminate intestinal stem cells. In fact, in the image provided in the manuscript, an intestinal stem is visible based on morphology. For clarity, we amended text, stating that we identified an “increase of Filipin III staining throughout the *Drosophila* midgut epithelium” to be more accurate.

7. The TCF-LacZ reporter should not be referred to as a “ubiquitous Wnt reporter” (page 17). Wingless signaling in the adult intestine, as revealed by expression of Wingless and the activation of Wingless target genes, is highest at all the intestinal compartment boundaries and also present as a gradient in regions contiguous to compartment boundaries in the *Drosophila* adult intestine during homeostasis, including the midgut-hindgut boundary and posterior midgut (Tian 2016 PLOS Genetics 12: e1005822; Tian 2019 PLOS Genetics 15:e1008111). If this TCF-LacZ reporter had accurately displayed Wingless signaling in the adult gut, then the reporter would have displayed activity at all these regions. Since the authors of this manuscript found that the TCF-LacZ reporter is not active in wild-type intestines, the reporter is not accurately reporting Wingless signaling in the adult gut—not even at the highest levels of Wingless signaling. Similarly, the cited paper (Wang 2012, Dev Biology) analyzed TCF-LacZ reporter expression in the adult gut and found no activity in the epithelium of the middle midgut or posterior midgut near the midgut-hindgut boundary, indicating a major problem with accuracy of the reporter. Equally concerning, the reporter was not active at the dorsal-ventral boundary of the wing disc, as reported in both cited papers (Wang 2012, Dev Biology; and Chang 2008, Current Biology). Wingless is expressed highly at the dorsal-ventral boundary and critical in this region for specifying cell fate, calling into question whether this TCF-LacZ reporter accurately reports Wingless pathway activation. Therefore, the conclusions based on this reporter need confirmation using an independent method.

Response: We apologize for the confusion in the previous version, as the TCF-LacZ reporter should not be referred to as a ‘ubiquitous’ reporter. For clarity, we amended the text, highlighting this as a ‘specific’ Wnt-activity reporter. We have also amended the text to clarify that we are only describing TCF-LacZ reporter activity in intestinal stem/progenitor cells, and not the midgut as a whole (discussed more below).

Many *Drosophila* Wnt activity reporters (and reporters in general) present limitations since many are randomly inserted into the *Drosophila* genome. Cell-specific changes in chromatin regulation (accessibility), as well as cell-specific changes in pathway activation, cause cellular diversity in reporter activity. This is why we often select a reporter based on the tissues/cells of interest. For example, in the Tian 2016 PLOS Genetics study, where the authors wanted to distinctly study compartment boundaries in the *Drosophila* midgut related to Wnt/Wg activity, they mainly used the Fz3-RFP reporter, which was shown to be active at the intestinal compartment boundaries. However, this same reporter could not be used to study intestinal

stem/progenitor cells because they show in that manuscript that reporter activity in the stem/progenitor cells is independent of classical Wnt/Wg pathway activation. Coupled with previous reports highlighting that TCF-LacZ is active in the intestinal stem/progenitor cells in the *Drosophila* midgut when Wnt/Wg activity is changed, that's why we utilized the TCF reporter over others. It's also important to note that the TCF-LacZ reporter is less sensitive to changes in Wnt/Wg activity in the midgut, again highlighting cellular diversity of these *in vivo* reporters.

Regarding the intestinal compartment boundaries, these anatomical structures are not part of our images, but we do visualize mild TCF reporter activity at these boundaries as expected (although not as strongly as the Fz3-RFP reporter).

Finally, we have provided additional data to highlight that the TCF transcription factor is downstream of hLRP6 activation (revised **Figure S14**, described in more detail in comment 9 below).

8. The *esg-Gal4* driver is not specific for ISCs, but also drives expression in enteroblasts. There are numerous incorrect "stem-cell specific" statements based on data from this driver in the text.

Response: This is correct, *esg-Gal4* drives expression in both progenitor cells (enteroblasts) and intestinal stem cells within the *Drosophila* midgut. For clarity, we amended the text, highlighting a more accurate description of the Gal4 genetic driver.

9. Neither the single *Apc* mutant nor the double *Apc* mutant phenotypes resemble the glassy surface reported in Fig S10, and therefore the conclusion that overexpression of LRP6 results in "adult eye phenotypes... that mimic loss of APC" is not correct. The statements that "loss of APC1 does not induce a retina/eye phenotype" and "Both APCs have overlapping role which means that silencing one is not enough to activate the Wnt pathway" are also not correct for the retina. In the retina, inactivation of *Apc1* singly or *Apc2* singly results in retinal phenotypes, but neither are similar to the eyes shown in Fig S10. Inactivation of *Apc1* singly results in apoptosis of all photoreceptors during mid-pupation and expansion of dorsal rim area fates. There is little change in retinal size and the surface is not glassy (Ahmed, 1998 Cell 93:1171; Lin, 2004 Development 131:2409; Benchabane, 2008 Development 135:936). Inactivation of *Apc2* singly results in mild expansion of the dorsal rim area (Benchabane, 2008 Development 135:936) with no change in retinal size and no glassy surface. *Apc1* and *Apc2* double mutant clones result in cell fate misspecification in the larval eye disc (Ahmed, 2002 Development 129:1751), and again, the retina is not glassy.

Response: We apologize for the confusion in the previous version, as using the phrase 'mimic loss of APC' was incorrect, since data describing 'eye phenotypes' related to ubiquitous loss of both APC1 and APC2 are currently lacking. The study by Ahmed 2002 in Development utilizes clonal analysis, a commonly employed genetic approach in *Drosophila* which induces a chimeric eye with cellular clones displaying both a double mutation for APC1 and APC2 and wild-type clones (cells), which can be used for a variety of analyses, but cannot be compared to an analysis with GMR-Gal4 or other development-related eye/retina genetic drivers. For clarity, we have amended text, highlighting a more accurate description of this genetic condition, focusing on Wnt activation and not loss of APC function.

Initially, we believed the major concern was that the gain of function hLRP6-induced 'eye phenotype' was based on apoptosis and not an activation of Wnt signaling. The 'glass eye phenotype' is more similar to Wnt activation. We also provided an additional experiment to strengthen our conclusion by showing that we can rescue the 'glass eye phenotype' induced by

hLRP6 activation by silencing the *Drosophila* Wnt transcription factor TCF (revised **Figure S14**), highlighting that this phenotype is likely through Wnt activation.

10. The model presented is a bit confusing. On one hand, the authors propose that mutant, truncated APC drives the change in levels of free cholesterol and membrane rigidity to promote receptor signalosome formation. On the other hand, the authors suggests that the mechanism is simply due to beta-catenin-mediated Wnt transcriptional activity (as evidence by their CHIR99021 data and implying that it is not APC-mutant specific). Either model would be interesting and significant. However, the authors never directly tested these two possibilities. More concerning, the authors never performed experiments that tie together aberrant Wnt signaling and cholesterol homeostasis at the molecular level. What are the key/critical components of the cholesterol machinery that is regulated by the Wnt pathway to alter plasma cholesterol and rigidity? How do mutants of APC selectively regulate this process? Without answers to these questions, this paper represents a primarily descriptive paper.

As an aside, the authors refer to Wnt nanoclusters, which I assume is another name for Wnt signalosome complexes originally coined by Christof Niehrs (Bilic et al., 2007). If this is the case, renaming these Wnt receptor complexes Wnt nanoclusters seems arbitrary.

Response: Our revised manuscript focuses on the functional effects of free cholesterol on a mutant APC background, which consequently leads to aberrant β cat transcriptional activity within the context of the plasma membrane. Firstly, APC truncating mutations and aberrant β cat transcriptional activity are two key molecular events of one important and extensively well documented mechanism shown to initiate intestinal tumorigenesis. Our highly mechanistic data, which leverages this established premise, describes what “key/critical components” of the cholesterol homeostatic machinery are dysregulated by mutant APC/aberrant β cat (**Figure 2F-Q, Figure S1E, new Figure S2 and new Figure S3**) and confirms their functional relevance at the cellular level (new **Figure 3 and Figure S15**). Secondly, we show the consequences of this altered cholesterol homeostatic machinery in terms of the biochemical (**Figure 2A-E, Figure 4 and Figure S1A-D**) and biophysical (**Figure 5A-G and new Figure 5L**) properties of the plasma membrane and the structure/organization of Wnt proteolipid condensate signaling platforms (**Figure 5H-K, Figure 6, Figure 7, revised Figure 8, new Figure S9, Figure S10, Figure S11 and new Figure S12**) at the molecular level. Most importantly, we confirm that cholesterol is sufficient to alter Wnt receptor clustering (**Figure 9E-G**) and downstream signaling (**Figure 9H-J and Figure S13A**). In attempt to be responsive to all the reviewer’s concerns, our highly mechanistic data were corroborated using multiple *in cellulo*, *ex vivo* and *in vivo* mutant APC CRC models and highly sophisticated instrumentation (see methods section for details). On the other hand, the question of how these “key/critical components” of the cholesterol machinery regulated by the Wnt pathway alter plasma cholesterol and rigidity is beyond the scope of our manuscript and must await a dedicated study in order to be answered meticulously.

We apologize for the potential confusion attributed to the use of the term Wnt nanoclusters in our manuscript. However, our decision to use this term was a calculated attempt to accurately describe the structures examined. As described by Christof Niehrs (cited by the reviewer), Wnt signalosomes are composed of a number of proteins and lipids, some of which we did not examine in our work. In addition, in the revised text, we have expanded on the importance of Wnt receptor nanoclustering as a key step during Wnt activation and downstream signaling. In most cases, the use of the term Wnt nanoclusters is more accurate and conceptually correct.

Reviewer #4 (Remarks to the Author):

In the revised version of the article called Novel role of mutant APC in reshaping plasma membrane cholesterol-dependent Wnt nanocluster structure-function and feedforward amplification of oncogenic β -catenin by Alfredo Erazo-Oliveras and colleagues, the authors thoroughly assessed the role of cholesterol homeostasis in different CRC models, including cellular assays, mouse models to test cholesterol homeostasis, FRET assays for interaction, and an in vivo fly model. In the present manuscript, the authors have addressed several of my main concerns. They elevated the discussion and added references. The use of Amplex Red cholesterol assay for confirming the cholesterol content and the change in fixation protocol, as well as the labeled primary antibody for the STORM experiment, is an improvement. Likewise, some minor comments have been addressed. The article reads well and is an advancement for the field.

Response: We thank the reviewer for recognizing the novelty and significance of our work.

REVIEWER COMMENTS

Reviewer #1 (Remarks to the Author):

A large majority of colorectal cancer (CRC) cells harbor variable Apc truncations, leading to hyperactivation of Wnt-beta catenin signaling. The key finding in this manuscript is that Apc truncations increase plasma membrane cholesterol, which then promotes molecular interactions in Wnt signalosomes. It is an interesting and potentially very important new discovery considering that it has been believed that truncated Apc causes Wnt-beta catenin signaling hyperactivation mainly because it cannot effectively support the steady-state beta catenin degradation. However, to establish biological significance and translational potential of this new finding, one must fully understand the mechanisms by which Apc truncations alter cholesterol homeostasis and Wnt signaling activity. Both cholesterol homeostasis and Wnt signaling are highly complex processes involving many protein and lipid components and thus full mechanistic elucidation of the Apc-cholesterol-Wnt signaling link would require thorough, systematic and quantitative studies.

In the past two reviews, my critiques focused on two questions: (1) How does Apc truncation alter cellular cholesterol homeostasis and the local cholesterol level at the plasma membrane and do variable Apc truncations have similar or different effects? (2) Are these cholesterol increases large enough to drive molecular interactions within Wnt signalosomes and, if so, how? Specifically, I raised concerns about the validity, credibility, and relevance of their model systems, the accuracy and sensitivity of their cholesterol quantification methods, and their data interpretation. Although the authors have made significant efforts to address these concerns technically, key mechanistic questions still remain answered.

1. As for the question regarding how Apc truncations alter cellular cholesterol homeostasis, the authors have collected RNAseq and western blot data primarily to identify cholesterol homeostasis proteins that are affected by the Apc truncation. They found that Apc truncation downregulated major players of cholesterol efflux, ABCA1 and ABCG1, and upregulated LRP8 and Scarf1 involved in cholesterol uptake, while also downregulating some proteins involved in de novo synthesis (Pmvk, Mvk and Mvd). Although these results are useful and consistent with the idea that Apc truncation disrupts cholesterol homeostasis, they do not explain how Apc truncation enriches cholesterol in the plasma membrane. As mentioned above, cholesterol homeostasis is a highly complex system that controls multiple processes, including de novo synthesis, storage, uptake, efflux, and transport. Also, it is not fully understood how subcellular local cholesterol levels (e.g., ER and the plasma membrane) are correlated. It is therefore unclear how all these altered proteins contribute to cholesterol enrichment in the plasma membrane. For example, how does downregulation of cholesterol biosynthetic proteins lead to cholesterol enrichment in the plasma membrane? I fully agree with the Reviewer 3 who recommended that the authors should perform systematic siRNA gene knockdown (and/or overexpression) to dissect the effects of altered proteins to the plasma membrane cholesterol enrichment, which for some reason the authors opted not to follow. Instead, they performed cell growth in the cholesterol-depleted media, the results of which do not explain how exactly cholesterol is enriched in the plasma membrane of Apc-truncated CRC cells.

2. In the previous review, I requested the authors to provide convincing evidence that the cholesterol enrichment in the plasma membrane is sufficient (i.e., above a defined threshold) for driving proposed molecular interactions and also to provide a clear mechanistic explanation for their results. As for the magnitude of cholesterol enrichment, although the authors describe it as “striking” throughout the manuscript, their data do not quantitatively support the claim. Specifically, filipin staining, which is their primary method, shows modest 20-50% increases in plasma membrane cholesterol caused by double-allele Apc truncation (Fig 2B, C, D). Ironically, much larger increases were detected by the supplementary Amplex Red method but the authors did not provide any explanation for these major differences. As such, the question as to whether the observed increases in plasma membrane cholesterol are large enough to drive all molecular interactions remain unanswered. As for the mechanism, I suggested they should either employ a more generally accepted mechanistic model (e.g., biomolecular condensates) independent of the controversial lipid raft model or provide a more quantitative correlation between the cholesterol level and the lipid raft functionality if they wish to stick with the lipid raft model. The authors took an ambitious approach of explaining the cholesterol-Wnt signaling link by both lipid rafts and biomolecular condensates models. Their new data are generally consistent with the idea that many players in Wnt signalosomes are clustered in the plasma membrane of Apc-truncated CRC cells. However, they do not fully explain how cholesterol enrichment observed in the plasma membrane of drives molecular interactions and clustering. Specifically,

1) FLIP-FRET of a cholesterol probe, D4H, shows that D4H is much more clustered in unstimulated Apc-truncated cells (Fig 5I and Fig S9B), consistent with higher plasma membrane cholesterol contents in these cells. However, FLIP-FRET between D4H and a raft marker, tH-RFP, shows only a small increase (Fig 5H and Fig S9A) in unstimulated Apc-truncated cells, indicating that cholesterol-clustering and the raft activity (as defined by the authors) are not quantitatively correlated. Furthermore, a better quantitative correlation is found between cholesterol clustering and some molecular interactions between proteins and lipids (e.g., Chol-Fzd7, Chol-LRP6, Fzd7-LRP6) in Wnt signalosomes, suggesting that elevated cholesterol in the plasma membrane of Apc-truncated cells selectively enhances molecular interactions among proteins and lipids within Wnt signalosomes in a lipid rafts-independent manner. New data on Ld-Lo miscibility do not add much to mechanistic understanding of the cholesterol-mediated Wnt activation because their interpretation is not straightforward and implications indirect.

2) The authors have also added new super-resolution imaging data to show that the formation of Wnt signalosomes (or nanoclusters or condensates) is facilitated in Apc-truncated cells. Although the new data are qualitatively consistent with the FLIP-FRET data, the differences observed between Apc-WT and APC-truncated cells are modest at best, except in the case of Dvl1 self-clustering. As such, they fail to provide compelling mechanistic insight.

3) The authors did not preclude the possibility that enhanced protein-protein interactions in Wnt signalosomes of Apc-truncated cells is mainly due to the increased availability of key cytosolic proteins, such as axin. That is, greatly impaired scaffolding activity of truncated Apc would allow them to readily translocate to the plasma membrane and interact with membrane components (i.e., Chol, Fzd, Lrp6) of Wnt signalosomes. They could test this possibility for example by depleting Apc in Apc-WT cells.

Reviewer #3 (Remarks to the Author):

The authors have now introduced new data in the revised manuscript, which have strengthened their paper. However, I still have a concern about the model (Fig 10).

The authors propose that Wnt-ligand signaling and APC truncations both result in changes in cholesterol homeostasis, membrane rigidity, and lipid rafts. The authors suggest that Wnt ligand-mediated signaling does not transcriptionally regulate cholesterol production (instead changes the membrane organization of the existing cholesterol pool), whereas APC truncations promote the synthesis of new plasma membrane cholesterol to induce Wnt receptor clustering and further activation.

How oncogenic beta-catenin mutants fit into their cholesterol model is unclear; they showed that the cell line HCT116, which has an oncogenic (nondegradable) beta-catenin mutant, does not exhibit elevated cholesterol. However, chemical inhibition of GSK3 (which mimics nondegradable beta-catenin) was shown to elevate cholesterol. The discrepancy in these results muddies the author's conclusions and should be reconciled. Finally, an APC-beta-catenin double mutant was shown to exhibit a greater change in cholesterol levels than a heterozygous APC mutant alone. As stated by the authors, "the levels of free cholesterol were proportional to the magnitude of Wnt signaling dysregulation."

To this reviewer, a better way to present the authors' data is to draw it as two separate plasma membrane events: A) Wnt ligand activation induces an acute change in plasma membrane cholesterol and receptor clustering that does not involve cholesterol gene transcription and B) prolonged Wnt signaling in CRC (with APC or beta-catenin mutations) activates cholesterol gene transcription followed by receptor clustering and activation. Induction of cholesterol gene transcription would represent a positive feedback mechanism. Trying to link these two events (as shown in Fig. 10) makes the model confusing.

RESPONSE TO REVIEWERS' COMMENTS

Reviewer #1 (Remarks to the Author):

A large majority of colorectal cancer (CRC) cells harbor variable Apc truncations, leading to hyperactivation of Wnt-beta catenin signaling. The key finding in this manuscript is that Apc truncations increase plasma membrane cholesterol, which then promotes molecular interactions in Wnt signalosomes. It is an interesting and potentially very important new discovery considering that it has been believed that truncated Apc causes Wnt-beta catenin signaling hyperactivation mainly because it cannot effectively support the steady-state beta catenin degradation. However, to establish biological significance and translational potential of this new finding, one must fully understand the mechanisms by which Apc truncations alter cholesterol homeostasis and Wnt signaling activity. Both cholesterol homeostasis and Wnt signaling are highly complex processes involving many protein and lipid components and thus full mechanistic elucidation of the Apc-cholesterol-Wnt signaling link would require thorough, systematic and quantitative studies.

In the past two reviews, my critiques focused on two questions: (1) How does Apc truncation alter cellular cholesterol homeostasis and the local cholesterol level at the plasma membrane and do variable Apc truncations have similar or different effects? (2) Are these cholesterol increases large enough to drive molecular interactions within Wnt signalosomes and, if so, how? Specifically, I raised concerns about the validity, credibility, and relevance of their model systems, the accuracy and sensitivity of their cholesterol quantification methods, and their data interpretation. Although the authors have made significant efforts to address these concerns technically, key mechanistic questions still remain answered.

1. As for the question regarding how Apc truncations alter cellular cholesterol homeostasis, the authors have collected RNAseq and western blot data primarily to identify cholesterol homeostasis proteins that are affected by the Apc truncation. They found that Apc truncation downregulated major players of cholesterol efflux, ABCA1 and ABCG1, and upregulated LRP8 and Scarf1 involved in cholesterol uptake, while also downregulating some proteins involved in de novo synthesis (Pmvk, Mvk and Mvd).

Response #1: We respectfully disagree with reviewer #1. Although this reviewer correctly states that RNA expression of genes mediating cholesterol efflux and uptake decreased and increased, respectively, they incorrectly state that *de novo* cholesterol synthesis is decreased overall. Indeed, the RNAseq findings from our orthogonal colorectal cancer (CRC) models indicate that the expression of genes linked to cholesterol synthesis, other than those mentioned by this reviewer, were significantly upregulated in human CRC patients (genes include: *GGPS1*, *FDPS*, *FDFT1*, *EBP*, *DHCR7*, *INSIG1*, *INSIG2*, *CYB5R3*, *SCAP* and *SREBF2*; **Figure S1E**). Additionally, the RNA expression of *Mvd*, *Mvk* and *Pmvk* were upregulated in mouse colonic crypts (**Figure 2J**), while *Mvk* RNA expression was also upregulated in mouse cultured colonocytes (**Figure 2I**), which contradicts the reviewer's statement. Finally, we observed a directional dysregulation of other genes involved in cholesterol homeostasis consistent with the accumulation of free cholesterol in cells. For example, the RNA expression of a gene involved in the conversion of free cholesterol to cholesterol esters, i.e., *Soat2*, decreased significantly (**Figure 2P and S3**). Together, these data suggest that mutant APC drives the global dysregulation of cholesterol homeostatic genes in a model independent manner, consistent with an increase in the levels of plasma membrane free cholesterol (**Figure 1, 4 and S1**). We have edited the manuscript text to further clarify this point.

Although these results are useful and consistent with the idea that Apc truncation disrupts cholesterol homeostasis, they do not explain how Apc truncation enriches cholesterol in the plasma membrane. As mentioned above, cholesterol homeostasis is a highly complex system that controls multiple processes, including de novo synthesis, storage, uptake, efflux, and transport. Also, it is not fully understood how subcellular local cholesterol levels (e.g., ER and the plasma membrane) are correlated. It is therefore unclear how all these altered proteins contribute to cholesterol enrichment in the plasma membrane. For example, how does downregulation of cholesterol biosynthetic proteins lead to cholesterol enrichment in the plasma membrane? I fully agree with the Reviewer 3 who recommended that the authors should perform systematic siRNA gene knockdown (and/or overexpression) to dissect the effects of altered proteins to the plasma membrane cholesterol enrichment, which for some reason the authors opted not to follow. Instead, they performed cell growth in the cholesterol-depleted media, the results of which do not explain how exactly cholesterol is enriched in the plasma membrane of Apc-truncated CRC cells.

Response #2: We agree with reviewer #1 that measuring filipin III (free cholesterol) in colonocytes grown under cholesterol-starved conditions (**Figure 3**) and RNAseq experiments (**Figure 2 and S1-S3**) as described in our manuscript, do not fully elucidate the mechanism by which truncated mutant APC increases plasma membrane cholesterol. Importantly, our findings (described above) serve as the initial steps in deciphering the effects of mutant APC on cellular cholesterol regulation. Without these data, the suggested systematic examination of gene function using siRNA knockdown with respect to cholesterol homeostasis would likely become a fishing expedition involving hundreds of genes. As mentioned in the manuscript, the aforementioned experiments provide "*mechanistic insight into the cellular processes altered by oncogenic truncated APC which could perturb cholesterol homeostasis*" and corroborate "*the functional relevance of the changes observed in the expression of cholesterol homeostatic genes driven by oncogenic truncated APC*". We have amended the manuscript text to clarify these points.

We agree that future experiments will involve the examination of a “*highly complex system that controls multiple processes*” (as stated by this reviewer). However, to carefully “*dissect the effects of altered proteins to the plasma membrane cholesterol enrichment*” (as stated by this reviewer), we would need to perform an exhaustive series of genetic and functional experiments to conclusively address these questions. For these reasons, we consider the reviewer’s proposed experiments beyond the scope of our manuscript.

2. In the previous review, I requested the authors to provide convincing evidence that the cholesterol enrichment in the plasma membrane is sufficient (i.e., above a defined threshold) for driving proposed molecular interactions and also to provide a clear mechanistic explanation for their results. As for the magnitude of cholesterol enrichment, although the authors describe it as “striking” throughout the manuscript, their data do not quantitatively support the claim. Specifically, filipin staining, which is their primary method, shows modest 20-50% increases in plasma membrane cholesterol caused by double-allele Apc truncation (Fig 2B, C, D). Ironically, much larger increases were detected by the supplementary Amplex Red method but the authors did not provide any explanation for these major differences. As such, the question as to whether the observed increases in plasma membrane cholesterol are large enough to drive all molecular interactions remain unanswered.

Response #3: We respectfully disagree with the statement that “*In the previous review, I requested the authors to provide convincing evidence that the cholesterol enrichment in the plasma membrane is sufficient (i.e., above a defined threshold) for driving proposed molecular interactions and also to provide a clear mechanistic explanation for their results. As for the magnitude of cholesterol enrichment, although the authors describe it as “striking” throughout the manuscript, their data do not quantitatively support the claim. Specifically, filipin staining, which is their primary method, shows modest 20-50% increases in plasma membrane cholesterol caused by double-allele Apc truncation... As such, the question as to whether the observed increases in plasma membrane cholesterol are large enough to drive all molecular interactions remain unanswered*”. Our cogent quantitative data derived from orthogonal CRC models demonstrate that intervention with free cholesterol-lowering drugs, i.e., M β CD and mevastatin, can rescue the effects of truncated mutant APC in terms of plasma membrane rigidity (**Figure S7 and S13**), Wnt signaling-associated proteolipid interactions (**Figure 5-7**), clustering (**Figure 7**) and β cat activity (**Figure 9 and S13**) to levels similar to or lower than WT APC healthy cells. Additionally, we confirmed that free cholesterol alone is sufficient to increase Wnt receptor homo- and hetero-clustering (**Figure 9**) as well as modulate β cat activity *in vivo* (**Figure 9**) in a dose-dependent manner.

With regard to the magnitude of cholesterol enrichment, reviewer #1 disagrees with our use of the adjective “striking” to describe the changes driven by mutant truncated APC. Instead, the reviewer proposes that “*...filipin staining, which is their primary method, shows modest 20-50% increases in plasma membrane cholesterol caused by double-allele Apc truncation (Fig 2B, C, D)*”. Firstly, our data clearly show that cholesterol increases by a magnitude higher than 50% or 1.5-fold. For example, AfGC homo mice (Apc 580/580) expressing truncated mutant APC from both Apc alleles showed a 3.3-fold or 230% increase in free cholesterol 5 weeks post expression of APC 580. Additionally, mutant cancer stem cells, which are proposed to fuel tumor development, displayed a 2.8-fold or 180% increase in free cholesterol. Finally, CRC-PDO and Crispr APC human organoids, which constitute a translationally relevant model expressing truncated mutant APC, displayed a 2.0-fold or 100% (CRC-PDO) and 1.9-fold or 90% (Crispr APC), respectively. This is noteworthy, since a typical plasma membrane displays a 10 to 30 mol% of cholesterol while a “low” cholesterol plasma membrane content is considered to be <10 mol%¹. In contrast, a “high”, “very high” or beyond the cholesterol solubility threshold (CST) level of plasma membrane cholesterol is achieved when a ~50 (67-400% change), >50 (> 400% change) or 66 mol% (560% change) of cholesterol, respectively, is detected¹. Therefore, the changes reported in our manuscript are not modest in nature. In order to more accurately describe the level of plasma membrane cholesterol in APC mutant cells, we have amended the list of suitable (high/large/significant/strong/chronic) descriptors. We have also amended our figures to clearly and consistently display the cholesterol changes described in the text as fold change.

As for the mechanism, I suggested they should either employ a more generally accepted mechanistic model (e.g., biomolecular condensates) independent of the controversial lipid raft model or provide a more quantitative correlation between the cholesterol level and the lipid raft functionality if they wish to stick with the lipid raft model. The authors took an ambitious approach of explaining the cholesterol-Wnt signaling link by both lipid rafts and biomolecular condensates models. Their new data are generally consistent with the idea that many players in Wnt signalosomes are clustered in the plasma membrane of Apc-truncated CRC cells. However, they do not fully explain how cholesterol enrichment observed in the plasma membrane of drives molecular interactions and clustering. Specifically,

1) FLIP-FRET of a cholesterol probe, D4H, shows that D4H is much more clustered in unstimulated Apc-truncated cells (Fig 5I and Fig S9B), consistent with higher plasma membrane cholesterol contents in these cells. However, FLIP-FRET between D4H and a raft marker, tH-RFP, shows only a small increase (Fig 5H and Fig S9A) in unstimulated Apc-truncated cells, indicating that cholesterol-clustering and the raft activity (as defined by the authors) are not quantitatively correlated.

Response #4: To clarify the link between our FLIM-FRET findings and their mechanistic significance, we reported that the increase in D4H-D4H FRET efficiency directly correlated with an increase in cholesterol clustering, as noted by the reviewer. Importantly, the increase in D4H-tH FRET efficiency indicates an increase of cholesterol molecules within plasma membrane rafts or liquid ordered (L_o) domains. Clearly, these are not mutually exclusive cellular events as an increase of cholesterol clustering (increased interactions between cholesterol molecules) can occur independently of an increase in the levels of cholesterol in raft domains (increased D4H-tH interactions) and vice versa². Indeed, the fact that both D4H-D4H and D4H-tH FRET efficiency significantly increased demonstrates the dysregulation of both cholesterol clustering and its levels in lipid rafts driven by truncated mutant APC. Furthermore, we employed robust methodologies, i.e., T_{misc} of phase separated GPMVs which identifies membrane subdomains (L_o and L_d), and imaged lateral structure to assess L_o and L_d domain stability. Thus, we were able to quantitatively correlate mutant APC-dependent increases in plasma membrane cholesterol with cholesterol clustering and L_d domain stability.

Furthermore, a better quantitative correlation is found between cholesterol clustering and some molecular interactions between proteins and lipids (e.g., Chol-Fzd7, Chol-LRP6, Fzd7-LRP6) in Wnt signalosomes, suggesting that elevated cholesterol in the plasma membrane of Apc-truncated cells selectively enhances molecular interactions among proteins and lipids within Wnt signalosomes in a lipid rafts-independent manner.

Response #5: This statement by the reviewer supports the notion that truncated mutant APC-driven changes in Wnt receptor and effector organization at the plasma membrane would lead to better quantitative correlated data. We strongly agree with this notion as it is highly plausible that alterations in the localization of interacting members of Wnt signalosomes, e.g., cholesterol-Fzd7, cholesterol-LRP6, Fzd7-LRP6, would have similar consequences due to their shared localization-dependent nature. Additionally, this notion further supports our response to query #4.

New data on L_d - L_o miscibility do not add much to mechanistic understanding of the cholesterol-mediated Wnt activation because their interpretation is not straightforward and implications indirect.

Response #6: We apologize to the reviewer for the lack of clarity associated with our written interpretation of the quantitative analysis of temperature dependence on L_d and L_o domain stability. We have amended the manuscript text to clarify the implications of these data in terms of the mechanisms by which truncated mutant APC drives loss of plasma membrane homeostasis.

2) The authors have also added new super-resolution imaging data to show that the formation of Wnt signalosomes (or nanoclusters or condensates) is facilitated in Apc-truncated cells. Although the new data are qualitatively consistent with the FLIP-FRET data, the differences observed between Apc-WT and APC-truncated cells are modest at best, except in the case of Dvl1 self-clustering. As such, they fail to provide compelling mechanistic insight.

Response #7: We respectfully disagree with the reviewer regarding the significance of the highly-quantitative new super-resolution microscopy data documenting the effects of truncated mutant APC on Dvl1 structure and organization at the plasma membrane. We are unsure what the reviewer means by "Dvl1 self-clustering", therefore we will highlight the importance of each parameter presented in **Supplementary Figure 12** and their mechanistic implications. Firstly, we observed a mutant APC dependent increase in Dvl1 cluster area (**Crispr APC:** 1.3-fold, 30%; **CRC-PDO:** 1.2-fold, 20%) (**Figure S12B**), which suggests overall larger Wnt signaling platforms, and potentially stronger Wnt signaling activation. Secondly, consistent with the increase in Dvl1 cluster area, we observed an increase in the frequency of formation of relatively large Dvl1 clusters (**Crispr APC:** 1.5-fold, 50%; **CRC-PDO:** 1.3-fold, 30%) (**Figure S12C**). Thirdly, we observed a significant increase in the total number of Dvl1 molecules (**Crispr APC:** 1.6-fold; **CRC-PDO:** 1.5-fold) (**Figure S12D**) and their density (**Crispr APC:** 2.2-fold, 120%; **CRC-PDO:** 2.3-fold, 130%) (**Figure S12E**) within clusters, which suggests an increase in the translocation of Dvl1 molecules to the plasma membrane and significantly impaired scaffolding activity associated with truncated mutant APC resulting from the increased availability of cytosolic Dvl1 and, potentially, other cytosolic members of Wnt signalosomes, e.g., Axin. Consequently, this would increase the interactions of Dvl1 with Wnt-associated membrane components, e.g., cholesterol, Fzd7 and LRP6. Fourthly, the percentage of Dvl1 molecules within clusters (of the total number of Dvl1 molecules detected; % of total) increased significantly (**Crispr APC:** 1.6-fold, 60%; **CRC-PDO:** 2.0-fold, 100%) (**Figure S12F**), which is consistent with the notion that truncated mutant APC scaffolding activity is impaired and drives an increase in the membrane translocation of Dvl1 molecules and its interactions with membranous members of Wnt signalosomes. Finally, the total number of clusters containing Dvl in cells (**Crispr APC:** 3.2-fold, 220%; **CRC-PDO:** 3.2-fold, 220%) (**Figure S12G**) and the cellular Dvl1 cluster density (**Crispr APC:** 2.7-fold, 170%; **CRC-PDO:** 2.8-fold, 180%) (**Figure S12H**) was increased in truncated mutant APC cells. The percentage change displayed by these multiple parameters driven by truncated mutant APC are not just statistically (quantitatively determined) robust but also, in multiple different cases, physiologically relevant. Consequently, we believe the consistency and magnitude of our data provide novel mechanistic insight into the effects of mutant APC on plasma membrane biophysical interactions. That being said, we relied primarily on statistical change to indicate relevant observations in our manuscript, since the biological

significance of a given change is likely to depend on the specific gene/protein of interest and on the experimental context³. We have modified aspects of our data presentation to enhance the clarity of the changes observed.

3) The authors did not preclude the possibility that enhanced protein-protein interactions in Wnt signalosomes of Apc-truncated cells is mainly due to the increased availability of key cytosolic proteins, such as axin. That is, greatly impaired scaffolding activity of truncated Apc would allow them to readily translocate to the plasma membrane and interact with membrane components (i.e., Chol, Fzd, Lrp6) of Wnt signalosomes. They could test this possibility for example by depleting Apc in Apc-WT cells.

Response #8: In response to query #7, we have addressed the notion concerning changes in protein-protein interactions due to increased availability of Wnt-associated cytosolic proteins and their translocation to the plasma membrane driven by truncated mutant APC.

Reviewer #3 (Remarks to the Author):

The authors have now introduced new data in the revised manuscript, which have strengthened their paper. However, I still have a concern about the model (Fig 10).

The authors propose that Wnt-ligand signaling and APC truncations both result in changes in cholesterol homeostasis, membrane rigidity, and lipid rafts. The authors suggest that Wnt ligand-mediated signaling does not transcriptionally regulate cholesterol production (instead changes the membrane organization of the existing cholesterol pool), whereas APC truncations promote the synthesis of new plasma membrane cholesterol to induce Wnt receptor clustering and further activation.

Response #9: We agree with reviewer #3 that the expression of various truncated APC proteins drive changes in plasma membrane cholesterol, rigidity and lipid raft domains. However, to clarify, we propose that “free cholesterol selectively activates canonical Wnt signaling over non-canonical Wnt signaling” as supported by the findings of Sheng et. al⁴. Additionally, we highlight data which describe the effect of cellular Wnt stimulation on Wnt receptor lipid raft localization (enrichment of activated Wnt receptors in cholesterol-enriched microdomains)⁴ and free cholesterol transbilayer asymmetry (enrichment of cholesterol at the plasma membrane inner leaflet)². Based on this premise, our proposed mechanistic model supports the notion that cellular Wnt stimulation leads to a greater magnitude of modulation of the various membrane related phenotypes in the context of a mutant Apc background when compared to wild type (WT). Consequently, our “four part” mechanistic model (**Figure 10**) integrates both newly discovered and previously established findings. We apologize for the lack of clarity in regard to this matter. We have revised our description to highlight the novel details and complexities of our proposed model.

Regarding the following statement: “*The authors suggest that Wnt ligand-mediated signaling does not transcriptionally regulate cholesterol production*”, we respectfully disagree with this reviewer. We have not proposed anywhere in our manuscript the notion that, contrary to truncated APC, cellular Wnt stimulation does not regulate free cholesterol in a transcriptional manner. Indeed, we consider it is plausible that Wnt ligand-dependent alterations in plasma membrane biochemical and biophysical properties involves transcriptional modulation. However, we have focused on describing the gene transcriptional changes associated with a mutant APC background. We have addressed this concern by editing the manuscript text and the legend to **Figure 10**.

Regarding the following statement: “*...whereas APC truncations promote the synthesis of new plasma membrane cholesterol to induce Wnt receptor clustering and further activation.*”, we would like to clarify a few details. We have provided quantitative evidence suggesting that the *de novo* cholesterol synthesis pathway plays a role in the mutant APC-dependent changes in plasma membrane homeostasis. In addition, our data suggest that cholesterol uptake and efflux play a key role in this aberrant response.

How oncogenic beta-catenin mutants fit into their cholesterol model is unclear; they showed that the cell line HCT116, which has an oncogenic (nondegradable) beta-catenin mutant, does not exhibit elevated cholesterol. However, chemical inhibition of GSK3 (which mimics nondegradable beta-catenin) was shown to elevate cholesterol. The discrepancy in these results muddies the author’s conclusions and should be reconciled. Finally, an APC-beta-catenin double mutant was shown to exhibit a greater change in cholesterol levels than a heterozygous APC mutant alone. As stated by the authors, “the levels of free cholesterol were proportional to the magnitude of Wnt signaling dysregulation.”

Response #10: With regard to our rationale to examine the effects of mutant β -catenin (β cat), many of our experiments employed isogenic cultured colonocytes displaying a single *Apc* or double *Apc*/ β cat mutation and compared them to colonocytes on a WT background. By interrogating the effects of these distinct genotypes, we were able to establish a “gene dose” effect (*Apc* -/+ β cat -/+ > *Apc* -/+ > *Apc* +/+) on plasma membrane cholesterol (**Figure 2**), rigidity (**Figure S7**), RNA

expression (**Figure 2**) and multiple proteolipid interactions (**Figures 5-7**), e.g., Fzd7-cholesterol. Collectively, these findings not only directly link Wnt signaling dysregulation to the loss of plasma membrane homeostasis but also highlight the important role of the magnitude of Wnt signaling dysregulation in driving CRC. The latter notion along with the “just right” Wnt dysregulation hypothesis⁵ and the ability of statins to suppress some of the examined phenotypes, e.g., plasma membrane rigidity, suggest that “therapeutic approaches aimed at restoring plasma membrane cholesterol homeostasis and spatiotemporal dynamics could prove an effective strategy to reduce cancer risk”.

We respectfully disagree with reviewer #3 regarding the following statement “...they showed that the cell line HCT116, which has an oncogenic (nondegradable) beta-catenin mutant, does not exhibit elevated cholesterol. However, chemical inhibition of GSK3 (which mimics nondegradable beta-catenin) was shown to elevate cholesterol. The discrepancy in these results muddies the author’s conclusions and should be reconciled”. More accurately, our data show that although HCT116 (Apc +/+ β cat -/+) expresses mutant β cat, western blot experiments demonstrate that HCT116 cells **exhibit significantly lower β cat protein levels** when compared to other models, which display higher levels of free cholesterol such as IMCE (Apc -/+), IMCE β cat (Apc -/+ β cat -/+), CRC cell lines and organoids (**Figure 1C**). Consequently, the relatively weaker downstream activation of Wnt signaling in the HCT116 model results in a lower dysregulation of plasma membrane homeostasis, e.g., low plasma membrane cholesterol. In contrast, the incubation of cultured colonocytes with CHIR99021 (GSK3 β inhibitor) for 72 h leads to a robust activation of β cat and downstream Wnt signaling. Thus, CHIR99021 incubation significantly dysregulates the biochemical and biophysical properties of the plasma membrane to that observed in CRC models exhibiting elevated levels of free cholesterol. We have revised the text to clarify this point.

To this reviewer, a better way to present the authors’ data is to draw it as two separate plasma membrane events: A) Wnt ligand activation induces an acute change in plasma membrane cholesterol and receptor clustering that does not involve cholesterol gene transcription and B) prolonged Wnt signaling in CRC (with APC or beta-catenin mutations) activates cholesterol gene transcription followed by receptor clustering and activation. Induction of cholesterol gene transcription would represent a positive feedback mechanism. Trying to link these two events (as shown in Fig. 10) makes the model confusing.

We have modified **Figure 10** in order to clarify the mechanistic link between mutant APC, loss of plasma membrane homeostasis and colon tumor formation.

References

1. Subczynski, W. K., Pasenkiewicz-Gierula, M., Widomska, J., Mainali, L. & Raguz, M. High cholesterol/low cholesterol: Effects in biological membranes Review. *Cell Biochem. Biophys.* **75**, 369 (2017).
2. Liu, S. L. *et al.* Orthogonal lipid sensors identify transbilayer asymmetry of plasma membrane cholesterol. *Nat. Chem. Biol.* **13**, 268 (2017).
3. McCarthy, D. J. & Smyth, G. K. Testing significance relative to a fold-change threshold is a TREAT. *Bioinformatics* **25**, 765 (2009).
4. Sheng, R. *et al.* Cholesterol selectively activates canonical Wnt signalling over non-canonical Wnt signalling. *Nat. Commun.* **5**, (2014).
5. Albuquerque, C. *et al.* The ‘just-right’ signaling model: APC somatic mutations are selected based on a specific level of activation of the β -catenin signaling cascade. *Hum. Mol. Genet.* **11**, 1549–1560 (2002).